# Quantifying flood-water impacts on a lake water budget via volume-dependent transient stable isotope mass balance

Janie Masse-Dufresne[1], Florent Barbecot[2], Paul Baudron[1] and John Gibson[3,4]

[1]Polytechnique Montréal, Department of Civil, Geological and Mining Engineering, Montreal, QC H3T 1J4, Canada

[2] Geotop-UQAM, Department of Earth and Atmospheric Sciences, Montreal, QC H2X 3Y7, Canada

[3] InnoTech Alberta, 3-4476 Markham Street, Victoria, BC V8Z 7X8, Canada

[4] University of Victoria, Department of Geography, Victoria, BC V8W 3R4, Canada

*Correspondence to*: Janie Masse-Dufresne (janie.masse-dufresne@polymtl.ca)

**Abstract.** Isotope mass balance models have undergone significant developments in the last decade, demonstrating their utility for assessing the spatial and temporal variability of hydrological processes, and revealing significant value for baseline assessment in remote and/or flood-affected settings where direct measurement of surface water fluxes to lakes (i.e., stream gauging) are difficult to perform. In this study, we demonstrate that isotopic mass balance modelling can be used to provide evidence of the relative importance of direct flood-water inputs and temporary subsurface storage of flood-water at ungauged

lake systems. A volume-dependent transient isotopic mass balance model was developed for an artificial lake (named Lake A) in southern Quebec (Canada). This lake typically receives substantial flood-water inputs during the spring freshet period, as an ephemeral hydraulic connection with a 150,000-km$^2$ large watershed is established. First-order water flux estimates to Lake A allow for impacts of flood-water inputs to be highlighted within the annual water budget. The isotopic mass balance model has revealed that groundwater and surface water inputs account for 60-71 % and 39-28 % of the total annual water

inputs to Lake A respectively, which demonstrates an inherent dependence of the lake on groundwater. However, when considering the potential temporary subsurface storage of flood-water, the partitioning between groundwater and surface water inputs tends to equalize, and Lake A water budget is found to be more resilient to groundwater quantity and quality changes. Our findings suggest not only that flood-water fluxes to Lake A have an impact on its dynamics during springtime, but significantly influence its long-term water balance and help to inform, understand and predict future water quality variations.

From a global perspective, this knowledge is useful for establishing regional-scale management strategies for maintaining water quality at flood-affected lakes, for predicting the response of artificial recharge systems in such settings, and to mitigate impacts due to land-use and climate changes.

**1 Introduction**

Lakes are complex ecosystems which play a valuable economic, social and environmental role within watersheds (Kløve et al., 2011). In fact, lacustrine ecosystems can provide a number of ecosystem services, such as biodiversity, water supply, recreation and tourism, fisheries and sequestration of nutrients (Schallenberg et al., 2013). The actual benefits that can be provided by lakes depend on the water quality, and poor resilience to water quality changes can lead to benefit losses (Mueller et al., 2016). Globally, the quantity and quality of groundwater and surface water resources are known to be affected by land-use (Baudron et al., 2013; Cunha et al., 2016; Lerner and Harris, 2009; Scanlon et al., 2005) and climate changes (Delpla et al., 2009). As both surface water and groundwater contribute to lake water balances (Rosenberry et al., 2015), changes that affect the surface water/groundwater apportionment can potentially modify or threaten lake water quality (Jeppesen et al., 2014). Understanding hydrological processes in lakes can help to depict the vulnerability and/or resilience of a lake to pollution (Rosen, 2015) as well as to invasive species (Walsh et al., 2016) and thus secure water quantity and quality over time for drinking water production purposes (Herczeg et al., 2003). In Quebec (Canada), there are an important number of municipal wells that receive contributions from surface water resources (i.e., lakes or rivers) and are thus performing unintentional (Patenaude et al., 2020) or intentional (Masse-Dufresne et al., 2021; Masse-Dufresne et al., 2019) bank filtration.

Over the past few decades, significant developments have been made in the application of isotope mass balance models for assessing the spatial and temporal variability of hydrological processes in lakes; most notably, the quantification of groundwater and evaporative fluxes (Herczeg et al., 2003; Bocanegra et al., 2013; Gibson et al., 2016; Arnoux et al., 2017b). In remote environments, such as in northern Canada, application of isotopic methods is particularly convenient, as direct measurements of surface water and groundwater fluxes is time-consuming, expensive, and difficult (Welch et al., 2018). Isotopic mass balance models can notably be applied to ungauged lake systems to efficiently characterize the impacts of floods on water apportionment (Haig et al., 2020). While isotopic frameworks were successfully used to assess the relative importance of flood-water inputs to lakes (Turner et al., 2010; Brock et al., 2007), no attempt was made at evaluating the timing of the flood-water inputs and to differentiating between the role of i) direct flood-water inputs and ii) temporary subsurface storage of flood-water on a lake's annual water budget. In this study, we define the direct inputs refer to the flood-water that enter a lake via the surface (e.g., by inundating and/or flowing through a stream), while temporary subsurface storage of flood-water encompasses the flood-water-like inputs that reach the lake via subsurface (e.g., through floodplain recharge or bank storage). To gain information on the timing of hydrological processes, one may use a transient and short time step isotopic mass balance. A previous study by Zimmermann (1979) used a transient isotope balance to estimate groundwater inflow and outflow, evaporation, and residence times for two young artificial groundwater lakes near Heidelberg, Germany, although these lakes had no surface water connections, and volumetric changes were considered negligible. Zimmermann (1979) showed that the lakes were actively exchanging with groundwater, which controlled the long-term rate of isotopic enrichment to isotopic steady state, but the lakes also responded to seasonal cycling in the magnitude of water balance processes. While informative, Zimmermann (1979) did not attempt to build a predictive isotope mass balance model, but rather used a best-fit approach to

obtain a solitary long-term estimate of water balance partitioning for each lake. Petermann et al. (2018) also constrained groundwater connectivity for an artificial lake near Leipzig, Germany, with no surface inlet or outlet. By comparing groundwater inflow rates obtained via stable isotope and radon mass balances on a monthly time-step, Petermann et al. (2018) highlighted the need to consider seasonal variability when conducting lake water budget studies. Our approach builds on that

of Zimmermann (1979) and Petermann et al. (2018), developing a predictive model of both atmospheric and water balance controls on isotopic enrichment, and accounting for volumetric changes on a daily time step.

The main objective of this study is to provide evidence of the relative importance of direct flood-water inputs and temporary subsurface storage of flood-water at ungauged lake systems using an isotopic mass balance model. To do so, we first aim to establish an isotopic framework based on the local water cycle, to verify the applicability of isotopic mass balance in the

70 present setting, as contrasting isotopic signatures are required between various water reservoirs and fluxes, including flood-water inputs. Secondly, we quantify the water budget according to two reference scenarios (A and B) to grasp the impact of site-specific uncertainties on the computed results. Then, we analyze the temporal variability of the groundwater inputs and the sensitivity of the lake to flood-water driven pollution. Finally, we demonstrate the implications of flood-water-like subsurface inputs on the water balance partition.

The water balance is computed via a volume-dependent transient isotopic mass balance model, which is applied to predict the daily isotopic response of an artificial lake in Canada that is ephemerally connected to a 150,000 km$^2$ watershed during spring freshet. During these flood events, the surficial water fluxes entering the study lake are not constrained in a gaugeable river or canal but occur over a 1-km wide surficial flood area. Our study period spans a flood with an average recurrence interval of 100 years, and is therefore an example of the response of the system to a major hydrological event.

**2 Study site**

**2.1 Geological and hydrological settings**

The study site is located in the area of Greater Montreal and is bordering the Lake Deux-Montagnes (further referred to as Lake DM), which corresponds to a widening of the Ottawa River at the confluence with St-Lawrence River in Quebec (Canada) (Fig.1). The Ottawa River is the second largest river in eastern Canada, draining a watershed of approximately 150000 km$^2$

(MDDELCC, 2015). The water level of Lake DM is partly controlled by flow regulation structures (e.g., hydroelectric dams) upstream on the Ottawa River. Lake DM water levels also show seasonal fluctuations in response to precipitations and snowpack melting over the Ottawa River watershed. High water levels at Lake DM are typically observed during springtime (April-May) and, less prominently, during autumn (November-December), while lowest water levels normally occur at the end of the summer (September) (Centre d'Expertise Hydrique du Québec, 2020).

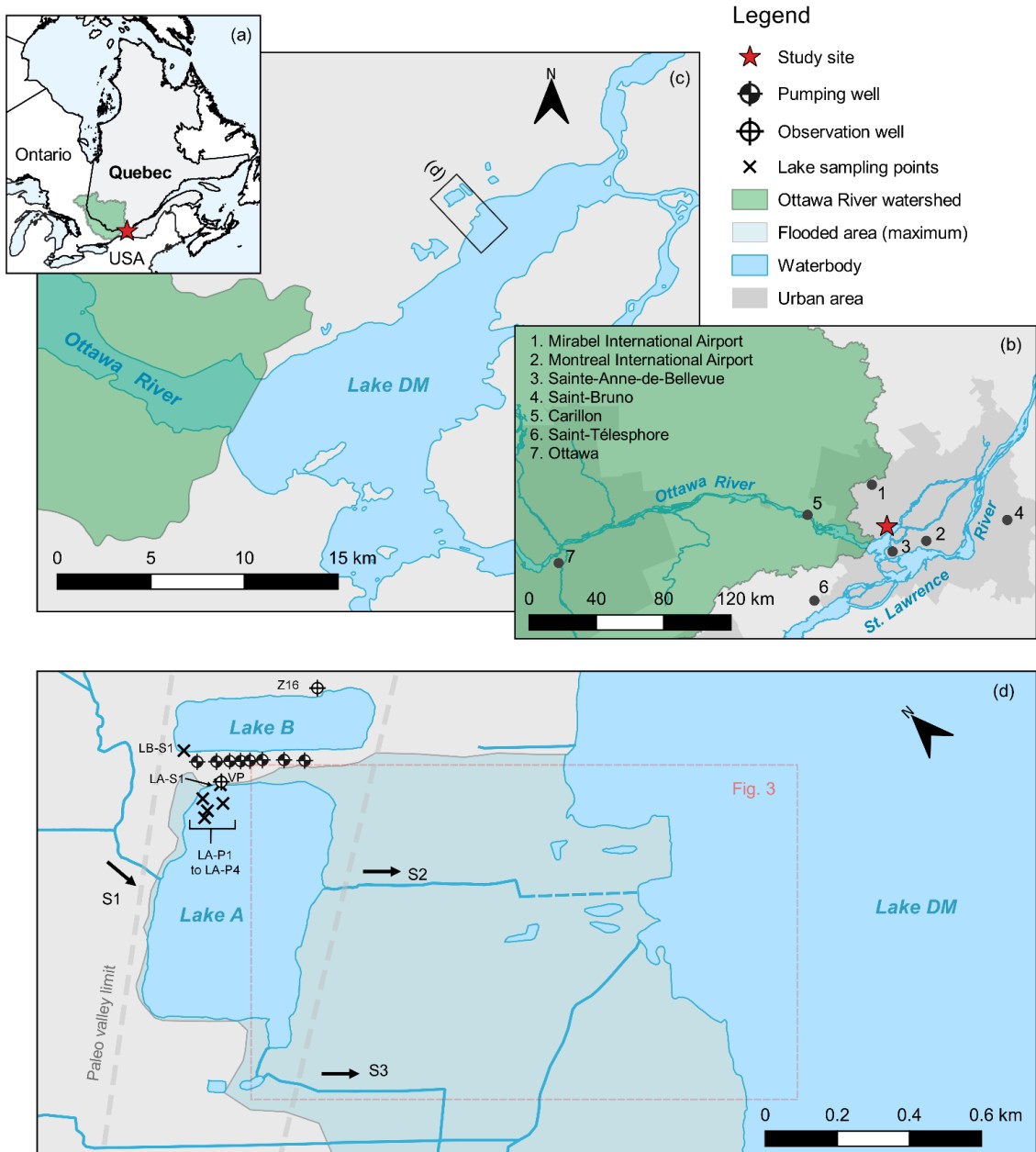

**Figure 1.** (a-c) Location of the study site, relative to the Ottawa River watershed, Lake Deux-Montagnes (DM) and the area of Greater Montreal, (d) location of Lake A and Lake B relative to Lake DM and schematic representation of the hydrogeological context. The grey dashed lines illustrate the approximative extent of the paleo valley. LA-S1 and LB-S1 are surface water sampling points at Lake A and Lake B, respectively. LA-P1 to LA-P4 correspond to vertical profile sampling locations at Lake A. The maps were created from openly available data and used in accordance with the Open Government Licence – Canada or the Open Data Policy, M-13-13 of the United States Census Bureau. Detailed source information is provided in Appendix A.

Lake A ($2.79 \times 10^5$ m$^2$) and Lake B ($7.6 \times 10^4$ m$^2$) are two small artificial lakes created from sand-dredging activities and are located at approximately 1 km from the shore of Lake DM. The dredging is still on-going at Lake A, while it ceased a few decades ago at Lake B. Both lakes are approximately 20 m deep (Masse-Dufresne et al., 2019) and were excavated within alluvial sands which were deposited in a paleo valley extending in the NE-SW direction and carved into the Champlain Sea Clays (Ageos, 2010). Lithostratigraphic data (i.e., well logs) suggest that the paleo valley is approximately 600 m wide and has a maximum depth of 25 m. Between Lake DM and Lake A, a thin layer (few centimeters to roughly 2 meters) of alluvial sands are deposited on top the clayey sediments (Figure S1) (Ageos, 2010).

Lake A is connected to a small stream (S1) with a mean and maximum annual discharge of 0.32 m$^3$ s$^{-1}$ and 1.19 m$^3$ s$^{-1}$, respectively (Ageos, 2010). Maximum discharge typically occurs during the month of April as S1 drains snowmelt water from a small watershed (14.4 km$^2$) (Centre d'Expertise Hydrique du Québec, 2019), whereas low flow is recorded for the rest of the hydrological year. For the springtime 2017, the surface water flow from S1 are deemed negligible compared to the flood-water inputs and are thus not considered in this study.

Two channelized outlet streams (S2 and S3) allow water to exit Lake A and flow towards Lake DM. The direction of the surface water fluxes at S2 can be reversed if water level at Lake DM exceeds both a topographic threshold at 22.12 m.a.s.l. (determined from a topographic land survey along S2) and the water level at Lake A (Ageos, 2010). Flow reversal also occurs in S3, but the elevation of the topographic threshold is unknown.

Lake A and Lake B both contribute to the supply of a bank filtration system which is composed of eight wells and is designed to supply drinking water for up to 18000 people (Ageos, 2010). Typically, two to three wells are operated on a daily basis at a total pumping rate ranging from 4000 m$^3$/d (in wintertime) to 7500 m$^3$/d (in summertime) (Masse-Dufresne et al., 2019). Although the operation of the bank filtration system does not form a complete hydraulic barrier between the two artificial lakes, it does lead to a lowering of Lake B water level below that of Lake A (Ageos, 2010).

### 2.2 Hydrodynamics of the major flood event

In 2017, a major flood event occurred in the peri-urban region of Montreal and was caused by the combination of intense precipitations and snowpack melting over the Ottawa River watershed (Teufel et al., 2019). Rapid water level rise at Lake DM occurred in late February, early April and early May at rates of approximately 0.11 m d$^{-1}$, 0.19 m d$^{-1}$ and 0.16 m d$^{-1}$, respectively. A historical maximum water level (i.e., 24.77 m.a.s.l.) was reached on May 8, 2017, corresponding to a net water level rise of >2.7 m compared to early February (Fig. 2). High water levels at Lake DM resulted in the inundation of the area between Lake A and Lake DM (Fig. 1d), and the surface water fluxes were not constrained in S2 and S3 but occurred over a 1 km wide area.

The water level in Lake A was equivalent to Lake DM during the flood peak (on May 8, 2017) and daily mean water levels at Lake A and Lake DM show good correlation ($R^2 = 0.98$, p-value < 0.01) for the observed period (April 27th, 2017 to May 17th, 2017). Daily mean water levels at observation well VP and Lake DM also follow a similar pattern from late February 2017 to late July 2017 ($R^2 = 0.93$, p-value < 0.01). Considering the above and a visible hydraulic connection between the

Lake DM and Lake A, the data indicates that the daily water level variations at observation well VP were controlled by Lake DM from late February to late July 2017. Lake A water level was also presumably controlled by Lake DM until late July 2017, but technical issues prevented confirmation (i.e., logger in Lake A broke on May 17th, 2017).

Then, from August 2017 to late October 2017, the water level in Lake DM was below the topographical threshold, and there is no similarity between the evolution of the water level at Lake DM and observation well VP ($R^2 = 0.11$, p-value > 0.01). It

is thus possible to infer that the Lake A water level was not controlled by Lake DM from August 2017 to late October 2017. This is also supported by the manual measurement of Lake A water level in September.

The water level of Lake DM exceeded the topographic threshold again from November 2017 to January 2018, but the daily mean water levels at Lake DM and observation well VP show a moderate correlation ($R^2 = 0.63$, p-value < 0.01). The manual measurements also indicate discrepancy between Lake DM and Lake A water levels in December 2017 and January 2018. The

weaker correlation between the water levels measurements suggest that Lake DM was not controlling the dynamics of Lake A water level. It is thus likely that Lake A received little to no surface water inputs from Lake DM from November 2017 to January 2018. In this context, surface water inflow from Lake DM during autumn and winter are considered negligible in this study and not included in the developed stable isotope mass balance model (Sect. 4.2).

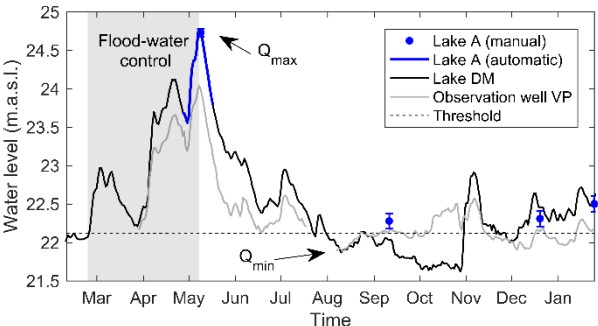

**Figure 2. Daily mean water levels at Lake A, Lake DM and observation well VP from February 9th, 2017 to January 25th, 2018. The grey shaded area corresponds to the flood-water control period. $Q_{max}$ and $Q_{min}$ indicate the timing of the adjusted maximum and minimum output from the lake.**

### 2.3 Conceptual model of Lake A water balance

Based on the geological and hydrological setting of the study site (Sect. 2.1) and flood-specific considerations (Sect. 2.2), we

established a conceptual model of Lake A water balance, as described below.

Considering that Lake A is sitting in alluvial sands (i.e., a highly permeable material), it is assumed that groundwater inputs ($I_G$) and outputs ($Q_G$) contribute to the water budget. Although it is difficult to interpret the location of $I_G$, it appears evident that $Q_G$ occur along the NE bank of Lake A. In fact, there are subsurface fluxes across the sandy bank that contribute to the bank filtration system or discharge into Lake B, as its water level is lower since the initiation of the bank filtration system

(Masse-Dufresne et al., 2019). Given the regional groundwater flow in the NE to SW , $Q_G$ can also presumably occur along

the SW bank of Lake A. Besides, it is likely that little to no subsurface fluxes exists in the area between Lake A and Lake DM, where clayey sediments are found.

For the study period, it is conceptualized that the direction of the surface water fluxes in S2 and S3 is from Lake A to Lake DM, except from February 27th, 2017, 2017 to May 8th, 2017. During this period (hereafter referred to as the flood-water control period), the water level of Lake DM exceeds the topographic threshold, and Lake A would receive surface water inflow ($I_S$) from Lake DM. Also, it is likely that high water level in Lake A imposed a hydraulic gradient at the lake-aquifer interface, which allowed for $Q_G$ from the lake and inhibited $I_G$. Then, as Lake A and Lake DM water levels started to decrease (from May 8th, 2017), it is assumed that water exits Lake A as surface water outputs ($Q_S$) towards Lake DM or as $Q_G$. Although Lake DM water level again exceeded the topographic threshold from November 2017 to January 2018, the weaker correlation between the water levels suggest that Lake A water level was not controlled by Lake DM. In this context, we conceptualized that Lake A water level variations are mainly controlled by groundwater flows ($I_G$ and $Q_G$). Surface water inputs ($I_S$) are set to zero during this period (see Sect. 2.2).

To summarize, for the year 2017, Lake A water budget can be conceptualized with two distinct hydrological periods: (a) the groundwater control period and (b) the flood-water control period (Fig. 3). While the groundwater control period concerns most of the hydrological year, the flood-water control period only applies from February 23rd, 2017 to May 8th, 2017. During the groundwater control period (Fig.3a), it is assumed that groundwater inflows ($I_G$) and precipitations (P) constitute the total water inputs to Lake A, while surface water inflows ($I_S$) are negligible. During this period, the outputs are occurring through evaporative fluxes (E), surface water outflows ($Q_S$) and groundwater outflows ($Q_G$). In contrast, it is assumed that $I_S$ and P represent the total water inputs to Lake A during the flood-water control period (Fig. 3b). High-water levels at Lake A impose a hydraulic gradient at the lake-aquifer interface which allow for $Q_G$ and inhibits $I_G$.

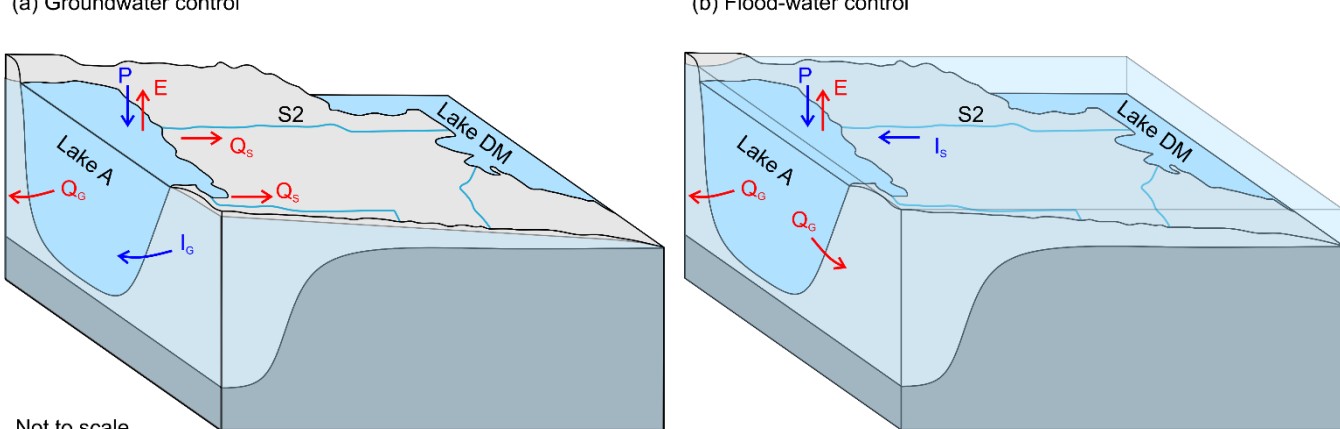

**Figure 3. Schematic representation of the hydrological processes at Lake A during (a) groundwater control, and (b) flood-water control periods. Inputs include precipitation (P), surface water ($I_S$) and groundwater ($I_G$) while outputs include evaporation (E), surface water outflow ($Q_S$) and groundwater outflow ($Q_G$). The area between Lake DM and Lake A is flooded in (b) and $I_S$ from Lake DM contribute to the water balance of Lake A.**

## 3 Methods

### 3.1 Field measurements

Pressure-temperature loggers (Divers®; TD-Diver and CTD-Diver, Van Essen Instruments, Delft, Netherland) were used to measure surface water levels at Lake A and groundwater levels at observation well VP on a 15-minute time step. Water levels were recorded from April 27th, 2017 (after the ice-cover melted) to May 17th, 2017 at Lake A and from March 29th, 2017 to January 25th, 2018 (except between July 19th, 2017 and August 6th, 2017) at observation well VP. All the level loggers' clocks were synchronized with the computer's clock when launching automatic measurements. This procedure was done via the Diver-Office 2018.2 software. Manual measurements of the water level were regularly performed to calibrate (relatively to a reference datum) and validate the automatic water level measurements. A level logger was also used to measure on-site atmospheric pressure and perform barometric compensation on water level measurements. Also, note that water levels in Lake A were not continuously recorded after May 17th, 2017 due to a logger failure, but manual water level measurements (in September 2017, December 2017 and January 2018) depict the general evolution of Lake A water level.

Mean daily water levels at Lake DM were retrieved with permission from the Centre d'Expertise Hydrique du Quebec database (Centre d'Expertise Hydrique du Québec, 2020). Meteorological data was measured at land-based meteorological stations near the study site and obtained from Environment and Climate Change Canada database (available online at weatherstats.ca). Daily air temperature, relative humidity, wind speed, dew point and atmospheric pressure were measured at Mirabel International Airport station (45.68 °N, -74.04 °E; 18 km from the study site). Daily precipitation and solar radiation were measured at Sainte-Anne-de-Bellevue station (45.43 °N, -73.93 °E; 10 km from the study site) and Montreal International Airport station (45.47 °N, -73.75 °E; 17 km from the study site), respectively.

### 3.2 Water sampling and analytical techniques

Physico-chemical parameters measurements and water sampling were performed at Lake A at approximately 0.3 m below the surface and 1 m from the lake shoreline (at LA-S1) on a weekly to monthly basis from February 9th, 2017 to January 25th, 2018. Physico-chemical parameters (including temperature, electrical conductivity, pH and redox potential) were measured using a multiparameter probe (YSI Pro Plus 6051030 and Pro Series pH/ORP/ISE and Conductivity Field Cable 6051030-1, YSI Incorporated, Yellow Springs, OH, USA). Additionally, vertical profile measurements and depth-resolved water sampling were conducted on February 9th, 2017, August 17th, 2017 and January 25th, 2018 (at LA-P1 to LA-P4). Lake A water sampling was performed in the northern part of the lake for logistical reasons and due to ease of accessibility. As horizontal homogeneity has been previously demonstrated by Pazouki et al. (2016), the water samples were deemed representative of the whole waterbody.

Flood water was sampled at two locations (near S2 and S3) on April 19th, 2017 and at Lake DM on May 10th, 2017. Water samples were also collected at the surface and at depth within Lake B and at observation well Z16, which is upstream of Lake B and, thus, representative of the regional groundwater contributing to the latter (Ageos, 2016).

Water samples were analyzed for major ions, alkalinity and stable isotopic compositions of water ($\delta^{18}O$ and $\delta^2H$). Water was filtered in the field using 0.45 μm hydrophilic polyvinylidene fluoride (PVDF) membranes (Millex-HV, Millipore, Burlington, MA, USA) prior to sampling for major ions and alkalinity. From December to March, cold weather prevented field filtration, so this procedure was performed in the laboratory on the same day. All samples were collected in 50-ml polypropylene containers and cooled during transport to the laboratory. The samples were then kept refrigerated at 4 °C until analysis, except for stable isotopes, which were stored at room temperature. Major ions were analyzed within 48 h via ionic chromatography (ICS 5000 AS-DP Dionex Thermo Fisher Scientific, Saint-Laurent, QC, Canada) at Polytechnique Montreal (Montreal, Quebec). The limit of detection was ≤0.2 mg/L for all major ions. Bicarbonate concentrations were derived from alkalinity, which was measured manually in the laboratory according to the Gran method (Gran, 1952) at Polytechnique Montreal (Montreal, Quebec). On samples with measured alkalinity ($n = 12$), the ionic balance errors were all below 8%. The mean and median ionic balance errors were 1%. Stable isotopes of oxygen and hydrogen were measured with a Water Isotope Analyser with off-axis integrated cavity output spectroscopy (LGR-T-LWIA-45-EP, Los Gatos Research, San Jose, CA, USA) at Geotop-UQAM (Montreal, Quebec). 1 ml of water was pipetted in a 2 ml vial and closed with a septum cap. Each sample was injected (1 microliter) and measured 10 times. The first two injections of each sample were rejected to limit memory effects. Three internal reference waters ($\delta^{18}O$ = 0.23±0.06‰, -13.74±0.07‰ & -20.35±0.10‰; $\delta^2H$ = 1.28±0.27‰, -98.89±1.12‰ & -155.66±0.69) were used to normalize the results on the VSMOW-SLAP scale. A 4th reference water ($\delta^{18}O$ = -4.31±0.08‰; $\delta^2H$ = -25.19±0.83) was analyzed as an unknown to assess the exactness of the normalization. The overall analytical uncertainty (1 σ) is better than ±0.1‰ for $\delta^{18}O$ and ±1.0 ‰ for $\delta^2H$. This uncertainty is based on the long-term measurement of the 4th reference water and does not include the homogeneity nor the representativity of the sample.

### 3.3 Stable isotope mass balance

Stable isotope mass balances for lakes can either be performed based on (i) a well-mixed single layer model or (ii) a depth resolved multi-layered model. Arnoux et al. (2017c) performed a comparison of both methods and reported that well-mixed and depth resolved multi-layered models yielded similar results and showed that groundwater inputs and outputs play an important role on lake water budgets. Arnoux et al. (2017c) further highlighted that the multi-layer model additionally allowed for the determination of groundwater flow with depth, but required a temporally- and depth-resolved sampling in order to ensure a thorough understanding of the stability/mixing of the different layers. Such time-consuming sampling and monitoring efforts are however often unrealistic in remote and/or flood-affected contexts. Additionally, Gibson et al. (2017) showed that the timing of the lake water sampling may introduce greater bias in a well-mixed isotopic mass balance model than the uncertainty related to the lake stratification. For these reasons, we opted to develop a well-mixed model in the context of this study. Note that, despite the biases underlying well-mixed models, this approach remains adequate to characterize the relative importance of hydrological processes and is particularly useful to give first-order estimate of water fluxes in ungauged basins. The water and stable isotope mass balance of a well-mixed lake can be described, respectively as Eq. (1) and Eq. (2):

$$\frac{dV}{dt} = I - E - Q \tag{1}$$

$$V\frac{d\delta_L}{dt} + \delta_L\frac{dV}{dt} = I\delta_I - E\delta_E - Q\delta_Q \tag{2}$$

where V is the lake volume, t is time, I is the instantaneous inflow, E is evaporation, Q is the instantaneous outflow. I corresponds to the sum of surface water inflow ($I_S$), groundwater inflow ($I_G$) and precipitations (P). Similarly, Q is the sum of surface water outflow ($Q_S$) and groundwater outflow ($Q_G$). $\delta_L$, $\delta_I$, $\delta_E$ and $\delta_Q$ are the isotopic compositions of the lake, I, E and Q, respectively. In the context of this study, the balance equations can be simplified based on the conceptual model. During the groundwater control period, $I_S = 0$ and, thus, $I = I_G + P$ and $\delta_I = (\delta_G I_G + \delta_P I_P)/I$. In contrast, $I_G = 0$ during the flood-water control period, $I = I_S + P$ and $\delta_I = (\delta_{Is} I_S + \delta_P I_P)/I$. Note that $\delta_G$ and $\delta_{Is}$ are the isotopic signatures of groundwater and surface water inputs, respectively.

The application of Eq. (1) and Eq. (2) for both $\delta^{18}O$ and $\delta^2H$ is valid during the ice-free period and also assumes constant density of water (Gibson, 2002). In this study, the potential impacts of the ice-cover formation and melting are neglected, as the ice volume is likely to represent only a small fraction (<2%) of the entire water body. Moreover, considering the ice-water isotopic separation factor, i.e., 3.1 ‰ for $\delta^{18}O$ and 19.3 ‰ for $\delta^2H$ (O'Neil, 1968) and assuming well-mixed conditions, the lake water isotopic variation would be comprised within the analytical uncertainty. Also, flood-water inputs from Lake DM were expected to be much more important and occurring simultaneously with ice-melt during the freshet period.

Thus, a volume-dependent model is applied, as described in Gibson (2002). The change in the isotopic composition of the lake ($\delta_L$) with f (i.e., the remaining fraction of lake water) can be expressed as Eq. (3):

$$\delta_L(f) = \delta_S - (\delta_S - \delta_0)f^{\left[\frac{-(1+mX)}{1-X-Y}\right]} \tag{3}$$

where X = E/I is the fraction of lake water lost by evaporation, Y=Q/I is the fraction of lake water lost to liquid outflows, m is the temporal enrichment slope (see Appendix B), $\delta_0$ is the isotopic composition of the lake at the beginning of the time-step, and $\delta_S$ is the steady-state isotopic composition the lake would attain if f tends to 0 (see Appendix B).

A step-wise approach is used to solve Eq. 3 on a daily time-step. At each time step, recalculation of $f=V/V_0$ is needed, where V is the residual volume at the end of the time step and $V_0$ the original volume at the beginning of the time step (or $V^{t-dt}$). Hence, Eq. (3) is based on the water level difference between two days. The water flux parameters (E, I and Q) and isotopic signatures ($\delta_E$, $\delta_A$, $\delta_I$ and $\delta_Q$) are thus evaluated on a daily time-step.

The flushing time ($t_f$) is defined as the ratio of the volume of water in a system to the rate of renewal (Monsen et al., 2002). In this study, $t_f$ by groundwater inputs is considered and be expressed as :

$$t_f = V/I_G \tag{4}$$

## 3.4 Daily volume changes at Lake A and water fluxes

The initial lake volume ($4.7 \times 10^6$ m$^3$) was estimated from the observed lake surface area ($2.79 \times 10^5$ m$^2$) and the maximal

depth (20 m) and assuming bank slopes of 25 degrees. Assuming bank slopes of 20 degrees or 30 degrees, a typical range for

saturated sands (Holtz and Kovacs, 1981), would result in an estimated initial lake volume of $4.84 \times 10^6$ m$^3$ (+3%) and

$4.32 \times 10^6$ m$^3$ (-8%). Lake A volume variations are estimated from daily water level changes and assuming a constant lake

area. As water level measurement are only available for a short period at Lake A, water levels at Lake DM and observation

well VP are used as proxies. Water levels at observation well VP were used as a proxy from August 24th, 2017 to October 30th,

2017, while water levels at Lake DM were assumed representative of Lake A for the rest of the study period (i.e., from February

9th, 2017 to August 23rd, 2017 and from October 31st, 2017 to January 25th, 2018). This approximation is deemed acceptable

because the simulation of $\delta_L$ depends on the remaining fraction of lake water f (not the absolute water level), and daily

variations of the water levels at Lake A, Lake DM and observation well VP were shown to be similar (see Sect. 2.2).

Evaporative fluxes (E) are calculated using the standardized Penman-48 evaporation equation, as described in Valiantzas

(2006):

$$E_{Penman-48} = \frac{\Delta}{\Delta+\gamma} \cdot \frac{R_n}{\lambda} + \frac{\gamma}{\Delta+\gamma} \cdot \frac{6.43 f(u) D}{\lambda} \qquad (5)$$

where $R_n$ is the net solar radiation (MJ m$^{-2}$ d$^{-1}$), $\Delta$ is the slope of the saturation vapor pressure curve (kPa °C$^{-1}$), $\gamma$ is the

psychrometric coefficient (kPa °C$^{-1}$), $\lambda$ is the latent heat of vaporization (MJ kg$^{-1}$), $f(u)$ is the wind function (see Appendix B)

and D is the vapor pressure deficit. For comparative purposes, estimation of the daily evaporative fluxes was also conducted

with the Linacre-OW equation (Linacre, 1977) and the simplified - Penman-48 equation (Valiantzas, 2006).These methods

yielded similar evaporation estimates from April to August but underestimated total evaporation by 24% to 33% compared to

the standardized Penman-48 equation. The discrepancy between the models is restricted to late summer and autumn (see

Appendix C, Fig. C1) and is attributed to the difference between the air and water surface temperature, which was estimated

based on the equilibrium method as described by de Bruin (1982) (see Appendix D). Note that E and P are set to zero during

the ice-cover period (i.e. from January 1st to March 31st, based on meteorological data and field observations).

For well-mixed conditions, the $\delta_{Qs}$ and $\delta_{Qg}$ are assumed to be equal to $\delta_L$. Hence, no separation of these two fluxes is attempted

and they are merged into one variable, i.e., the outflow (Q). The direction and intensity of the water flux at the lake-aquifer

interface can be conceptually described by Darcy's Law which states that Q = KAi, where K is the hydraulic conductivity, A

is the cross-sectional area through which the water flows, and i is the hydraulic gradient. Given the significant depth of Lake A

(i.e., 20 m) in comparison to the maximum water level change during the flooding event (i.e., 2.7 m), the variation of the A

and K are expected to have minor impact on Q. Hence, the change in outflows from the lake is expected to be mainly controlled

by i changes and, consequently, to be roughly proportional to the change in lake water level,. Considering the above, it was

assumed that the daily outflow flux from Lake A varied linearly according to the lake water level; the minimum and maximum

outflow ($Q_{min}$ and $Q_{max}$) correspond to the minimum and maximum water level, respectively. The outflow range (i.e., minimum

and maximum values) was adjusted to obtain best fit between the calculated and observed $\delta_L$.

Total daily inflow (sum of daily P, $I_S$ and $I_G$) into Lake A compensates for the adjusted daily outflow and daily lake volume difference. The precipitations (P) are evaluated from the available meteorological data (see Sect. 3.1), while direct measurement of $I_S$ and $I_G$ was not possible in this hydrogeological context (see Sect. 2.1). Consequently, further assumptions are needed to apportion these contributions. Considering the proposed conceptual model of the groundwater-surface water interactions (see Sect. 2.2), $I_S$ is set to zero, while $I_G$ is contributing to the lake during groundwater control period. On the other hand, during the flood-water control period (i.e., from February 23$^{rd}$, 2017 to May 8$^{th}$, 2017), it is assumed that the rising water level at Lake A results in a hydraulic gradient forcing the lake water to infiltrate into the aquifer, inhibiting $I_G$.

## 4 Results

From February 23$^{rd}$, 2017 to May 8$^{th}$, 2017, the net water fluxes are mainly positive, and an overall volume increase is observed at Lake A. The maximum volume change of Lake A was 7.6 x 10$^5$ m$^3$, which represents 16 % of the lake's initial volume. The maximum net water flux was 1.2 x 10$^5$ m$^3$ d$^{-1}$, corresponding to a water level rise of 0.43 m (on April 5$^{th}$, 2017 only). From May 9$^{th}$, 2017 to mid-August 2017, Lake A volume was decreasing, and the daily net water fluxes were mainly negative. In early August 2017, Lake A regained its initial volume. Then, in autumn and winter, the volume of Lake A was oscillating, and the net water fluxes were ranging from -6.4 x 10$^4$ m$^3$ d$^{-1}$ to 5.3 x 10$^4$ m$^3$ d$^{-1}$. At the end of the study period (i.e., on January 25$^{th}$, 2018), a net volume difference of 1.5 x 10$^5$ m$^3$ remained at Lake A compared to February 9$^{th}$, 2017.

However, the evolution of Lake A volume and the net water fluxes are not representative of the surface water/groundwater interactions. Indeed, gross water fluxes are likely to exceed net water fluxes at natural and dredged lakes sitting in permeable sediments (Zimmermann, 1979; Arnoux et al., 2017a; Jones et al., 2016). In the context of this study, we conceptualized two main hydrological periods, during which the lake water can either drain towards Lake DM or exit the lake as groundwater output. To balance out these outputs, the inflows to Lake A must therefore be greater than the net water fluxes.

For that reason, the development of a volume-dependent transient stable isotope mass balance was required to correctly depict the importance of the flood-water inputs on the water mass balance of the lake.

### 4.1 Isotopic and geochemical framework

The isotopic composition of precipitation ($\delta_P$), Lake A and flood-water are depicted in Fig. 4. The Local Meteoric Water Line (LMWL) was defined using an ordinary least squares regression (Hughes and Crawford, 2012) using isotope data in precipitation from Saint-Bruno station IRRES database ($n = 27$; from December 2015 to June 2017).

For the study period, the isotopic composition of bulk precipitation was available on a biweekly to monthly time-step ($n = 15$) and ranged from -19.19‰ to -6.85‰ for $\delta^{18}O$ and -144‰ to -38‰ for $\delta^2H$. Interpolation was used to simulate the $\delta_P$ on a daily-time step for the isotope mass balance model computation.

Isotopic compositions of Lake A water samples ($n = 39$) are linearly correlated (see solid blue line) and all plot below the Local Meteoric Water Line (LMWL), which confirms that Lake A is influenced by evaporation. Linear regression of Lake A water

samples defines the Local Evaporation Line (LEL), which is $\delta^2H = 5.68$ ($\pm 0.27$) * $\delta^{18}O$ - 12.80 ($\pm 2.83$) ($R^2 = 0.92$). Some samples from the surface of Lake A plot below the LEL, likely indicating snowmelt water inputs as noted in previous studies

of Canadian lakes (Wolfe et al., 2007).

The isotopic composition of the flood-water samples ($n = 3$) is indeed more depleted than Lake A waters (i.e. $\delta^{18}O$ from -11.85 ‰ to -11.18 ‰ and $\delta^2H$ from -81 ‰ to -78 ‰) and is most likely to reflect the significant contribution from heavy isotope depleted snowmelt waters. The flood-water samples are also linearly correlated and plot along a line ($\delta^2H = 5.33 \ \delta^{18}O$-18.82) which slope is similar to Lake A LEL, suggesting that the sampled flood-water evaporated under the

same conditions as Lake A water samples. For simplification purposes, the isotopic composition of the surface water inflow ($\delta_{Is}$) was set to the intersection between the flood-water LEL and the LMWL ($\delta^{18}O$ = -12.00 ‰ and $\delta^2H$ = -83 ‰). The long-term (1997-2008) average, minimum and maximum isotopic signature of Ottawa River water at Carillon (~34 km upstream from Lake DM; see Fig.1b for the month of April are -11.19 ‰, -12.01 ‰ and -10.23 ‰ for $\delta^{18}O$ and -81 ‰, -85 ‰ and -77 ‰ for $\delta^2H$, respectively (Rosa et al., 2016). The mean and minimum values compare well with the observed isotopic signatures

at Lake DM during springtime 2017.

The isotopic composition of groundwater ($\delta_G$) can be determined from direct groundwater samples or indirectly from the amount-weighted mean $\delta_P$. However, in highly seasonal climates, there is a widespread cold season bias to groundwater recharge (Jasechko et al., 2017), and estimating $\delta_G$ via groundwater samples or amount-weighted mean $\delta_P$ may be misleading. In fact, it has been argued that the LMWL-LEL intersection better represents the isotopic composition of the inflowing water

to a lake and is thus commonly used to depict the $\delta_G$ in isotopic mass balance applications (Gibson et al., 1993; Wolfe et al., 2007; Edwards et al., 2004). Concerning the study site, the estimated $\delta_G$ is -11.26 ‰ for $\delta^{18}O$ and -77 ‰ for $\delta^2H$ (i.e., the Saint-Bruno LMWL and Lake A LEL intersection). The latter compares well with the mean isotopic signature of groundwater at Saint-Télesphore station (-11.1‰ for $\delta^{18}O$ and -78.5‰ for $\delta^2H$) (Larocque et al., 2015) and is more depleted than the long-term amount-weighted mean $\delta_P$ at Ottawa (-10.9‰ for $\delta^{18}O$ and -75‰ for $\delta^2H$) (IAEA/WMO, 2018). Note that the location of

Saint-Télesphore station and Ottawa are depicted in Fig.1b.

The geochemical facies of Lake A and Lake DM samples are illustrated in Fig. 5 by the means of a Piper diagram. Mean values for Lake B and regional groundwater (GW) geochemical facies are also plotted for comparison purposes. Both Lake A and flood-water were found to be Ca-HCO$_3$ types, which is typical for precipitation- and snowmelt-dominated waters (Clark, 2015). The geochemistry of Lake A is relatively constant throughout the year and reveals a depth-wise homogeneity. The

geochemistry of Lake B is distinct from Lake A and appears to be influenced by regional groundwater characterized by a Na-Cl water type.

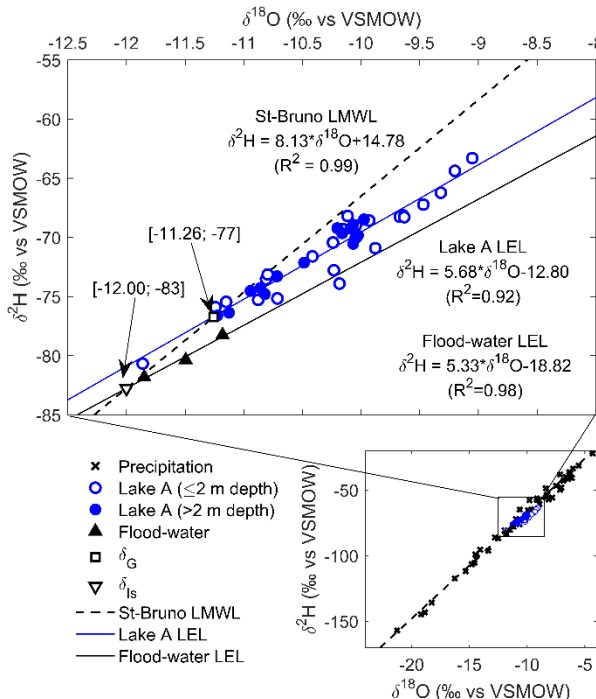

**Figure 4. Isotopic composition of precipitation, Lake A water, and flood-water from March 2017 to January 2018. Hollow and solid blue circles correspond to samples collected at ≤2 m and >2 m depth, respectively. Analytical precision is 0.15‰ and 1‰ at 1σ for δ¹⁸O and δ²H. Precipitation data are retrieved from the research infrastructure on groundwater recharge database (Barbecot et al., 2019).**

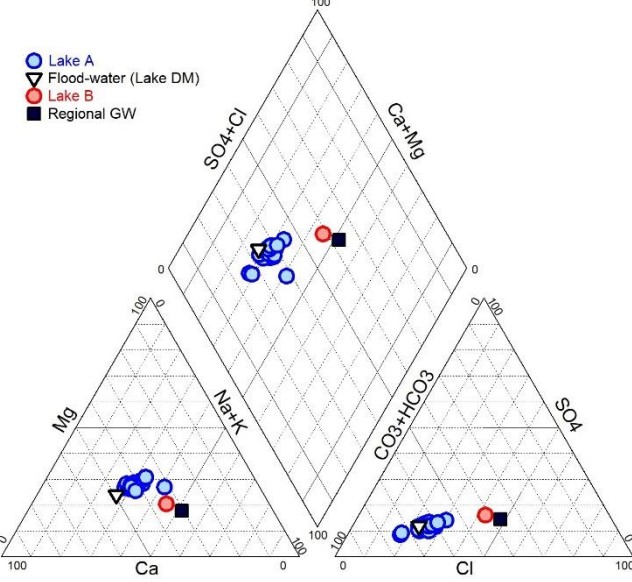

Figure 5. Geochemical facies of Lake A ($n = 23$) and flood-water ($n = 1$). Mean values for Lake B ($n = 42$) and regional groundwater (GW) ($n = 11$) geochemical facies are also plotted. Lake A and flood-water are characterized by Ca-HCO₃ water types, while Lake

B and regional GW correspond to Na-Cl water types. Note that regional GW was sampled upstream of Lake B.

## 4.2 Evaluation of the water budget

### 4.2.1 Volume dependent isotopic mass balance model

As described in Sect. 3.3, the isotopic mass balance model was solved iteratively by recalculating $\delta_L$ on a daily time-step. This model was developed assuming (1) well-mixed conditions and (2) that the outflow fluxes changes are roughly proportional to

the lake's water level changes. We adjusted minimum and maximum outflow fluxes ($Q_{min}$ and $Q_{max}$) so that they correspond to the minimum and maximum water levels (see Fig. 3).

Three sampling campaigns (i.e., on February 9th, 2017, August 17th, 2017 and January 25th, 2018) were conducted at Lake A in order to collect water samples for isotopic analyses from the epilimnion, metalimnion and hypolimnion (Fig. 6; Appendix E, Fig. E1) to account for the vertical stratification of the isotopic signature (Gibson et al., 2017). The vertical isotopic profiles

were volume-weighted according to the representative layer for each discrete measurement in order to obtain the observed $\delta_L$ for each campaign (Table 1). The depth-averaged isotopic composition of the lake on February 9th, 2017 (i.e., $\delta^{18}O$ = -10.15 ‰ and $\delta^2H$ = -70 ‰) was used as the initial modelled $\delta_L$.

While depth-averaged $\delta_L$ was not available during the flood-water control period (i.e., late February to early May), water samples from the surface of Lake A provide relevant evidence to better constrain the model. It is likely that Lake A was fully

mixed during the flood-water control period, and that the water samples collected at the surface of Lake A on April 27th, 2017 or May 9-10th, 2017 are representative of the whole water body. Indeed, the observed surface water temperature was < 5°C until early May (see Fig. C1) and suggests a limited density gradient along the water column which does not allow for the development of thermal stratification. In this context, we opted to simulate two scenarios (A and B), for which the isotopic mass balance model is either constrained at $\delta^{18}O$ = -11.20 ‰ and $\delta^2H$ = -76 ‰ on May 9-10, 2017 or at $\delta^{18}O$ = -11.86 ‰ and

$\delta^2H$ = -80.68 ‰ on April 27th, 2017.

Table 1. Observed depth-averaged (or mean) and standard deviation (std) of isotopic composition of Lake A for the sampling campaigns in February 2017, August 2017 and January 2018 and all samples. The isotopic composition of the samples collected at the surface of Lake A on May 9-10, 2017 and April 27th, 2017 are also listed. The asterisks (*) indicate that a mean value was calculated (instead of a depth-averaged value).

| Period | Date | $n$ | $\delta^{18}O$ (‰) | | $\delta^2H$ (‰) | |
| --- | --- | --- | --- | --- | --- | --- |
| | | | depth-averaged | std | depth-averaged | std |
| Groundwater control | Feb 9th, 2017 | 9 | -10.15 | 0.11 | -69.92 | 0.41 |
| Flood-water control | May 9-10, 2017 (Scenario A) | 2 | -11.20 | 0.05 | -75.68 | 0.23 |
| | April 27th, 2017 (Scenario B) | 1 | -11.86 | - | -80.68 | - |
| Groundwater control | Aug 17th, 2017 | 7 | -10.61 | 0.82 | -73.33 | 4.41 |
| Groundwater control | Jan 25th, 2018 | 6 | -10.70 | 0.26 | -73.70 | 1.22 |

| | All samples | 34 | -10.32* | 0.62 | -71.35* | 3.69 |
| --- | --- | --- | --- | --- | --- | --- |

The results of the volume-dependent isotopic mass balance for $\delta^{18}O$ and $\delta^2H$ are illustrated in Fig. 6. The fitted $Q_{min}$ and $Q_{max}$ from Lake A are 3.7 x 10$^4$ m$^3$ d$^{-1}$ and 8.0 x 10$^4$ m$^3$ d$^{-1}$ for scenario A and 1.0 x 10$^3$ m$^3$ d$^{-1}$ and 2.8 x 10$^5$ m$^3$ d$^{-1}$ for scenario B. These first-order water fluxes estimates represent equivalent water level variations ranging from 0.004 m d$^{-1}$ and 1.0 m d$^{-1}$. From February 23$^{rd}$, 2017 to May 8$^{th}$, 2017 (see grey shaded area), hydraulic conditions allowed for surface inputs ($I_s$) from Lake DM to Lake A at a mean rate of 6.61 x 10$^4$ m$^3$ d$^{-1}$ with a total flood-water volume of 4.82 x 10$^6$ m$^3$ for scenario A. The total flood-water volume was twice as important (9.96 x 10$^6$ m$^3$) for scenario B. Then, from May 9$^{th}$, 2017, we considered that these flood-water inputs stopped, as the lake water level started to decrease. As a consequence, the model yielded a gradual enrichment of $\delta_L$ due to the combined contribution from $I_G$ and E for both scenarios. From May 9$^{th}$, 2017 to January 25$^{th}$, 2018, the total $I_G$ were 1.16 x 10$^7$ m$^3$ and 1.48 x 10$^7$ m$^3$ for scenario A and B respectively. Overall, the $\delta^{18}O$ and $\delta^2H$ models were better at reproducing the January 2018 and August 2017 observed $\delta_L$, respectively. This is likely linked to the uncertainties and representativeness of the meteorological data, which is controlling the isotopic fractionation due to evaporation.

While the computed flows for scenario A are within a plausible range for the combination of surface and groundwater outflow processes (i.e., minimum and maximum equivalent water level variations of 0.13 m d$^{-1}$ and 0.29 m d$^{-1}$), scenario B yielded less realistic results (i.e., minimum and maximum equivalent water level variations of 0.004 m d$^{-1}$ and 1.0 m d$^{-1}$). As mentioned above, scenario B was constrained at $\delta^{18}O$ = -11.86 ‰ and $\delta^2H$ = -80.68 ‰ in late April (Fig. 6), based on a surface water sample which was taken during a temporarily decreasing water level period (Fig. 3) and is thus likely less representative of the overall lake's dynamics compared to scenario A. This is demonstrating the limit of the approach and that it is important to correctly constrain the model during flood events in order to perform precise estimations of the water balance.

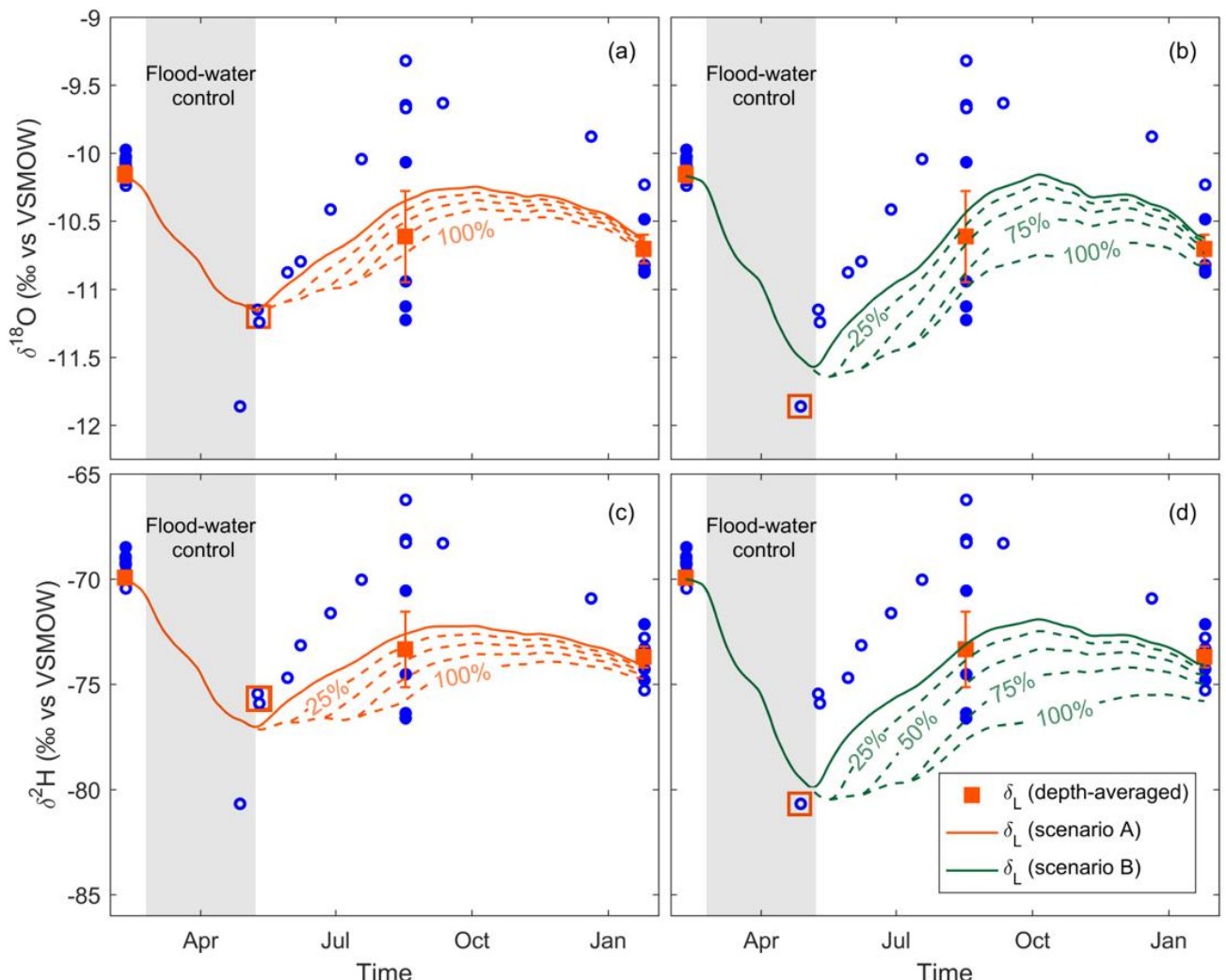

**Figure 6. Observed and modelled depth-average isotopic composition of the lake ($\delta_L$) for $\delta^{18}O$ (a-b) and $\delta^2H$ (c-d) from February 9th, 2017 to January 25th, 2018 for scenario A and B. The hollow and solid blue circles correspond to Lake A water samples collected at ≤2 m and >2 m, respectively. The modelled $\delta_L$ is fitted against the three depth-averaged $\delta_L$ and an additional sample collected at ≤2 m depth on May 9-10, 2017 (scenario A) and April 27th, 2017 (scenario B). These samples are marked by the hollow red squares. The grey shaded area corresponds to the flood-water control period. The error bars correspond to the standard error on the samples for each campaign. The dashed lines represent the modelled $\delta_L$ when considering that 25% to 100% of the outputs from the lake during the flood-water control period were temporally stored in the aquifer and discharged to the lake as flood-water-like inputs ($\delta_{Is}$).**

The water mass balance of Lake A from February 9th, 2017 to January 25th, 2018 is summarized in Table 2 for both scenarios. The difference between the total inputs and total outputs correspond to the lake volume difference (1.48 x 10^5 m^3) between the start and the end of the model run. Groundwater inputs ($I_G$) and surface water inputs ($I_S$) account for 71 % and 28 % of the total water inputs to the lake for scenario A. While $I_S$ are twice as important for scenario B, it only accounts for 39% (+11%) of the total inputs and the $I_G$ are 60% (-11%). It thus appears that the annual dynamic of Lake A is dominated by groundwater

inputs for both scenarios, despite the intensity of the flood event. For scenarios A and B, $t_f$, as defined in Eq. 4,4, is similar (i.e., 135 days and 110 days). Precipitations are contributing 1% of the total annual inputs and evaporation only accounts for 2% of the total annual outputs. Although the establishment of a hydraulic connection between Lake DM and Lake A is a

435 recurring yearly hydrological process, it is important to note that the magnitude and duration of the flooding event of 2017 was particularly important and, thus, had a greater impact on the dynamic of Lake A in comparison to other years.

**Table 2. Water mass balance of Lake A for scenario A and B. The difference between the total inputs and total outputs corresponds to the lake volume difference over the study period. The total inputs (I) correspond to the sum of precipitations (P), surface water inflow ($I_S$) and groundwater inflow ($I_G$). The total outputs (Q) correspond to the sum of evaporation (E) and surface water and**
440 **groundwater outflow (Q). The mean flushing time ($t_f$) is the ratio of the lake volume to the mean groundwater inputs ($I_G$).**

| Scenario | Inputs (x $10^6$ m$^3$) | | | Total I | Outputs (x $10^6$ m$^3$) | | Total Q | $t_f$ |
|---|---|---|---|---|---|---|---|---|
| | P | $I_S$ | $I_G$ | (x $10^6$ m$^3$) | E | Q | (x $10^6$ m$^3$) | (days) |
| A | 0.2 | 4.8 | 12.2 | 17.3 | 0.4 | 16.8 | 17.2 | 135 |
| B | 0.2 | 10.0 | 15.1 | 25.3 | 0.4 | 24.8 | 25.2 | 110 |
| Difference | 0.0 | 5.1 | 2.9 | 8.0 | 0.0 | 8.0 | 8.0 | -25 |
| | (0%) | (+107%) | (+24%) | (+46%) | (0%) | (+48%) | (+47%) | (-19%) |

### 4.2.2 Sensitivity analysis

A one-at-a-time (OAT) sensitivity analysis was performed to grasp the relative impact of the input parameters' uncertainties on the model outputs. For each parameter, we tested two scenarios which delimit the uncertainty for each parameter. First, we tested the sensitivity of the model for V + 3 % and V – 8 % (i.e., estimated with slopes of 30° and 20°). Concerning $\delta_{Is}$ and $\delta_G$,

the model was tested for ± 0.5 ‰ for $\delta^{18}$O and ± 4 ‰ for $\delta^2$H, assuming they would both evolve along the LMWL (see Fig. 3). Then, we assessed for the sensitivity of the model to $\delta_A$, by fixing the seasonality factor k at 0.5 and 0.9. Evaporation was computed with ± 20%, whereas the meteorological parameters (i.e., RH, $T_{air}$, U, P and Rs) were tested for ± 10%. As E and $\delta_A$ are dependent on the water surface temperature, we also tested the sensitivity of the model when considering that T is equal to the daily mean air temperature ($T_{air}$). Finally, we tested for the uncertainties concerning the definition of the LMWL. For the

reference scenario, the LMWL ($\delta^2$H = 8.13 * $\delta^{18}$O + 14.78) was estimated using an ordinary least square regression (OLSR). For the sensitivity analysis, we estimated the LMWL via a precipitation amount weighted least square regression (PWLSR), which was developed by Hughes and Crawford (2012). Using the PWLSR method, the LMWL is defined as $\delta^2$H = 8.28 * $\delta^{18}$O + 17.73, and $\delta_{Is}$ and $\delta_G$ are estimated at -12.39 ‰ and -11.74 ‰ for $\delta^{18}$O and at -85 ‰ and -79 ‰ for $\delta^2$H, respectively. Recalculation of $\delta_{Is}$ and $\delta_G$ was needed, as they were both assumed to plot on the LMWL (see Sect. 4.1).

The results of this sensitivity analysis are listed in Table F1 and Table F2 (Appendix F) for scenarios A and B. Overall, the model was found to be highly sensitive to the uncertainties associated with $\delta_{Is}$, $\delta_G$ and E, as the annual mean water fluxes (Q and I) varied up to -31% and +46% compared to the reference scenarios A and B. A negligible to slight change on the modelled $\delta_L$ was found when considering the uncertainties for V, $\delta_A$, RH, $T_{air}$, U, P and Rs. For these variables, the mean flux estimate (Q and I) changes ranged from -8% to +4% compared to the reference scenarios A and B. As expected, the value of $\delta_{Is}$ affects

the modelled $\delta_L$ exclusively during the flood-water control period. Similarly, the values of $\delta_G$ and E particularly influence the modelled $\delta_L$ from late summer to early winter. This is due to the fact that Q and E are the dominant fluxes during this period. When considering that T is equal to $T_{air}$, despite the significantly different maximum and minimum values for Q, the mean Q was relatively similar to the reference scenarios and only a small change for $t_f$ (+3% and +2% compared to reference scenarios A and B) was found. Finally, the model is highly sensitive to the uncertainties associated with the LMWL, as a translation of

the LMWL implies an enrichment or depletion of both the $\delta_{Is}$, $\delta_G$ at the same time. Such modifications result in mean flux estimate (Q and I) changes of up to -38% and -43% compared to reference scenarios A and B.

## 4.3 Importance of temporary flood-water storage on the water balance partition

The developed isotopic mass balance model yielded significant flood-water inputs during springtime to best-fit the observed $\delta_L$. A first-order estimate of the total flood-water volume summed to 4.82 x $10^6$ m$^3$ (for scenario A), which is nearly equal to

the lake's initial volume (i.e., 4.70 x $10^6$ m$^3$). Similar results were obtained by Falcone (2007) who studied the hydrological processes influencing the water balance of lakes in the Peace-Athabasca Delta, Alberta (Canada) using water isotope tracers. They reported that a springtime freshet (in 2003) did replenish the flooded lakes from 68% to >100% (88% in average).

As mentioned in Sect. 2.3, it was conceptualized that the high surface water elevation of Lake A during springtime resulted in hydraulic gradients that forced lake water to infiltrate into the aquifer and induce local recharge (see Fig. 3). An important

volume of flood-derived water could thus be stored during the increasing water level period and eventually discharged back to the lake as its water level decreased. Hence, the groundwater inputs to Lake A following the flooding event likely corresponded to flood-derived surface water originating from Lake DM. Considering that these fluxes are characterized by a flood-water-like isotopic signature ($\delta_{Is}$), rather than the isotopic signature of groundwater ($\delta_G$), the temporal evolution of the modelled $\delta_L$ would be modified. Such consideration is noteworthy to better depict the importance of flood-water inputs in the

water balance partition.

Assuming that from 25% to 100% of the outputs (Q) from the lake during the flood-water control period were temporally stored in the aquifer and did eventually discharge back to the lake, the modelled $\delta_L$ diverges more or less from the reference scenarios A and B (Fig. 6, see the dashed lines). It is noteworthy that a better fit between the modelled $\delta_L$ and depth-averaged $\delta_L$ is obtained when considering that 25% to 50% of the outputs (Q) from the lake during the flood-water control period

discharges back to the lake. In fact, it is likely that part of the potential stored flood-water could have effectively discharged back to the lake. For instance, part of the flood-water-like groundwater could have been abstracted by the pumping wells at the adjacent bank filtration site or discharged to Lake B. These results illustrate the importance of considering temporary subsurface flood-water storage when assessing water balances, especially as the magnitude and frequency of floods are likely to be more important in the future (Aissia et al., 2012).

## 4.4 Temporal variability in the water balance partition

The water balance presented in Table 2 provides an overview of the relative importance of the hydrological processes at Lake A for the study period (i.e., February 2017 to January 2018). As the surface water inputs (as flood-water) only occurred during springtime at Lake A, it is also important to decipher the temporal variability of the water fluxes. The dependence of a lake on groundwater can be quantified via the G-Index, which is the ratio of cumulative groundwater inputs to the cumulative total inputs (Isokangas et al., 2015). Fig. 7 shows the temporal evolution of the G-Index from February 9$^{th}$, 2017 to January 25$^{th}$, 2018 for scenario A and the associated scenarios (A1 to A22) considered in the sensitivity analysis. Note that the G-Index is calculated at a daily time-step, based on the cumulative water fluxes. It is used to understand the relative importance of groundwater inputs over the studied period and does not consider the initial state of the lake. In early February, the G-Index is 100 %, because no surface water inputs ($I_S$) or precipitation (P) had yet contributed to the water balance. During the flood-water control period (see grey shaded area), the G-Index rapidly decreased and reached 12 % on May 8$^{th}$, 2017 (for the reference scenario A). A gradual increase of the G-Index is then computed for the rest of the study period. On January 25$^{th}$, 2018, the G-Index is 71 % and is likely more representative of annual conditions. Despite the sensitivity of the model to the input parameters, all scenarios yielded similar results. The G-Index ranged from 62 % to 75 % on an annual timescale for the different scenarios.

The impact of the potential temporary subsurface storage is also depicted in Fig. 7 (see dashed lines). As highlighted in Sect. 4.3, part of the potentially stored flood-water in the aquifer could have discharged back to the lake as flood-water-like inputs ($\delta_{I_S}$) after the flooding event. Considering these fluxes as surface water inputs ($I_S$), rather than groundwater inputs ($I_G$), would alter the temporal evolution of the G-Index. Assuming that 25% to 100% of the outputs from the lake during the flood-water control period did eventually discharge back to the lake, the flood-water inputs would contribute to the lake water balance until early June to early August (Fig. 7). Lake A would thus be dependent on flood-derived water during a 1- to 3-month period after the flooding event. On an annual timescale, the temporary subsurface storage could lower the G-Index to a minimum value of 47%.

## 5. Discussion

### 5.1 Resilience of lakes to groundwater pollution

Resilience of a system has been defined as its capacity to cope with perturbations (i.e., internal and/or external changes) while maintaining its state (Cumming et al., 2005). In the case of a lake, perturbations can manifest as a change in the water quantity and quality contributing to the water balance. According to Arnoux et al. (2017b), the impact of a perturbation to a lake is not only dependent on the relative importance of water budget fluxes, but also on the residence time of water in the lake. Thus, they proposed an interpretation framework which relates the response time of a lake to changes in groundwater quantity and/or quality, thereby linking the G-Index with $t_{f,}$ the mean flushing time by groundwater fluxes(Fig. 8). They depict a general case,

applicable to any pollution, regardless of reactivity or fate of contaminants. Hence, care should be taken when interpreting the sensitivity to specific contaminants which are subject to attenuation processes, such as degradation and sorption.

In their study, Arnoux et al. (2017b) assessed the resilience of kettle lakes ($n = 20$), located in southern Quebec (Canada), in similar morpho-climatic contexts to Lake A. The surveyed lakes were found to be characterized by a wide range of conditions;

from resilient (i.e., G-Index <50% and $t_f$ >5 years) to highly sensitive to groundwater changes (i.e., G-Index >50% and $t_f$ <1 year). This is related to the variability of the hydrogeological contexts, resulting in variations in the importance of groundwater contributions and the range of mean flushing times of the lakes (see grey arrow in Fig. 8). The majority of the lakes (i.e., 50%) were found to be characterized by intermediate conditions (G-Index >50% and 5< $t_f$ <1 years) and, thus, were classified as being relatively resilient to both surface and groundwater changes.

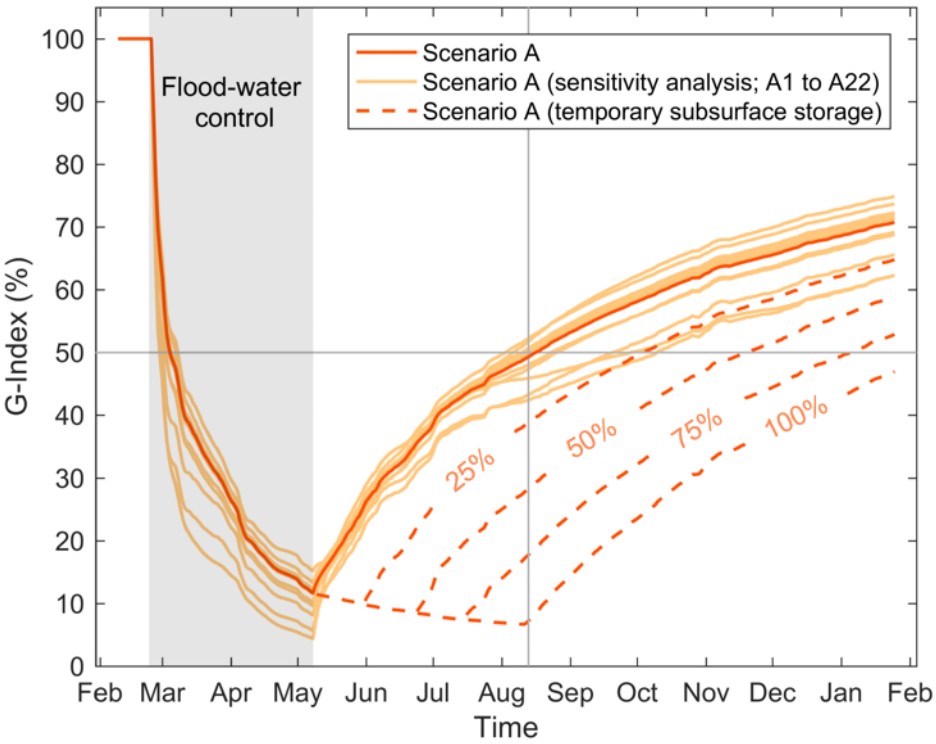

**Figure 7. Temporal evolution of the G-Index from February 9th, 2017 to January 25th, 2018 for scenario A and the associated scenarios considered in the sensitivity analysis (i.e., A1 to A22). The grey shaded area corresponds to the flood-water control period. The dashed lines correspond to the G-Index when considering that 25% to 100% of the outputs from the lake during the flood-water control period were temporally stored in the aquifer and discharged to the lake as flood-water-like inputs ($I_S$, $\delta_{Is}$).**

Concerning Lake A, studied scenarios (i.e., reference scenarios A and the sensitivity analysis) yielded values for G-Index >50% and $t_f$ <1 year, i.e., highly sensitive to groundwater changes, but resilient to surface pollution. Nevertheless, it was shown that temporary flood-water storage and discharge to lakes are crucial to correctly representing the G-Index by accounting for the origin of water fluxes (Fig.7; Sect. 4.4). While flood-water storage lowers the G-Index, the $t_f$ slightly increases (see orange

arrow in Fig. 8). Therefore, the studied lake receives a reduced groundwater contribution relative to the initial estimated

apportionment when not accounting for flood-water storage, but is still characterized by a rapid flushing time. This implies

that flood-affected lakes are more likely to be characterized by an intermediate condition, and thus are relatively resilient to

groundwater quantity and quality changes. The geochemical data (Sect. 4.2) is in accordance with this interpretation. Indeed,

a low-mineralization and Ca-HCO$_3$ water type at Lake A is consistent with the significant flood-water contributions (to the

lake and aquifer). In comparison, the neighboring lake (i.e., Lake B) does not undergo yearly recurrent flooding and was shown

to be more mineralized with a Na-Cl water type, likely originating from road-salt contamination of regional groundwater

(Pazouki et al., 2016). Biehler et al. (2020) similarly reported hydrological controls on the geochemistry of a shallow aquifer

in an hyporheic zone, where river stage influenced the mixing ratio between river water and the deeper aquifer.

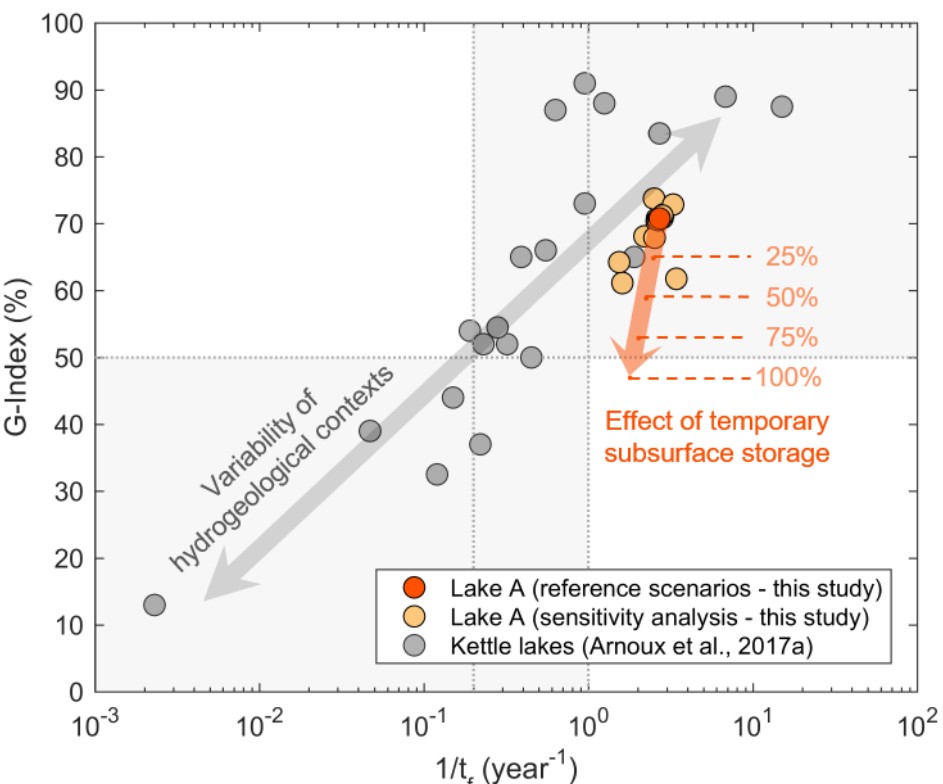

**Figure 8. Resilience of lakes to groundwater quantity and quality changes for Lake A (this study) and kettle lakes (Arnoux et al.,**
**2017b) in southern Quebec (Canada). G-Index is the ratio of groundwater inputs to total inputs and t$_f$ is the mean flushing time by**
**groundwater. This representation is adapted from Arnoux et al. (2017b).**

Considering the above, it is possible to speculate about the potential future impacts of climate change on Lake A. Globally,

future meteorological scenarios are predicting changes in precipitation and climate extremes, including floods and droughts

(Salinger, 2005). In Quebec (Canada), river stages are expected to increase across various watersheds in response to future

climate scenarios (Roy et al., 2001; Dibike and Coulibaly, 2005; Minville et al., 2008). These hydrological responses could

result in floods of longer duration and higher intensity (Aissia et al., 2012) and more pronounced droughts (Wheaton et al., 2007). Such changes could directly affect the quality of Lake A. If flooding becomes more prevalent, enhanced flood-water input to Lake A would likely occur. In this case, the surface water inputs from floods would buffer the sensitivity of Lake A to groundwater quality changes originating from its watershed. On the other hand, if floods become less important and/or less frequent, we can expect that the water quality of Lake A would be more dependent on regional groundwater quality. In such a case, the geochemistry of Lake A could potentially shift towards that of Lake B, and an increase of the salinity and the concentration of $Na^+$, $Ca^{2+}$, $SO_4^{2+}$ and $Cl^-$ would be expected for Lake A.

## 5.2 Implications for water management

Water budget assessments at natural lakes can serve as a tool for quantifying local human impacts (i.e., land use changes and climate changes) on water resources Arnoux et al. (2017b). Based on the results of this study, it becomes apparent that water budget assessments at artificial lakes (such as Lake A) can also be used to track human impacts on water resources. Recurring water budget assessments at a specific lake over time will serve to document changes in groundwater and surface water apportionment and can help to detect changes in local groundwater availability, and to anticipate impacts on local water supply utilities. As the response time of a lake to changes is controlled by its flushing time, the evolution of the G-Index will manifest at various rates. Indeed, lakes with different $t_f$ would reflect changes at different timescales. For instance, lakes with $t_f > 5$ yr would be expected to respond to decadal changes, while lakes with $t_f < 5$ yr would track annual or interannual variability. By analogy, we might postulate that it would be informative to study lakes with rapid response times (i.e., $t_f < 1$ yr), as they will act as precursors of the evolution of nearby surface water bodies characterized by longer flushing times.

As demonstrated, isotopic approaches may be efficiently employed to solve water budget unknowns as the method can be performed at low-cost and requires limited sampling and monitoring efforts for flood-affected environments which may be difficult or dangerous to monitor using traditional approaches. To enhance the effectiveness of our approach, the sampling strategy may be improved. Firstly, surface water sampling for isotopic analyses is recommended during turnover periods (i.e., springtime and autumn) and should be combined with depth-resolved measurements of physico-chemical parameters to confirm the vertical homogeneity or stratification. Secondly, for long-duration flood events, monitoring of potential evolution in flood-water isotopic signatures could help to improve the accuracy and realism of the model. Groundwater level monitoring and groundwater sampling in the vicinity of the lake could also help to strengthen the conceptual model by providing data to interpret the direction of groundwater fluxes and the variability of isotopic composition through time.

## 6 Conclusions

In this study, a volume-dependent transient isotopic mass balance model was developed and applied to a flood-affected lake in an ungauged basin in southern Quebec (Canada). This allowed for better understanding of the resilience of a flood-affected lake to changes in the surface/groundwater water balance partition, to understand the role of flood-water, and to predict

resilience to groundwater quantity and quality changes for a local water supply. Given the contrasting isotopic signature of the flood-water, the isotopic mass balance model was effectively applied at the study site. We anticipate that the isotopic framework is likely to be transferable to other lake systems subject to periodic flooding including lowland lakes fed by mountain flood-waters, river deltas, wadis, or nival (snowmelt-dominated) regimes, the latter of which dominates the high latitude and high altitude cold-regions including much of the Canadian landmass.

The isotopic mass balance model revealed that groundwater inputs dominated the annual water budget. To test the sensitivity, representativeness and resilience of the model, several model scenarios were evaluated to account for uncertainty in important input variables. Despite sensitivity to some variables, all model scenarios considered in the sensitivity analysis converged on the results that Lake A is mainly dependent on groundwater inputs and has a rapid (<1 year) flushing time by groundwater, suggesting that Lake A would be highly sensitive to groundwater quantity and quality changes.

When taking into account for potential subsurface storage, a better fit could be obtained between the modelled and depth-averaged isotopic signature of the lake, suggesting that the contribution of flood-water-like subsurface inputs is important to consider when assessing for water balance at flood-affected lakes. In fact, the increased contribution of surface water (from subsurface storage) resulted in a lower the contribution from groundwater and, consequently, in an increased resilience to groundwater changes. This finding provides a basis for postulating the impact of climate change on the water quality of Lake A. If the importance of floods increases, more flood-water inputs to Lake A can be expected during springtime, causing increased recharge. In this case, the surface water inputs from floods would increase the resilience of flood-affected lakes to groundwater quantity and quality changes at the watershed scale. On the other hand, if floods become less severe and/or less frequent, we can expect that the water quality of flood-affected lakes become more dependent on regional groundwater quality. From a global perspective, performing water balance assessments at lakes with rapid flushing time (<1 year) can help to predict the evolution of other surface water bodies with longer flushing times in their vicinity and, therefore, is useful for establishing regional-scale management strategies for maintaining lake water quality.

**Table A1. Detailed source information and download links for the openly available geospatial data in Figure 1. All data are openly available and are used in accordance with the Open Government Licence – Canada or the Open Data Policy, M-13-13 of the United States Census Bureau.**

| Layer | Database description | Author | Year | Database website | Download link (if applicable) |
|---|---|---|---|---|---|
| Canada borders (contours) | Provinces and territories (cartographic boundary file) | Statistics Canada© | 2016 | https://www12.statcan.gc.ca/census-recensement/2011/geo/bound-limit/bound-limit-2016-eng.cfm | From database website |
| USA borders (contours) | Nation and states (cartographic boundary file) | United States Census Bureau© | 2018 | https://www.census.gov/geographies/mapping-files/time-series/geo/carto-boundary-file.2018.htmll | From database website |
| Ottawa River watershed | Ontario watershed boundaries | Provincial Mapping Unit, Government of Ontario© | 2019 | https://geohub.lio.gov.on.ca/datasets/53a1c537b320404087c54ef09700a7db?geometry=-108.934%2C40.791%2C-53.431%2C51.408 | From database website |
| Urban area | Census metropolitan area (cartographic boundary file) | Statistics Canada© | 2016 | https://www12.statcan.gc.ca/census-recensement/2011/geo/bound-limit/bound-limit-2016-eng.cfm | From database website |
| Lakes and streams | National Hydrographic Network (NHN_0210001 and NHN_02OAA01) | Natural Resources Canada© | 2017 | https://www.nrcan.gc.ca/science-and-data/science-and-research/earth-sciences/geography/topographic-information/geobase-surface-water-program-geeau/national-hydrographic-network/21361 | https://ftp.maps.canada.ca/pub/nrcan_rncan/vector/geobase_nhn_rhn/shp_en/02/ |
|  | CanVec Hydro (watercourse_1 and waterbody_2) | Natural Resources Canada© | 2017 | https://ftp.maps.canada.ca/pub/nrcan_rncan/vector/canvec/shp/ | https://ftp.maps.canada.ca/pub/nrcan_rncan/vector/ |

| | | | | canvec/shp/Hydro/ |
|---|---|---|---|---|
| Flooded area | Flood Extent Polygon (Lac des Deux Montagnes, Quebec - 2017-05-06 22:54:32) | Natural Resources Canada© | 2017 | https://open.canada.ca/data/en/dataset/34085f6d-106a-41af-a29b-53ed6947c249 | ftp://data.eodms-sgdot.nrcan-rncan.gc.ca/EGS/2017/Flood_Products/QC/ |

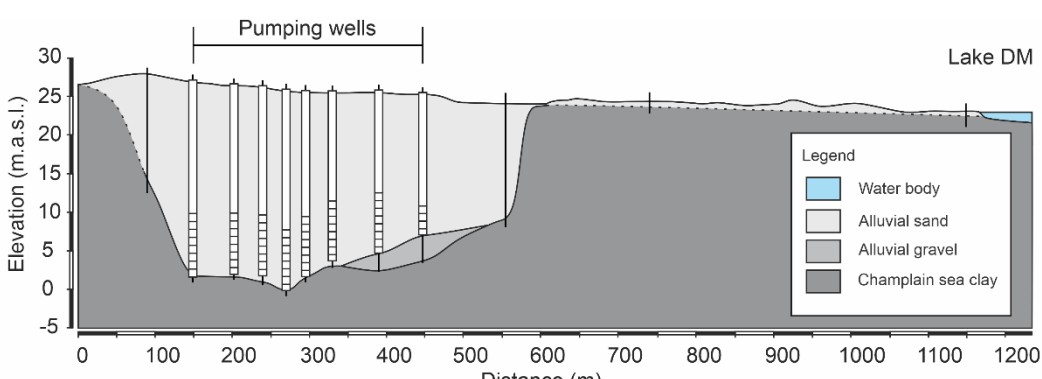

**Figure A1. Geological cross-section along the pumping wells showing the buried valley carved into the Champlain Sea clays and filled with alluvial gravels and sands.**

## Appendix B

### Computation of isotope mass balance parameters

The parameter $f(u)$, for the estimation of E (Eq. (5)), is calculated according to the area-dependent expression described by McJannet et al. (2012):

$$f(u) = (2.36 + 1.67u)A^{-0.05} \tag{6}$$

where $u$ is the wind speed (m s$^{-1}$) measured at 2 m above the ground and $A$ is the area (m$^2$) of the lake. Note that Eq. (6) was developed for land-based meteorological data.

The isotopic composition of the evaporating moisture ($\delta_E$) is estimated based on the Craig and Gordon (1965) model and, as described by Gonfiantini (1986), is:

$$\delta_E = \frac{\frac{(\delta_L - \varepsilon^+)}{\alpha^+} - h\delta_A - \varepsilon_K}{1 - h + 10^{-3}\varepsilon_K} \; (‰) \tag{7}$$

where $h$ is the relative humidity normalized to water surface temperature (in decimal fraction), $\delta_A$ is the isotopic composition of atmospheric moisture (described later on), $\varepsilon^+$ is the equilibrium isotopic separation and $\varepsilon_K$ is the kinetic isotopic separation, with $\varepsilon^+ = (\alpha^+ - 1)10^3$ and $\varepsilon_K = \theta^* C_K (1-h)$. $\alpha^+$ is the equilibrium isotopic fractionation, $\theta$ is a transport resistance parameter and $C_K$ is the ratio of molecular diffusivities of the heavy and light molecules. $\theta$ is expected to be close to 1 for small lakes (Gibson et al., 2015) and $C_K$ is typically fixed at 14.2 ‰ and 12.5 ‰ for $\delta^{18}O$ and $\delta^2H$ respectively in lake studies as these values represent fully turbulent wind conditions (Horita et al., 2008). Experimental values for $\alpha^+$ were used (Horita and Wesolowski, 1994):

$$\alpha^+(^{18}O) = \exp\left[-\frac{7.685}{10^3} + \frac{6.7123}{(T+273.15)} - \frac{16666.4}{(T+273.15)^2} + \frac{350410}{(T+273.15)^3}\right] \tag{8a}$$

$$\alpha^+(^2H) = \exp\left[1158.8\left(\frac{(T+273.15)^3}{10^{12}}\right) + 1620.1\left(\frac{(T+273.15)^2}{10^9}\right) + 794.84\left(\frac{(T+273.15)}{10^6}\right) - \frac{161.04}{10^3} + \frac{2999200}{(T+273.15)^3}\right] \tag{8b}$$

where T is the water surface temperature (°C), which was estimated according to the equilibrium method as described by de Bruin (1982) (see Appendix D).

The parameters m and $\delta_s$, for the computation of $\delta_L$ (Eq. (3)), are calculated as (Gibson, 2002):

$$m = \frac{\left(h - 10^{-3} \cdot \left(\varepsilon_K + \frac{\varepsilon^+}{\alpha^+}\right)\right)}{(1 - h + 10^{-3} \cdot \varepsilon_K)} \tag{9}$$

$$\delta_S = \frac{\delta_I + mX\delta^*}{1 + mX} \tag{10}$$

where, and $\delta^*$ is the limiting isotopic composition that the lake would approach as V $\to$ 0 and is calculated as:

$$\delta^* = \left(h\delta_A + \varepsilon_K + \frac{\varepsilon^+}{\alpha^+}\right) \Big/ \left(h - 10^{-3} \cdot \left(\varepsilon_K + \frac{\varepsilon^+}{\alpha^+}\right)\right) \tag{11}$$

The isotopic composition of atmospheric moisture ($\delta_A$) is estimated using the partial equilibrium model of Gibson et al. (2015):

$$\delta_A = \frac{\delta_P - k\varepsilon^+}{1 + 10^{-3} \cdot k\varepsilon^+} \tag{12}$$

where $\delta_P$ is the isotopic composition of precipitation and k is a seasonality factor, fixed at 0.5 in this study. The k value (ranging from 0.5 to 1) is selected to provide a best-fit between the measured and modelled local evaporation line. In Eq. (12), $\delta_P$ and monthly exchange parameters ($\varepsilon^+$, $\alpha^+$ and $\varepsilon_K$) are evaporation flux-weighted based on daily evaporation records.

## Appendix C

**Comparison of the evaporative fluxes (E) estimations**

See Fig. C1

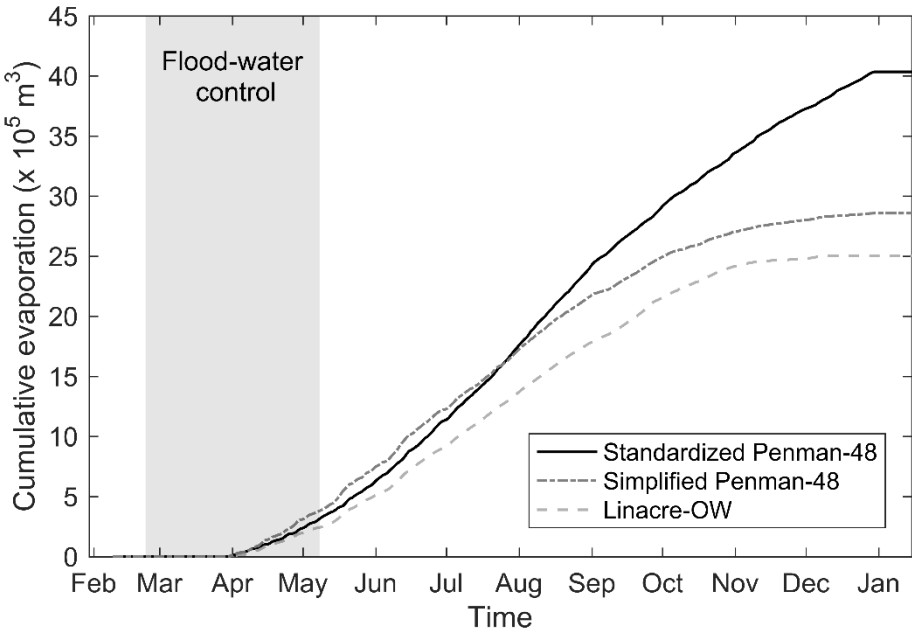

**Figure C1. Cumulative evaporative fluxes from Lake A via the standardized Penman-48, simplified Penman-48 (Valiantzas, 2006) and Linacre-OW (Linacre, 1977) equations.**

## Appendix D

**Estimation of the water surface temperature based on the equilibrium method (de Bruin, 1982)**

The water surface temperature (T) was estimated via the equilibrium method presented by de Bruin (1982), because no continuous measurements were available. This model is based on the assumption of a well-mixed surface body and was developed from standard land-based weather data. It was tested on two adjacent reservoirs in the Netherlands with average depths of 5 m and 15 m, respectively. Similarly to de Bruin (1982), we used the 10-day mean values, because we are interested in the annual variations of the water temperature. Moreover, the 10-day mean values were found to better simulate the observed water surface temperature. Differences between the observed and modelled water temperature is typically ≤1 °C, except in July and December where discrepancies of up to 5 °C were observed (Fig. D1). This is likely because Lake A develops a thermal stratification over summertime and in wintertime. Potential uncertainties in isotopic mass balance models due to stratification in lakes up to 35 m were previously described and discussed by Gibson et al. (2017) and Gibson et al. (2019). They reported that sampling methods and lake stratification can lead to volume-dependent bias in the water balance partition. In this study, not accounting fully for thermal stratification will lead to overestimation of evaporation fluxes, and groundwater exchange will potentially be underestimated.

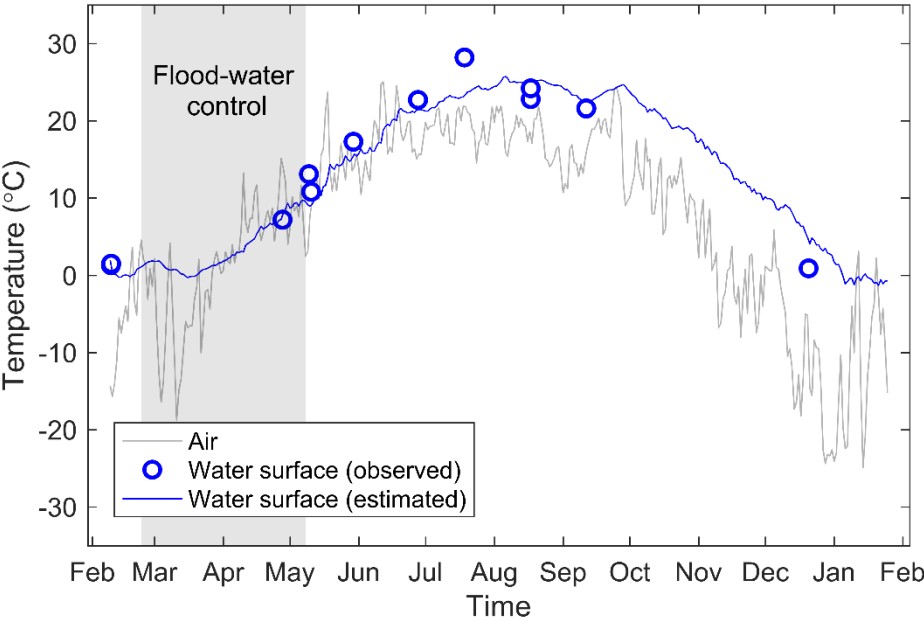

**Figure D1. Temporal evolution of air temperature and observed and estimated water surface temperatures at Lake A. Water surface temperature estimations were computed according to the equilibrium method described by de Bruin (1982).**

**Appendix E**

**Isotopic composition along vertical profiles in Lake A**

See Fig. E1

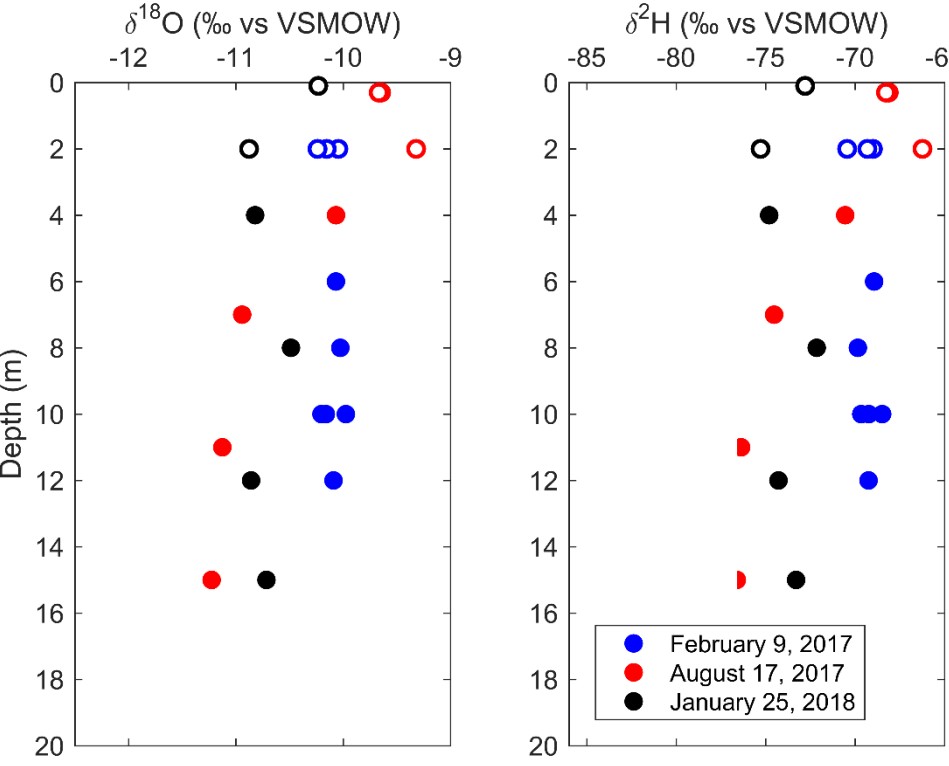

**Figure E1. Isotopic composition of Lake A water samples against depth on February 9, 2017, August 17, 2017 and January 25, 2018. The hollow circles and solid circles represent samples collected at ≤2 m and >2 m depth, respectively.**

## Appendix F

**Results of the sensitivity analysis for reference scenarios A and B**

See Table F1 and Table F2.

**Table F1. Sensitivity analysis on the input parameters of the isotopic mass balance model. Q is the output flux from Lake A, I the input flux and $t_f$ the mean flushing time by groundwater.**

| Scenario | | Maximum Q | Minimum Q | Mean Q | | Mean I | | $t_f$ |
|---|---|---|---|---|---|---|---|---|
| | | | | Flooding | Annual | Flooding | Annual | |
| | | (x 10⁴ m³/day) | (x 10⁴ m³/day) | (x 10⁴ m³/day) | | (x 10⁴ m³/day) | | (days) |
| A | Reference | 8.0 | 3.7 | 5.64 | 4.77 | 6.61 | 4.86 | 135 |
| A01 | V + 3% (slope 30°) | 8.0 | 3.7 | 5.64 | 4.77 | 6.61 | 4.86 | 140 |
| A02 | V - 8% (slope 20°) | 7.8 | 3.7 | 5.55 | 4.72 | 6.51 | 4.81 | 130 |
| A03 | $\delta_{Is}$ $^{18}$O + 0.5 ‰ $\delta_{Is}$ $^{2}$H + 4.06 ‰ | 25.0 | 1.0 | 11.82 | 6.99 | 12.79 | 7.08 | 107 |
| A04 | $\delta_{Is}$ $^{18}$O - 0.5 ‰ $\delta_{Is}$ $^{2}$H - 4.06 ‰ | 4.3 | 4.2 | 4.25 | 4.22 | 5.21 | 4.31 | 146 |
| A05 | $\delta_{G}$ $^{18}$O + 0.5 ‰ $\delta_{G}$ $^{2}$H + 4.06 ‰ | | | Not possible to fit data | | | | |
| A06 | $\delta_{G}$ $^{18}$O - 0.5 ‰ $\delta_{G}$ $^{2}$H - 4.06 ‰ | 10.0 | 1.0 | 5.06 | 3.25 | 6.02 | 3.34 | 227 |
| A07 | $\delta_A$ minimum | | | Not possible to fit data | | | | |
| A08 | $\delta_A$ maximum | 8.0 | 4.0 | 5.80 | 5.00 | 6.77 | 5.09 | 128 |
| A09 | E + 20% | 8.0 | 4.8 | 6.24 | 5.60 | 7.22 | 5.72 | 112 |
| A10 | E - 20% | 8.0 | 2.7 | 5.09 | 4.02 | 6.05 | 4.09 | 1665 |
| A11 | RH + 10% | | | Negligible change | | | | |
| A12 | RH - 10% | | | | | | | |
| A13 | $T_{air}$ + 10% | 8.0 | 3.9 | 5.75 | 4.92 | 6.71 | 5.01 | 130 |
| A14 | $T_{air}$ - 10% | 8.0 | 3.5 | 5.53 | 4.62 | 6.50 | 4.71 | 140 |
| A15 | U + 10% | 8.0 | 3.9 | 5.75 | 4.92 | 6.72 | 5.01 | 130 |
| A16 | U - 10% | 8.0 | 3.6 | 5.58 | 4.70 | 6.55 | 4.78 | 138 |
| A17 | P + 10% | | | Negligible change | | | | |
| A18 | P - 10% | | | | | | | |
| A19 | $T = T_{air}$ | 10.0 | 2.9 | 6.10 | 4.67 | 7.07 | 4.73 | 145 |
| A20 | Rs + 10% | 8.0 | 3.9 | 5.75 | 4.92 | 6.72 | 5.02 | 130 |
| A21 | Rs - 10% | 8.0 | 3.6 | 5.58 | 4.70 | 6.55 | 4.78 | 138 |
| A22 | LMWL (PWLSR method) | 7.0 | 1.6 | 4.04 | 2.95 | 5.00 | 3.03 | 237 |

**Table F2. Sensitivity analysis on the input parameters of the isotopic mass balance model for the reference scenario B. Q is the output flux from Lake A, I the input flux and $t_f$ the mean flushing time by groundwater.**

| | Scenario | Maximum Q (x $10^4$ m³/day) | Minimum Q (m³/day) | Mean Q Flooding (x $10^4$ m³/day) | Mean Q Annual | Mean I Flooding (x $10^4$ m³/day) | Mean I Annual | $t_f$ (days) |
|---|---|---|---|---|---|---|---|---|
| B | Reference | 28.0 | 1.0E+03 | 12.68 | 7.07 | 13.65 | 7.16 | 110 |
| B01 | V + 3% (slope 30°) | 28.0 | 1.0E+01 | 12.63 | 6.99 | 13.59 | 7.07 | 115 |
| B02 | V - 8% (slope 20°) | 26.0 | 1.0E+01 | 11.73 | 6.49 | 12.69 | 6.57 | 114 |
| B03 | $\delta_{Is}$ $^{18}$O + 0.5 ‰<br>$\delta_{Is}$ $^{2}$H + 4.06 ‰ | | | Not possible to fit data | | | | |
| B04 | $\delta_{Is}$ $^{18}$O - 0.5 ‰<br>$\delta_{Is}$ $^{2}$H - 4.06 ‰ | 12.0 | 2.5E+04 | 6.78 | 4.87 | 7.75 | 4.95 | 141 |
| B05 | $\delta_{G}$ $^{18}$O + 0.5 ‰<br>$\delta_{G}$ $^{2}$H + 4.06 ‰ | | | Not possible to fit data | | | | |
| B06 | $\delta_{G}$ $^{18}$O - 0.5 ‰<br>$\delta_{G}$ $^{2}$H - 4.06 ‰ | | | Not possible to fit data | | | | |
| B07 | $\delta_A$ minimum | 26.0 | 1.0E+01 | 11.73 | 6.49 | 12.69 | 6.57 | 120 |
| B08 | $\delta_A$ maximum | | | Negligible change | | | | |
| B09 | E + 20% | 28.0 | 1.0E+04 | 13.18 | 7.74 | 14.15 | 7.84 | 96 |
| B10 | E - 20% | 27.0 | 1.0E+01 | 12.18 | 6.74 | 13.13 | 6.80 | 116 |
| B11 | RH + 10% | | | Negligible change | | | | |
| B12 | RH - 10% | | | | | | | |
| B13 | $T_{air}$ + 10% | | | Negligible change | | | | |
| B14 | $T_{air}$ - 10% | | | | | | | |
| B15 | U + 10% | 28.0 | 2.0E+03 | 12.74 | 7.14 | 13.70 | 7.23 | 108 |
| B16 | U - 10% | 28.0 | 1.0E+01 | 12.63 | 6.99 | 13.59 | 7.08 | 111 |
| B17 | P + 10% | | | Negligible change | | | | |
| B18 | P - 10% | | | | | | | |
| B19 | T = $T_{air}$ | 28.0 | 1.0E+01 | 12.63 | 6.99 | 13.60 | 7.05 | 112 |
| B20 | Rs + 10% | 28.0 | 3.0E+03 | 12.79 | 7.22 | 13.76 | 7.31 | 106 |
| B21 | Rs - 10% | 28.0 | 1.0E+01 | 12.63 | 6.99 | 13.59 | 7.07 | 111 |
| B22 | LMWL (PWLSR method) | 16.0 | 1.0E+01 | 7.22 | 4.00 | 8.18 | 4.08 | 199 |

## Code and data availability

The code and data are available on request to the corresponding author.

## Author contribution

JMD: Conceptualization, Data curation, Investigation, Methodology, Visualization, Roles/Writing - original draft. FB: Conceptualization, Methodology, Supervision, Writing - review & editing. PB: Conceptualization, Funding acquisition, Project administration, Supervision, Writing - review & editing. JG: Methodology, Writing - review & editing.

## Competing interests

The authors declare that they have no conflict of interest.

## Acknowledgements

This research was funded by NSERC, grant numbers CRSNG-RDCPJ: 523095-17 and CRSNG-RGPIN-2016-06780.The authors are grateful to the Town and G. Rybicki to allow access and water sampling on their property. Thanks to the students (M. Patenaude, T. Crouzal, R.-A. Farley, just to name a few) who participated in the fieldwork. We also gratefully acknowledge J.-F. Helie and M. Tcaci from Geotop-UQAM and M. Leduc and J. Leroy from the Laboratoire de géochimie de Polytechnique Montréal. We thank Chani Welch, Bruno Hamelin, Michael Ronsen and two anonymous reviewers for their thorough reading of the manuscript and their many insightful comments and suggestions which helped at improving the manuscript.

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
