# Peer review of "Quantifying flood-water impacts on a lake water budget via volumedependent transient stable isotope mass balance"

_Hydrology and Earth System Sciences, 2020_

## Short Comment (SC1) · 22 May 2020

While this paper is a detailed look at a small man-made (or influenced lake), it isn't clear what the overall usefulness is to others working on larger and more complex lake systems. The manuscript is overall relatively well written, but there are many parts that aren't always clear. Most importantly there is no discussion outside of the local issues of the lake, which makes this a very site specific study.

There are also many line by line points that need to be made. These are as follows:

Line 43: The reference to Klove et al, 2011 is to Groundwater Dependent Ecosystems

[Figure]

(GDE) and lakes are not the same thing as GDEs, they are a subset of GDEs in most, but not all cases. This reference should be something specific to lake systems. GDE also refer to streams wetlands and other non-lake surface waters. For example, Rosen 2015 would be a better reference. Rosen, M.R., 2015, The influence of hydrology on lacustrine sediment contaminant records. In Blais, J.M., Rosen M.R., Smol J.P. (eds) Environmental Contaminants: Using natural archives to track sources and long-term trends of pollution. Springer, Dordrecht. 5 – 33 p. https://DOI.org10.1007/978-94-017-9541-8_2 Line 44: "few decades"... references only list the last decade. you could add: Herczeg AL, Leaney FW, Dighton JC, Lamontagne S, Schiff SL, Telfer AL, English MC (2003) A modern isotope record of changes in water and carbon budgets in a groundwater-fed lake: Blue Lake, South Australia. Limnol Oceanogr 48:2093–2105 if you want to go back two decades.

Line 56: …. "but occur over a 1 km long area." Do you mean 1 km "wide" area? The length of the river or canal is of no importance, it is the width that will make it hard to measure flux. Please change to "wide"

Line 58: "The democratization of isotope mass balances in Quebec…." What does the "democratization" of isotope mass balance mean? Was this auto corrected from the original word to be used. I hope so, as I had no idea that isotopes were political! Should the word be "demonstration"? Not really sure what is going on here.

Line 60-70: It would be good to include Herczeg et al (2003) here as well as they determined changes in isotopic composition due to groundwater pumping, this also shows how transient changes can affect the isotopic composition of lakes.

Line 78: There is no hypothesis indicated in this manuscript. The objectives are clear but there is no indication of what mechanisms they propose may be important. A hypothesis should be added.

Figure 1. Water courses shown don't match up with the description. There is supposed to be one inlet and outlet to Lake A, but at least two inlets are shown (or outlets). Flow

[Figure]

directions are needed on the other streams (canals?) shown. The reorienting of the North arrow is somewhat confusing, probably needed. All sampling is done in one corner of the lake, why was this done? Presumably the lake is well mixed? Samples taken near the shore, like LB-S1 could have some evaporation signature in them. Was this accounted for? If Lake DM really has a name (Deux-Montagnes), then the name should be put on the map with the (DM) in parentheses. One might also argue to use the French for the whole name, Lac des Deux Montagnes. Lake B also appears to be called Lac Val des Sables in google earth, is this not correct?

Line 116: "All water levels are reported relative to a reference water levels measured on February 9, 2017" One reference water level or many? Please fix, this is a combination of both.

Line 118 change to "...over the Ottawa River watershed..."

Line 125-126: "....synchronous with those of Lake DM (Fig. 2) from late February to late July 2017". How do you know water levels are synchronous from Late February when water level measurements weren't begun until April? This can't be known. Given the sparse data in figure 2, and the non-synchronous relation between the observation well and Lake DM in the autumn, this can't be conclusively known. In addition, some of the well peaks appear to actually occur before the lake level rises, which is a bit strange. In any case, more information is needed to be able to say this. It may be true for the flood period, but that would be expected. The low flow period doesn't appear to be completely synchronous. While it may be true that Lake DM controls Lake A water level during flood periods and/or high water periods, there is no data presented that shows that Lake A water levels are synchronous with Lake DM during low flow or low water levels. Clearly the groundwater is not synchronous during September to November. Section 3.1 Field measurements section. There is no mention of calibration for the water level loggers. Without calibration how do you know they were synchronous or that they water levels were the same? Please give all the calibrations that were done on instruments and isotopic analyses. Section 3.2 Water sampling and analytical techniques: Other than ion balances, was any other QA/QC done? This needs to be stated. In addition, were isotope measurements compared with standard mass spectroscopy? Ring cavity measurements have been shown to be in error in some cases and should be viewed with some skepticism unless comparison is made to standard mass spectroscopy or the methodologies listed below have been followed. See Wassaner et al (2014) and Sengupta (2014) for examples. Also, were any samples taken under ice? Ice will fractionate the isotopic composition and make the mass balance different. Has this been accounted for? It also isn't clear why methods are included for water quality sampling. These data don't appear to have been used in this manuscript, so it simply takes up space. Please remove the methods for chemical sampling and concentrate only on the isotopic measurements. Perhaps more detail about replicates and comparisons to mass spectroscopy measurements can be done to alleviate concerns over the accuracy of the ring cavity measurements. References: Sengupta, S., 2014, Pros and Cons of Laser Based Isotope Measurements of Water and Real Time Vapour Samples: A User's Perspective. Gond. Geol. Mag., V. 29 (1 and 2), pp.45-51 Wassenaar, L.L., Coplen T.B., Aggarwal, P.K., 2014 Approaches for Achieving Long-Term Accuracy and Precision of $\delta$18O and $\delta$2H for Waters Analyzed using Laser Absorption Spectrometers. Environ. Sci. Technol. 2014, 48, 2, 1123-1131. Line 170: "The water and stable isotope mass balance of a well-mixed lake can be described..." The authors haven't actually demonstrated that the lake is well mixed. A figure showing the lake profiles should be presented. Line 198: So, evaporation was held constant for the entire month. Particularly in the spring, that is a brave assumption. This seems to be the coarsest time step. Why was this needed? Line 208: limiting isotopic composition (Gibson et al., 2015). This is not a common term. Although this can be found in the reference listed, it should be detailed more here. Line 216: " The above-mentioned equations are computed on a daily time step to calculate the isotopic composition of the lake ($\delta$L)." Yet, some parameters have monthly time steps. How do you reconcile that? Does this mean the monthly time steps aren't that important, or should it all be done monthly? This seems like a limitation to the daily time step. Line 218: It has been

stated a few times that the lake is well mixed, but this has not been demonstrated with any measurements. The reader needs evidence that the lake is well mixed, particularly over the time period of measurement, which is over the springtime period, when mixing may not be complete. Line 223: "Assuming homogenous hydraulic conductivity of the sediments" This is a big assumption and likely not accurate overall, but in a sandy aquifer, might be a reasonable assumption given other errors in the model. This should be explained more. Line 259 and 263: This is a pet peeve of mine, but "since' is a time word and shouldn't be used to replace "because", please change to "because" everywhere in the manuscript when it is not used as a temporal term. Line 269: change to "….lead to overestimation…" Line 270, so is the potential underestimation of groundwater exchange underestimated here? Or was something done to account for this. Please explain. Line 290-295: Why not just measure the GW input? Why does it need to be estimated from the intersection with the LEL? Also, although the evaporation process is the same between flood water and lake water (having the same slope) that is not unusual. What is unusual is that they don't intersect at the same place, so the floodwater is a different source from the recharge from GW or rainfall. There do appear to be five lake values (one of which appears to be unevaporated floodwater) that fall on the floodwater line, so there is some influence from floodwater on the isotopic composition of the lake. This should be address more fully. Line 331-332: The authors say: "Lake A volume variations are estimated from water level records assuming a constant lake area. When not available, the surface elevation of Lake A is assumed to be equal to the water level at other observation points." I don't understand what this means. Unless this is a pit lake with perfectly straight vertical sides, the Lake area will increase as elevation increases and it will take more water for fill shallower stage heights as the lake gets bigger. Please explain if this is not true for this lake. Furthermore, water levels in a well cannot be used unless there is no GW flow to the lake. If the groundwater level is the same as the lake level, then there will be no flow to the lake and the flow is stagnant. Has this been observed? If not, this GW elevation should not be used as a surrogate for lake level. Figure 2 actually show that

Lake A water level is at no time equal to Observation well VP, and is generally higher than the well elevation, except in late summer, suggesting the lake is losing water to the well except when precipitation slows down and the lake level lowers. Lake DM, which is a possible surrogate for Lake A elevation, is also never equal to the elevation of well VP, except on the rising limb of the floodwater. Therefore, the well VP elevation is not a good surrogate for lake A elevation and should not be used as such, unless a better explanation can be given. Line 338-340: This also a time of groundwater input (at least following the Lake DM elevation compared to Well VP). Is this considered in the fluxes? Line 359: So, here the vertical profiles are volume-weighted, which suggests the sides of the lake are not vertical, if they were then you wouldn't need to volume-weight them. But above you say you use a constant lake area to get the volume. Which is it? Line 382-384: you do have 3 vertical profiles; you could have at least estimated how big a difference using a stratified model using some max and min values for the isotopes. It also isn't clear from the discussion above this if the direction of groundwater low, in or out of the lake is considered, as the water level data suggests in changes through the modeling period. Table 2: A small point, but I'm not sure why commas are used in this table. Scientific notation usually uses a period even for large numbers. Europeans use commas for decimals and then periods for large numbers, so I'm not sure what style is being used here. I would prefer these to all be periods not commas. A larger point for this table is that the sensitivity analysis doesn't appear to use very wide values to check how sensitive the variables are. A change of 0.5 per mil for oxygen is not that far outside the error of the measurement. It looks like most of the differences looked at are between 10 and 20 percent. Is that reasonable, what is the variability of the rainfall amounts over time. Granted E isn't likely to have a large range, but some of the variable could have larger ranges than are estimated here. Line 414: What about groundwater influx at this time? Ok, I see discussed in the next section.

Line 440: Table 3 provides the relative importance of the hydrological processes for that year that was measured, not for an annual timescale. Measurements for all parameters weren't done for the whole year as well. This should be modified.

Line 485: tG the mean flushing time by groundwater isn't included in equation 13 and is instead written as tf, which I assume is the time of flushing (by groundwater). This needs to either be explained better, if I don't understand this, or the notation needs to be corrected. Everywhere else it is tf.

Figure 9. The caption also has reference to tG is this a different variable or is it tf?

The climate change part of this paper is somewhat of a throw away suggestion. There is really no data or simulations that support either conclusion and the modeling doesn't appear to help either. Given the possibility of either more or less flooding the conclusions seem pretty obvious.

While the model and the system are relatively well characterized it isn't clear what this gives other scientists other than a look at a local system. How can this be used in other lake systems and can a lake with fewer measurements or larger area or volume be characterized using this model? It would be good if some bigger questions were answered rather than just the local questions that have no real interest to scientists or the public outside of the area.

---

## Referee Comment (RC1) · Bruno Hamelin (Referee) · 31 May 2020

On one hand, I consider this study as a stimulating contribution to the ongoing effort of improving the reliability of isotope-based modeling for the prediction of lakes hydrological behavior. Based on an attractive, although complex, case study, it presents a commendable attempt to assess quantitatively the sensitivity of the model results to the variations of the parameters. On the other hand, reading the manuscript is a bit frustrating due to a number of weaknesses in the data set and hypotheses, and also in the description of the model algorithm.

As for the data, the confidence that we can have on the robustness of the authors'

conclusions clearly suffers from the lack of continuity of the lake level monitoring, and from the lack of documentation of the lake stratification. Due to the ice cover (line 124) and logger failure (line 140), the level of the lake is available only for a part of the flooding period. In particular, we miss the comparison between lake A and lake DM for summer and fall 2017, when lake DM was seemingly above the threshold, and should thus have overflown again into lake A (Figure 2). More importantly, the lack of temperature and isotope data during the peak of the flooding period, and around the fall overturning, casts a strong doubt on the justification of the assumption of lake homogeneity. This weakness is acknowledged by the authors at several places (line 260, 269-270, 311, 433, 519). However, the reader is left with the impression that a two-layer model would definitely be needed, and that the model results might be much more sensitive to the seasonal stratification than to the other parameters tested in the study.

Another source of worry arises from a possibly spurious choice for the isotope composition of the groundwater inflow end member. As soon as the authors show that one of the two main supplies of lake A, i.e. the flood water overflowing from lake DM, shows an isotope composition which is below the lake evaporation line, one would expect the other main source, i.e. groundwater, to lie above that line, and not on it (line 290). This needs to be thoroughly discussed, all the more as the authors emphasize that ïĄďG is among the most stringent parameters. More generally, some explanation is needed about the origin of the isotopic difference between the groundwater and flood water. Would it be possible to collect samples from the aquifer, away from the influence from the lake? How does the aquifer composition compare with the amount-weighted average of the rainfall seasonal variation?

On another aspect, I have some reservation about the way the model is described. A luxury of details is given on several very classical aspects already extensively described in previous works (i.e. Craig & Gordon's approach of the isotopic budgets), which could thus be better placed in appendix, while the description takes shortcuts on

other key linkage in the modeling procedure. For instance, the reader should not have to wait until line 357 (results section) to learn the hypotheses leading to the outflow estimate! Another example is the emphasis put on the volume-dependent modeling (line 49 and 54, 200-210). If the authors want to convince the reader that this is important, they have to better explain equation (6) to (10), which are quite cryptic, and compare with the same equation based on a constant volume approximation. This should also be discussed when looking at the results. This whole section should be written again, as a true instruction manual for anyone willing to apply such a model to another case-study.

In general, I have the feeling that the overall structure of the manuscript is a bit messy. I would recommend giving first all the information that can be deduced from the lake level variations (i.e. line 330-349), in order to introduce properly the aims of using isotopes to better unravel the contribution of the different sources.

Specific comments, in addition to those already pointed out by the other reviewer:

Line 220-239: "Outflow fluctuations were derived from water level variations at Lake A using linear interpolation between adjusted daily minimum and maximum outflow. Daily inflow into Lake A was calculated to compensate for the adjusted outflow, as the net water fluxes are required to be equal to the lake's daily volume variation." I still do not understand what is done exactly on this key point. This needs to be written in equations and related to the main unknowns in equations (1) to (10).

Line 275: "Interpolation was used to simulate the $\delta$P on a daily-time step." This suggests that the rain data show a smooth evolution through time along the season. Is this really the case?

Line 284: same evaporation slope for lake waters and flood water. Is this not surprising, as this slope depends on the climate parameters of Craig & Gordon's equation, while flooding and evaporation do not occur at the same period of the year?

Line 333: lake elevation assessed from well VP. Unclear what is meant by this statement as the difference of elevation of the water level between lake A and well VP is supposed to change with time along with the recharge/discharge alternation. (already pointed out by the other reviewer). Line 357: "the outflow fluxes are proportional to the lake's water level. We adjusted minimum and maximum outflow fluxes (Q) so that the latter respectively correspond to the minimum and maximum water levels." Again, (see comment above), I do not understand what this means. Line 368 and figure 6: The results obtained from ïĄďD are strictly redundant to those from ïĄď18O. What is really missing in this figure is some data at the beginning of May!

Line 434-437: scenarios A and B are supposed to compensate for the lack of data at the peak and end of the flood period. However, just mentioned like this without description, and sent back to Appendix C leaves a disastrous impression on the reader. Line 452: "The isotopic mass balance model revealed it was necessary to allow for significant groundwater outflow from Lake A during springtime to correctly reproduce the observed $\delta L$ ". A best illustration of this conclusion would have been to compare the results of the model with and without the groundwater outflow. Line 485-487: confusion between tG and tf. (already pointed out by the other reviewer).

Line 503-504: confusion on "increase" ?

---

## Referee Comment (RC2) · Anonymous Referee #2 · 24 Jun 2020

* GENERAL COMMENTS

In this manuscript entitled "Quantifying flood-water impacts on a lake water budget via volume-dependent transient stable isotope mass balance", the authors focus on an artificial lake and justify their study by stating that "[understanding] the relative importance of the hydrological processes in lakes can also help to depict the vulnerability and/or resilience of a lake to pollution". They aim to develop a predictive model of both atmospheric and water balance controls on isotopic enrichment, quantify of flood-water inputs to the lake, and conduct a model sensitivity analysis was conducted to evaluate potential sources of uncertainty. Overall, the manuscript is of appropriate length and

well written. Figures and tables are also of good quality and rich in information without being too crowded. While I enjoyed reading this manuscript, I think that the authors need to make a strong case for the broader relevance, impact and transferability of their methods or conclusions, in addition to revisiting the structure of manuscript. My most major criticisms are as follows:

** In its present state, the manuscript pretty much reads like a case study report. There is nothing wrong with case studies per se, as the uniqueness of place makes the conclusions of many papers inherently site-specific. That being said, I think that the authors should try to extrapolate their conclusions (or speculate about how their conclusions might extend) to other lakes (artificial or not) in Canada, North America and around the World. What makes Lake A and Lake DM different (or not) than other lakes where similar isotope mass balance approaches have been used in the past? In other words, what makes the present study novel? What are the really key contributions that represent an advancement to the science – and that may be relevant beyond the particular site that the authors focused on? Can the results be extrapolated to depressional wetlands which are affected by flooding as well? And if results and conclusions cannot be extrapolated, what about some of the methods applied in the current manuscript? My asking those questions is not my way to say that there are no novel contributions in this manuscript, but rather to say the authors have not explicitly identified them and should highlight them better.

** The introduction lacks an overarching goal or research question for the study, as well as specific research objectives or questions. Instead, the last paragraph of the intro just states that the present study builds upon two other studies. The only sentence of the introduction that could be seen as a research goal is the one that reads as: "The main purpose of this study was initially to expand our understanding of flood-affected lake dynamics in the context of a seasonal climate". It is quite vague, though, so I suggest that the authors include some more specific objectives or questions at the end of the introduction. This should also help highlight the novel contributions that the present

study intends to make.

** There seem to be a lot of "results" that are listed in Section 3, which most readers would equate to a Methods section (and not a results section). I would suggest that the authors try to reorganize their text a bit, so that section 3 only focuses on methods while section 4 summarizes results.

** Following up on the previous point: Section 4 does not seem to focus on "plain results" only, as it includes several interpretations, discussions and linkages to the literature. Section 4 therefore reads as a combined Results and Discussion section, which is a bit surprising as there is a separate discussion section later. I suggest that the authors try to better distribute methodological aspects, results and discussions/interpretation into distinct sections (and sub-sections).

** Along the same lines as the two previous points, Section 5 is a bit confusing. There are completely new results (e.g., Table 3, Figure 7) first reported on in this section. Conversely, there are not a lot of literature references (none in sub-section 5.1, and only 1 literature reference, as far as I can see, in sub-section 5.2). So, a lot of the text listed under the "5 - Discussion" header does not really read like a typical discussion section, in the sense that there is very little confrontation of the present study results with the existing literature. The authors should rectify that as much as possible.

** Sub-section 5.3 is a bit confusing. The authors provided a list of physical water quality parameters + ions earlier in their manuscript. Based on their introduction, I expected those physical water quality parameters + ions to be used to support the "surface water pollution" aspects of the manuscript. However, in sub-section 5.3, there is no reference to those parameters/ions, and the assessment of resilience to water pollution is solely based on mean flushing time. Why were the parameters/ions described earlier not used? And is it adequate to use the mean flushing time as a proxy measure for a lake's resilience to "all" surface water pollution, regardless of the reactivity/sorption coefficients of the chemical determinants under consideration? This last question may

be out of scope for the present manuscript, but a clarification sentence would help manage readers' expectations.

** The first sentence of the conclusion states reiterates that the "goal" of the present study was to "develop a volume-dependent transient isotopic mass balance model, assuming well-mixed conditions, in order to better understand the dynamics of the hydrological processes at a flood-affected lake in southern Canada". As commented upon above, I find this to be rather vague. After reading through the details of the manuscript, it seems like the authors specifically want to address questions related to the relative importance of groundwater for Lake A on an aggregated annual scale as well as through different seasonal/wetness conditions (what they refer to as temporal variability of hydrological processes). The authors also dedicated a fair amount of time/manuscript space to discuss many different elements, e.g.: the peculiarities of lake dynamics under flooding conditions, uncertainties associated with their isotope mass balance model (those uncertainties are multiple in nature, i.e., input data uncertainty, structural data uncertainty, output data uncertainty, even maybe model parameter uncertainty), and the application of pollution resilience assessment framework. It is quite difficult, from the whole manuscript, to figure out which of those elements are primary versus secondary targets/goals/objectives of the manuscript and how they relate (or not) to one another.

I think that there is a nice science story in the manuscript, and I hope the authors will see my comments as suggestions for strengthening it and making it interesting to the broad readership of HESS.

* SPECIFIC COMMENTS ABOUT SOME TEXT SECTIONS OR FIGURES

See "sticky notes" and yellow highlights in the pdf proofs

* TYPOS AND EDITORIAL SUGGESTIONS

See "sticky notes" and yellow highlights in the pdf proofs

[Figure]

Please also note the supplement to this comment:
https://www.hydrol-earth-syst-sci-discuss.net/hess-2020-101/hess-2020-101-RC2-
supplement.pdf

———————————————

[Figure]

**Supplement:**

5

**Quantifying flood-water impacts on a lake water budget via volumedependent transient stable isotope mass balance**

Janie Masse-Dufresne1, Florent Barbecot2, Paul Baudron1,3 and John Gibson4,5

[revised manuscript text omitted]

---

## Author Comment (AC1) · 21 Jul 2020

article [utf8]inputenc

graphicx xcolor ragged2e textcomp

While this paper is a detailed look at a small man-made (or influenced lake), it isn't clear what the overall usefulness is to others working on larger and more complex lake systems. The manuscript is overall relatively well written, but there are many parts that aren't always clear. Most importantly there is no discussion outside of the local issues of the lake, which makes this a very site specific study.

[Figure]

There are also many line by line points that need to be made. These are as follows:

**SC1-1.** Line 43: The reference to Klove et al, 2011 is to Groundwater Dependent Ecosystems (GDE) and lakes are not the same thing as GDEs, they are a subset of GDEs in most, but not all cases. This reference should be something specific to lake systems. GDE also refer to streams wetlands and other non-lake surface waters. For example, Rosen 2015 would be a better reference. Rosen, M.R., 2015, The influence of hydrology on lacustrine sediment contaminant records. In Blais, J.M., Rosen M.R., Smol J.P. (eds) Environmental Contaminants: Using natural archives to track sources and long-term trends of pollution. Springer, Dordrecht. 5 – 33 p. https://DOI.org10.1007/978-94-017-9541-8

**Answer: Done.** We agree with the reviewer. The suggested reference will be used instead.

**SC1-2**. Line 44: "few decades"... references only list the last decade. You could add: Herczeg AL, Leaney FW, Dighton JC, Lamontagne S, Schiff SL, Telfer AL, English MC (2003) A modern isotope record of changes in water and carbon budgets in a groundwater-fed lake: Blue Lake, South Australia. Limnol Oceanogr 48:2093–2105 if you want to go back two decades.

**Answer: Done.** Thank you. The suggested reference will be used instead.

**SC1-3.** Line 56: "...but occur over a 1 km long area." Do you mean 1 km "wide" area? The length of the river or canal is of no importance, it is the width that will make it hard to measure flux. Please change to "wide".

**Answer: Done.** You are right. We meant 1 km "wide" area. This is to be corrected in the manuscript.

**SC1-4.** Line 58: "The democratization of isotope mass balances in Quebec..." What does the "democratization" of isotope mass balance mean? Was this auto corrected

from the original word to be used. I hope so, as I had no idea that isotopes were political! Should the word be "demonstration"? Not really sure what is going on here.

**Answer: Clarification.** This is a concept we translated from French, but the Reviewer's comment made us realize that the meaning is not accurate in English. We wanted to convey the fact that isotopic methods are not widely used in Quebec (Canada) yet and that we could benefit from their application. As this is only a local implication, we opt to withdraw this sentence from the manuscript.

**SC1-5.** Line 60-70: It would be good to include Herczeg et al (2003) here as well as they determined changes in isotopic composition due to groundwater pumping, this also shows how transient changes can affect the isotopic composition of lakes.

**Answer: Done.**This publication is indeed very interesting and is related to our study. Herczeg et al. (2003) demonstrated the impact of various forcings, such as the rainfall variability, land-use changes and increased pumping rates, on the water residence time in the Blue Lake, in Australia. Note that the pumping corresponds to direct surface water abstraction (Lamontagne and Herczeg, 2002), and not groundwater pumping. Nonetheless, Herczeg et al. (2003) showed the importance of studying lake water budgets in order to identify potential governing forcing in order to secure water quantity and quality overtime, as lakes are important water resources for the production of drinking water. In that sense, the following material could be added to Line 43:

*"...and to secure water quantity and quality overtime for drinking water production purposes (Herczeg et al., 2003)."*

**SC1-6.** Line 78: There is no hypothesis indicated in this manuscript. The objectives are clear but there is no indication of what mechanisms they propose may be important. A hypothesis should be added.

**Answer: Done.** Some information concerning our conceptual model would help to better understand the objectives of the developed model. We propose to add the following at Line 74:

*"In this hydrological context, it was conceptualized that groundwater inputs to the lake are very limited, while flood-water inputs (via the surface) and groundwater outputs are governing the water balance during the flooding event. Contrastingly, as the water level of the lake decreased after the flooding event, it was assumed that groundwater inputs and surface water outflows were dominant over the lake water balance during low water level periods."*

**SC1-7a.** Figure 1. Water courses shown don't match up with the description. There is supposed to be one inlet and outlet to Lake A, but at least two inlets are shown (or outlets). Flow directions are needed on the other streams (canals?) shown.

**Answer: Done.** A description of the streams (or canals) in the southwestern part of Figure 1 b should be added. As the topography in the study area is nearly flat, the flow direction can evolve according to Lake DM elevation, similarly to S2. It is important to note that these streams are channelized (i.e., man-made). We propose to correct Line 91-92 to:

*"Two channelized outlet streams (S2 and S3) allow water to exit Lake A and flow towards Lake DM. The flow direction at S2 and S3 can be temporally reversed. . ."*

**SC1-7b.** The reorienting of the North arrow is somewhat confusing, probably needed.

**Answer: Done.** The orientation of Figure 1 b can be adjusted to have both Figure 1 b and c in the same orientation (see submitted Fig. 1 with this response).

**SC1-7c.** All sampling is done in one corner of the lake, why was this done? Presumably the lake is well mixed? Samples taken near the shore, like LB-S1 could have some evaporation signature in them. Was this accounted for?

**Answer: Clarification.** Lake A is an actively mined sand pit. Access to the Lake was limited to this area due to logistical and safety issues. In this study, the horizontal variability of the waterbody was not assessed. However, Pazouki et al. (2016) showed that temperature, dissolved oxygen, pH and turbidity are very similar along four 10-meter vertical profiles (one in each corner) in Lake A. We suggest the following addition to Line 151-152:

> *"Additional field campaigns were conducted on February 9, 2017, August 17,2017 and January 25, 2018 in order to perform vertical profile measurements and water sampling at various depths (e.g. 2 m, 4 m, 8 m, 12 m and 15 m) at LA-P1 to LA-P2. Lake water sampling was only performed in the northern region of the lake due to accessibility and logistics procedures. As horizontal homogeneity has been previously demonstrated by Pazouki et al. (2016), the water samples were deemed representative of the whole waterbody."*

**SC1-7d.** If Lake DM really has a name (Deux-Montagnes), then the name should be put on the map with the (DM) in parentheses. One might also argue to use the French for the whole name, Lac des Deux Montagnes. Lake B also appears to be called Lac Val des Sables in google earth, is this not correct?

**Answer: Moot.** The Reviewer is correct; Lake B is called "Lac Val-des-Sables". The names "Lake A", "Lake B" and "Lake DM" were used to keep the location of the study site as anonymous as possible, due to an agreement with the Town. However, this comment clearly shows that it is easy to figure it out. Still, we prefer to use the names "Lake A", "Lake B" and "Lake DM" to facilitate the reading.

Considering all the above, a revised version of Figure 1 is proposed (see submitted Fig. 1 with this response).

**SC1-8.** Line 116: "All water levels are reported relative to a reference water levels measured on February 9, 2017" One reference water level or many? Please fix, this is a combination of both.

**Answer: Done.** There is a reference water level (z=0) for each well starting on February 9, 2017. Considering this comment and SC1-10, we propose a revised version of Figure 2 (see submitted Fig. 2 with this response).

**SC1-9.** Line 118 change to "...over the Ottawa River watershed..."

**Answer: Done.** Thank you.

**SC1-10.** Line 125-126: "...synchronous with those of Lake DM (Fig. 2) from late February to late July 2017". How do you know water levels are synchronous from Late February when water level measurements weren't begun until April? This can't be known. Given the sparse data in figure 2, and the non-synchronous relation between the observation well and Lake DM in the autumn, this can't be conclusively known. In addition, some of the well peaks appear to actually occur before the lake level rises, which is a bit strange. In any case, more information is needed to be able to say this. It may be true for the flood period, but that would be expected. The low flow period doesn't appear to be completely synchronous. While it may be true that Lake DM controls Lake A water level during flood periods and/or high water periods, there is no data presented that shows that Lake A water levels are synchronous with Lake DM during low flow or low water levels. Clearly the groundwater is not synchronous during September to November.

**Answer: Clarification and done.** A hydraulic connection between Lake DM and Lake A only occurs when a topographic threshold (at 22.12 m.a.s.l.) is exceeded. During autumn, the water levels of Lake DM and VP are below this threshold, which

explains why there is a different evolution for Lake DM and VP during this period. This information is not explicitly shown on the original version of Figure 2. Considering this comment and SC1-8, we propose to modify Figure 2 (see response to comment SC1-8) in which the water level is illustrated relatively to a known common datum (i.e., meters above sea level). By doing so, it is possible to graphically represent the periods of hydraulic connection between Lake A and Lake DM. In our opinion, this will facilitate the reading and help understanding of the hydrodynamic context. Furthermore, we added four manual measurements of the water level at Lake A. These observations confirm that the water level of Lake DM is not governing the one of Lake A during autumn (September to November).

Note that the absolute water level at the pumping well P5 is not known (missing information concerning the positioning of the Town's pressure logger). The water level at P5 could thus not be represented on the revised version of Figure 2.

**SC1-11.** Section 3.1 Field measurements section. There is no mention of calibration for the water level loggers. Without calibration how do you know they were synchronous or that they water levels were the same? Please give all the calibrations that were done on instruments and isotopic analyses.

**Answer: Clarification.** The following could complement Line 140:

> *"All the level loggers' clocks were synchronized with the computer's clock when launching automatic measurements for a 3-month period. This procedure was done via the Diver-Office 2018.2 software. Manual measurements of the water level were regularly performed to calibrate (relatively to a reference datum) and validate the automatic water level measurements."*

**SC1-12a.** Section 3.2 Water sampling and analytical techniques: Other than ion

balances, was any other QA/QC done? This needs to be stated. In addition, were isotope measurements compared with standard mass spectroscopy? Ring cavity measurements have been shown to be in error in some cases and should be viewed with some skepticism unless comparison is made to standard mass spectroscopy or the methodologies listed below have been followed. See Wassaner et al (2014) and Sengupta (2014) for examples. Perhaps more detail about replicates and comparisons to mass spectroscopy measurements can be done to alleviate concerns over the accuracy of the ring cavity measurements. References: Sengupta, S., 2014, Pros and Cons of Laser Based Isotope Measurements of Water and Real Time Vapour Samples: A User's Perspective. Gond. Geol. Mag., V. 29 (1 and 2), pp.45-51 Wassenaar, L.L., Coplen T.B., Aggarwal, P.K., 2014 Approaches for Achieving Long-Term Accuracy and Precision of 18O and 2H for Waters Analyzed using Laser Absorption Spectrometers. Environ. Sci. Technol. 2014, 48, 2, 1123-1131.

**Answer: Clarification and done.** We agree that it is worth to provide more details on the analytical procedures. Thank you for questioning this point. Line 166-168 could be corrected to:

*"1 ml of water was pipetted in a 2 ml vial and closed with a septum cap. Each sample was injected (1 microliter) and measured 10 times. The first 2 injections of each sample were rejected to limit memory effects. Three internal reference waters ($\delta^{18}O = 0.23 \pm 0.06$‰, $-13.74 \pm 0.07$‰ $-20.35 \pm 0.10$‰; $\delta^2H = 1.28 \pm 0.27$‰, $-98.89 \pm 1.12$‰ $-155.66 \pm 0.69$‰; $\delta^{17}O = 0.03 \pm 0.04$‰, $-7.32 \pm 0.06$‰ $-10.80 \pm 0.06$‰) were used to normalize the results on the VSMOW-SLAP scale. A $4^{th}$ reference water ($\delta^{18}O = -4.31 \pm 0.08$‰; $\delta^2H = -25.19 \pm 0.83$‰; $\delta^{17}O = 2.31 \pm 0.04$‰) was analyzed as an unknown to assess the exactness of the normalization. The overall analytical uncertainty ($1\sigma$) is better than $\pm0.1$‰ for $\delta^{18}O$, $\pm1.0$‰ for $\delta^2H$ and $\pm0.1$‰ for $\delta^{17}O$. This uncertainty is based on the long-term measurement of the $4^{th}$ reference water and does not*

*include the homogeneity nor the representativity of the sample (Light
stable isotope geochemistry laboratory of Geotop-Uqam)."*

**SC1-12b.** Also, were any samples taken under ice? Ice will fractionate the isotopic
composition and make the mass balance different. Has this been accounted for?

**Answer: Clarification.** We opted to neglect the ice fractionation in the isotopic model,
because assumptions concerning the forming (or melting) rate and isotopic signature
of the ice would have been needed. To support this assumption, we calculated the
isotopic signature of the residual water (i.e., lake water), which is described by:

$$\delta \approx \delta_0 + \epsilon * ln(f_{residual})$$

where $\delta_0$ is the initial water isotopic signature, $\epsilon$ is the ice-water isotopic separation
factor and fresidual is the residual water fraction.

If the ice thickness is 0.35 m (observed on March 4, 2019) over a surface of
$2.79x10^5 m^2$ and the lake volume is $4.7x10^6 m^3$, the ice volume would be $8.7x10^4 m^3$
and f would be 0.98. Considering an ice-water isotopic separation factor ($\epsilon$) of 3.1 for
$\delta^{18}O$ and 19.3 for $\delta^2 H$ (O'Neil, 1968) and a $\delta_0$ of $-10‰$ for $\delta^{18}O$ and $-71‰$ for $\delta^2 H$,
the isotopic signature of the residual water ($\delta$) would be $-10.06‰$ for $\delta^{18}O$ and
$-71.39‰$ for $\delta^2 H$. Such variation falls within the analytical uncertainty (i.e., $\pm0.1‰$ for
$\delta^{18}O$, $\pm1.0‰$ for $\delta^2 H$), as shown in the Fig. 3 submitted with this response. Note that
a well-mixed lake is assumed for this calculation.

**SC1-12c.** It also isn't clear why methods are included for water quality sampling.
These data don't appear to have been used in this manuscript, so it simply takes up
space. Please remove the methods for chemical sampling and concentrate only on
the isotopic measurements.

**Answer: Clarification.** It was deemed preferable not to discuss the physico-chemical parameters and the major ion data. This decision was made in order to limit the manuscript length. However, we agree that section "5.3 Resilience to surface water pollution" could benefit from an additional discussion concerning the geochemical data. We suggest that the geochemical signature of Lake A is compared to the one of Lake DM, Lake B and regional groundwater (i.e., observed at piezometers upstream of Lake B). Below is the proposed additional figure (Fig. 4 submitted with this response) and interpretation to add to section "4.1 Water fluxes and isotopic framework":

> *"Water samples were additionally collected at the surface and at various depths within Lake A (n = 23) and at Lake DM (n = 1) during the study period and analyzed for major ions, as detailed in section 3.2. The geochemical facies of Lake A and Lake DM samples are illustrated in Figure 5 by the means of a Piper diagram. Both Lake A and flood-water were found to be Ca-HCO3 types, which is typical for precipitation- and snowmelt-dominated waters (Clark, 2015). The geochemistry of Lake A is relatively constant throughout the year and confirms depth-wise homogeneity. Water samples were also collected for comparison at the surface and depth of Lake B (n = 42) and at observation well Z16 (n = 11), which is upstream of Lake B and, thus, representative of the local groundwater contributing to the latter (Ageos, 2016). The geochemistry of Lake B is significantly distinct from Lake A and appears to be influenced by regional groundwater of Na-Cl water type."*

Also, we propose to add the following material to section "5.3 Resilience of Lake A to surface water pollution" (at L509):

> *"Considering the Ca-HCO3 water type of Lake A (see section 4.1) and the importance of the flood-water inputs to the annual lake water budget (see*

*sections 5.1 and 5.2), the flood-water contribution is likely to dampen intra-annual geochemical variations at Lake A and is controlling the low-mineralization levels compared to the neighboring lake (i.e., Lake B). If a reduction in the flood-water inputs was to occur, the geochemistry of Lake A could potentially shift towards that of Lake B. In such a case, an increase of the salinity and in the concentration of $Na^+$, $Ca^{2+}$, $SO_4^{2+}$ and $Cl^-$ would be expected for Lake A.*

Note that this issue was also highlighted by Reviewer 2 (see RC2-6a).

**SC1-13.** Line 170: "The water and stable isotope mass balance of a well-mixed lake can be described. . ." The authors haven't actually demonstrated that the lake is well mixed. A figure showing the lake profiles should be presented.

**Answer: Clarification.** In a recent study, Arnoux et al. (2017) compared a well-mixed model and a depth-resolved multi-layer model. Both models yielded similar results and provided a general understanding of the groundwater-surface water interactions. The multi-layer model additionally allowed for the determination of groundwater flow with depth, which is beyond the scope of our study.

The advantage of developing well-mixed models lies in their simplicity. Another motivation for the development of a well-mixed model was to be able to compare our results to the Canadian context. In fact, numerous isotopic mass balance models for lakes have been developed across the western provinces (i.e., British-Columbia and Alberta) assuming well-mixed conditions, but only few studies have been carried in the eastern part of the country. Hence, our study is an addition to the actual knowledge in Canada and can be useful in providing data to address pan-Canadian perspectives.

It is worth noting that we developed an isotopic mass balance model assuming well-mixed conditions, but we did take into account the vertical variability of the isotopic signature by fitting the simulated $\delta_L$ on the depth-average observed $\delta_L$.

Although this method does not perfectly reproduce the reality, it is nevertheless more accurate than a model calibrated on the observed isotopic composition at the surface of the lake only.

In a recent paper, Gibson et al. (2017) studied the impact of sampling strategies on the water yield estimations for the Turkey Lake (32 m deep) under stratified and well-mixed conditions. They reported 18% difference on the water yield when performing grab sampling (i.e., 1 sample at 1 m depth) and bulk sampling (i.e., assessment of the whole lake water column). The difference was less important (i.e., 11%) when comparing bulk sampling to integrated sampling for epilimnion, metalimnion and hypolimnion. They also reported discrepancies up to 20% for the water yield at the same lake according to the timing of the lake water sampling. This last result shows that temporal shifts may induce more important bias than the uncertainty related to the lake stratification.

Considering the above, we conceive that the developed model yields a valuable estimation of Lake A water balance, because we did consider the lake stratification by using a depth-average isotopic signature to represent $\delta_L$.

Note that the stratification of the lake is also raised in comments SC1-7c, SC1-17, RC1-1b.

**SC1-14.** Line 198: So, evaporation was held constant for the entire month. Particularly in the spring, that is a brave assumption. This seems to be the coarsest time step. Why was this needed?

**Answer: Clarification.** In the model, the evaporation is specified at a daily time step. At Line 198, we are referring to the input parameters (i.e,. $\delta_P$, $\epsilon^+$, $\alpha^+$ and $\epsilon_K$) for the calculation of the atmospheric moisture ($\delta$A). In this calculation, the $\delta_P$, $\epsilon^+$, $\alpha^+$ and $\epsilon_K$ are evaporation flux-weighted. Given daily evaporation rate time series, $\delta_A$ can only be estimated at a monthly time step. For more details, see Gibson et al. (2015).

**SC1-15.** Line 208: limiting isotopic composition (Gibson et al., 2015). This is not a common term. Although this can be found in the reference listed, it should be detailed more here.

**Answer: Done.** Gibson et al. (2015) states that "$\delta^*$ is the isotopic composition that a desiccating water body would approach under non-steady-state conditions as it dries up (i.e. $V \to 0$)."

Line 208 could be corrected to:

> "...$\delta^*$ is the isotopic composition that the lake would approach as $V \to 0$ (Gibson et al., 2015)."

However, this concept is referring to a boundary condition (mathematically). It could be more appropriate to define $\delta^*$ as: "the limiting isotopic condition (i.e., the boundary condition as $V \to 0$)".

**SC1-16.** Line 216: "The above-mentioned equations are computed on a daily time step to calculate the isotopic composition of the lake (dL)." Yet, some parameters have monthly time steps. How do you reconcile that? Does this mean the monthly time steps aren't that important, or should it all be done monthly? This seems like a limitation to the daily time step.

**Answer: Clarification.** Please see response to comment SC1-14.

**SC1-17.** Line 218: It has been stated a few times that the lake is well mixed, but this has not been demonstrated with any measurements. The reader needs evidence that the lake is well mixed, particularly over the time period of measurement, which is over the springtime period, when mixing may not be complete.

**Answer: Clarification.** Please see response to comment SC1-13.

**SC1-18.** Line 223: "Assuming homogenous hydraulic conductivity of the sediments" This is a big assumption and likely not accurate overall, but in a sandy aquifer, might be a reasonable assumption given other errors in the model. This should be explained more.

**Answer: Clarification.** Thank you for questioning this point. As you pointed out, this assumption might be a reasonable in the context of this study, but care should be taken when stating so. Your comment made us reanalyze our reasoning leading to this assumption. We did not state the correct hypothesis. In the context of our study, we should simply hypothesize the following:

*"The outflow from the lake are roughly proportional to the lake water level."*

**SC1-19.** Line 259 and 263: This is a pet peeve of mine, but "since' is a time word and shouldn't be used to replace "because", please change to "because" everywhere in the manuscript when it is not used as a temporal term.

**Answer: Done.** Thank you.

**SC1-20.** Line 269: change to "...lead to overestimation. . .:"

**Answer: Done.** Thank you.

**SC1-21.** Line 270, so is the potential underestimation of groundwater exchange underestimated here? Or was something done to account for this. Please explain.

**Answer: Clarification.** A sensitivity analysis was performed over the evaporative fluxes (E) to address this point specifically (i.e., the overestimation of E, leading to a potential underestimation of the groundwater exchange). When considering E $-20\%$, the model yields to total annual outputs of $1.44x10^7 m^3$, while the reference scenario yields $1.72x10^7 m^3$. This translates to a $16\%$ decrease of the total annual outputs (and

inputs). The evaporation represents roughly $2\%$ of the total outputs in both scenarios. The remaining $98\%$ corresponds to the total groundwater and surface water outputs (Q). The partition of groundwater outflow and surface water outflow cannot be determined (see Line 219-220), as their isotopic signatures are conceptually the same (i.e., $\delta_L$). Also, when considering E $-20\%$, the model yields, the surface water inputs (Is) correspond to $68\%$, which remains very similar to the partition of the reference scenario (i.e., $71\%$). While E was found to be one of the most stringent parameters, the water balance partition remains similar for both scenarios over an annual basis. Hence, an overestimation of E is misleading.

We could refer the reader to the sensitivity analysis at Line 270.

Does this answer your question? If not, can you please reformulate your question?

**SC1-22a.** Line 290-295: Why not just measure the GW input? Why does it need to be estimated from the intersection with the LEL?

**Answer: Clarification.** In hydrogeological contexts where groundwater-surface water interactions are important, the chemistry and isotopic signature of groundwater typically bear some local heterogeneity. Hence, the representativity from a groundwater sample can be hard to understand. In the context of this study, it was preferable to estimate $\delta$L from the intersection between the lake's LEL and the LMWL, as it represents the mean isotopic signature of the local groundwater contributing to the lake.

For further details, please see response to comment RC1-2.

**SC1-22b.** Also, although the evaporation process is the same between flood water and lake water (having the same slope) that is not unusual. What is unusual is that they don't intersect at the same place, so the floodwater is a different source from the recharge from GW or rainfall.

**Answer: Clarification.** The flood-water and groundwater have different sources. The flood-water is mainly composed of springtime rainwater and snowmelt water and is originating from a large watershed which extends to the North. The isotopic signature of the flood-water was thus expected to be more depleted relatively to the groundwater (see Line 286-288). The local groundwater is conceptualized as a mixture between local precipitations and flood-water (due to the yearly recurrent flooding of the study area).

**SC1-22c.** There do appear to be five lake values (one of which appears to be unevaporated floodwater) that fall on the floodwater line, so there is some influence from floodwater on the isotopic composition of the lake. This should be address more fully.

**Answer: Clarification.** Indeed, some surface water samples from Lake A are plotting near the flood-water LEL and one might hypothesize that this is suggesting an influence of the flood-water on the isotopic composition of the lake. However, of the five Lake A ($\leq 2$ m depth) water samples that appear to plot near the flood-water LEL, only the most depleted sample was collected during the flooding event in 2017. The other samples were taken in April 2016, June 2016, December 2017 and January 2018. Considering the timing of the sampling dates, these four surface water samples are more likely suggesting mixing between lake water and precipitations.

**SC1-23a.** Line 331-332: The authors say: "Lake A volume variations are estimated from water level records assuming a constant lake area. When not available, the surface elevation of Lake A is assumed to be equal to the water level at other observation points." I don't understand what this means. Unless this is a pit lake with perfectly straight vertical sides, the Lake area will increase as elevation increases and it will take more water for fill shallower stage heights as the lake gets bigger. Please explain if this is not true for this lake.

**Answer: Clarification.** The Reviewer is correct. Net water fluxes were calculated

considering constant area (i.e., perfectly vertical banks). This assumption was made because of the relatively flat topography (outside of the lake's banks). Attempting to delineate the lake's contour with a Digital Elevation Model (DEM) would have led to unrealistic results. Therefore, we opted to neglect the surface variations. This assumption is not likely to have a significant impact on the model outputs. In fact, the lake water level variations extend over a 2.9 m range only (from 21.87 m.a.s.l. to 24.77 m.a.s.l.), which is relatively small compared to the maximum depth. Hence, the calculated lake volumes are very similar when considering $25°$ slopes or $90°$ slopes over the range of water level variations (see Fig. 5 submitted with this response). However, for the calculation of the isotopic signature of the lake (i.e., $\delta_L$), assuming vertical banks would have led to less representative values. We assumed $25°$ slopes, to calculate a depth-average $\delta_L$. In this case, the lower depths have less impact than the shallower parts of the lake on the estimation of $\delta_L$.

**SC1-23b.** Furthermore, water levels in a well cannot be used unless there is no GW flow to the lake. If the groundwater level is the same as the lake level, then there will be no flow to the lake and the flow is stagnant. Has this been observed? If not, this GW elevation should not be used as a surrogate for lake level.

**Answer: Moot.** Please see response to comment SC1-24.

**SC1-24.** Figure 2 actually show that Lake A water level is at no time equal to Observation well VP, and is generally higher than the well elevation, except in late summer, suggesting the lake is losing water to the well except when precipitation slows down and the lake level lowers. Lake DM, which is a possible surrogate for Lake A elevation, is also never equal to the elevation of well VP, except on the rising limb of the floodwater. Therefore, the well VP elevation is not a good surrogate for lake A elevation and should not be used as such, unless a better explanation can be given.

**Answer: Moot.** The Reviewer is correct, there is groundwater flow between Lake A and VP. We know that the pumping wells induce a hydraulic gradient, which forces

Lake A water to infiltrate the sandy bank (year-round). However, the isotopic composition of the lake $\delta L$ is iteratively solved at each time step and is dependent on f, which is the remaining fraction of lake water. The model is thus based on the water level difference between two time steps (not the absolute water level). From August to November, the daily water level variations at VP are expected to be of the same range as the ones of Lake A. Moreover, the water level of VP is a better approximation than Lake DM during the period of no hydraulic connection (i.e., from August to November). Considering the above, the observed water level at VP can be used as a surrogate for Lake A from August to November 2017.

This issue was also addressed in response to comment RC1-8.

**SC1-25.** Line 338-340: This also a time of groundwater input (at least following the Lake DM elevation compared to Well VP). Is this considered in the fluxes?

**Answer: Clarification.** At Line 338-340, we refer to the net water fluxes, which include all inputs and outputs. During the flood period (i.e., February 23, 2017 to May 8, 2017), the high water level at Lake A was very likely to impose a hydraulic gradient towards the aquifer, which led to very limited contribution of groundwater to the lake's water balance in comparison to the floodwater inputs (from Lake DM). Hence, we developed the water balance model assuming that the groundwater inputs were null during the flooding period. Contrastingly, surface water inputs were neglected for the rest of the simulated period, while groundwater inputs were expected to play a major role in the water balance partition.

**SC1-26.** Line 359: So, here the vertical profiles are volume-weighted, which suggests the sides of the lake are not vertical, if they were then you wouldn't need to volume-weight them. But above you say you use a constant lake area to get the volume. Which is it?

**Answer: Clarification.** Please see response to comment SC1-23a.

**SC1-27a.** Line 382-384: you do have 3 vertical profiles; you could have at least estimated how big a difference using a stratified model using some max and min values for the isotopes.

**Answer: Clarification.** The development of a multi-layer model was beyond the scope of this study (see response to comment SC1-17). Although our model is a simplification of the reality, it provides a reasonable estimation of the water balance partition of Lake A and allows to discuss the impact of future changes.

**SC1-27b.** It also isn't clear from the discussion above this if the direction of groundwater low, in or out of the lake is considered, as the water level data suggests in changes through the modeling period.

**Answer: Clarification.** Please see comment SC1-25.

**SC1-28a.** Table 2: A small point, but I'm not sure why commas are used in this table. Scientific notation usually uses a period even for large numbers. Europeans use commas for decimals and then periods for large numbers, so I'm not sure what style is being used here. I would prefer these to all be periods not commas.

**Answer: Done.** Indeed, all the commas are to be replaced by periods.

**SC1-28b.** A larger point for this table is that the sensitivity analysis doesn't appear to use very wide values to check how sensitive the variables are. A change of 0.5 per mil for oxygen is not that far outside the error of the measurement. It looks like most of the differences looked at are between 10 and 20 percent. Is that reasonable, what is the variability of the rainfall amounts over time. Granted E isn't likely to have a large range, but some of the variable could have larger ranges than are estimated here.

**Answer: Clarification.** Sensitivity analysis aims at identifying the input parameters that most affect the robustness of a model and can help in the model parameterization, calibration, optimization, and uncertainty quantification (Song et al.,

2015). Depending on the complexity of the hydrological model and the authors' objectives, different methods can be employed. In this study, a one-at-a-time (OAT) sensitivity analysis was performed to grasp the relative impact of the input parameter's uncertainties on the model outputs. In order words, our objective was to assess the reliability of the model outputs against a range of possible input values. As the model outputs remained comparable to the reference scenario, we concluded that the model was representative of the local hydrological processes. The selected range of input variables was carefully chosen. Concerning the isotopic framework, a change of $\pm 0.5$‰ for $\delta^{18}O$ was considered adequate to depict the potential bias introduced by sampling and analytical methods. Note that the overall analytical uncertainty $(1\sigma)$ is $\pm 0.1$‰ for $\delta^{18}O$. For the meteorological parameters, a range of $\pm 10\%$ was selected to represent the potential spatial variability (not temporal variability), as the data was retrieved from off-site meteorological stations. Furthermore, a range of $\pm 20\%$ for the evaporative fluxes (E) was deemed necessary because it was calculated from a selected evaporation model (i.e., Penman-48 equation), which is dependent on numerous meteorological parameters. A comparison with two other evaporation models (i.e., Linacre-OW and open-water simplified version of Penman-48) revealed adequation between the estimations from April to August, but discrepancies during late summer and autumn (see Line 250-258).

**SC1-29.** Line 414: What about groundwater influx at this time? Ok, I see discussed in the next section.

**Answer: Done.** Ok.

**SC1-30.** Line 440: Table 3 provides the relative importance of the hydrological processes for that year that was measured, not for an annual timescale. Measurements for all parameters weren't done for the whole year as well. This should be modified.

**Answer: Done.** Thank you.

**SC1-31.** Line 485: tG the mean flushing time by groundwater isn't included in equation 13 and is instead written as tf, which I assume is the time of flushing (by groundwater). This needs to either be explained better, if I don't understand this, or the notation needs to be corrected. Everywhere else it is tf.

**Answer: Done.** Indeed, it needs to be corrected to tf (the mean flushing time by groundwater).

**SC1-32.** Figure 9. The caption also has reference to tG is this a different variable or is it tf?

**Answer: Done.** It is $t_f$. Thank you for pointing that out.

**SC1-33a.** The climate change part of this paper is somewhat of a throw away suggestion. There is really no data or simulations that support either conclusion and the modeling doesn't appear to help either. Given the possibility of either more or less flooding the conclusions seem pretty obvious.

**Answer: Done.** We propose to add the following material at L500: "Considering the above, it is possible to speculate about the potential future impacts of climate change on Lake A."

Additionally, it is possible to add a discussion concerning the impact of the flood-water inputs on the geochemistry of Lake A (see response to comment SC1-12c).

**SC1-33b.** While the model and the system are relatively well characterized it isn't clear what this gives other scientists other than a look at a local system. How can this be used in other lake systems and can a lake with fewer measurements or larger area or volume be characterized using this model? It would be good if some bigger questions were answered rather than just the local questions that have no real interest to scientists or the public outside of the area.

**Answer: Done and clarifications.** First, we are grateful to Reviewer 2 for this

valuable comment. First, the following material could be added to the introduction at L43 to better highlight the broad relevent of such study:

*"Knowledge concerning the surface water-groundwater interactions in flooded environments are of global importance, as there is an increasing interest worldwide for undamming rivers and floodplain restoration as management tools for flood risk and/or enhancing ecosystem services (Dixon et al., 2016)."*

Another original aspect of our study is related to the approach, i.e., the use of isotopic tools to cope with the impossibility (or difficulty) to perform direct measurements of surficial water fluxes. The following material could be added to the introduction at L51:

*"Furthermore, most isotopic mass balance models are applied to contexts where there are no surface water inputs and/or the surface water inputs are quantified by direct measurements (i.e., river stage). However, in floodplains, direct measurement of surface water inflow is difficult (or nearly impossible), while isotopic mass balance models can be useful in providing estimation of the water balance partition with a minimum sampling and monitoring effort."*

Concerning the applicability of the method to other environments, please see response to RC2-2.

Besides, we think it is relevant to add the following material to the introduction at L58 to better explain the outcomes of this study:

*"The study period spanned a 100-year flood, an event which may no longer occur on dammed rivers (Bednarek, 2001). The present case study is thus*

*a precursor to future research on the impact of restoring the river corridors, as it provides an example of the importance of flood-water inputs on the water balance of a lake in an urban area and during a 100-year flood. This study also provides insights on the usefulness of isotopic data from lakes to complement the hydrogeomorphic methodology for the delineation of the flooding space of rivers. This concept is an important aspect of the recently developed methodology for determining the freedom space for river by Biron et al. (2014) and which was proven to be an economically viable river management approach (Buffin-Bélanger et al., 2015)."*

**References**

Ageos: Drinking water supply: Monitoring of piezometric fluctuations in the water table and lake levels: Period from April 27, 2012 to December 17, 2015: Annual Report 2015, AGEOS, Brossard, QC, Canada, 42, 2016.

Arnoux, M., Gibert-Brunet, E., Barbecot, F., Guillon, S., Gibson, J., and Noret, A.: Interactions between groundwater and seasonally ice-covered lakes: Using water stable isotopes and radon-222 multilayer mass balance models, Hydrological Processes, 31, 2566-2581, 10.1002/hyp.11206, 2017.

Bednarek, A. T.: Undamming Rivers: A Review of the Ecological Impacts of Dam Removal, Environmental Management, 27, 803-814, 10.1007/s002670010189, 2001.

Biron, P. M., Buffin-Bélanger, T., Larocque, M., Choné, G., Cloutier, C.-A., Ouellet, M.-A., Demers, S., Olsen, T., Desjarlais, C., and Eyquem, J.: Freedom Space for Rivers: A Sustainable Management Approach to Enhance River Resilience, Environmental Management, 54, 1056-1073, 10.1007/s00267-014-0366-z, 2014.

Buffin-Bélanger, T., Biron, P. M., Larocque, M., Demers, S., Olsen, T., Choné, G., Ouellet, M.-A., Cloutier, C.-A., Desjarlais, C., and Eyquem, J.: Freedom space for rivers: An economically viable river management concept in a changing climate,

Geomorphology, 251, 137-148, 10.1016/j.geomorph.2015.05.013, 2015.

Clark, I.: Groundwater Geochemistry and Isotopes, Boca Raton, FL, 2015.

Dixon, S. J., Sear, D. A., Odoni, N. A., Sykes, T., and Lane, S. N.: The effects of river restoration on catchment scale flood risk and flood hydrology, Earth Surface Processes and Landforms, 41, 997-1008, 10.1002/esp.3919, 2016.

Gibson, J. J., Birks, S. J., and Yi, Y.: Stable isotope mass balance of lakes: a contemporary perspective, Quaternary Science Reviews, 131, 316-328, 10.1016/j.quascirev.2015.04.013, 2015.

Gibson, J. J., Birks, S. J., Jeffries, D., and Yi, Y.: Regional trends in evaporation loss and water yield based on stable isotope mass balance of lakes: The Ontario Precambrian Shield surveys, Journal of Hydrology, 544, 500-510, 10.1016/j.jhydrol.2016.11.016, 2017.

Herczeg, A. L., Leaney, F. W., Dighton, J. C., Lamontagne, S., Schiff, S. L., Telfer, A. L., and English, M. C.: A modern isotope record of changes in water and carbon budgets in a groundwater-fed lake: Blue Lake, South Australia, Limnology and Oceanography, 48, 2093-2105, 10.4319/lo.2003.48.6.2093, 2003.

Lamontagne, S., and Herczeg, A. L.: Consultancy report: Predicted trends for NO3-concentration in the Blue Lake, South Australia, CSIRO Land and Water, 2002.

O'Neil, J. R.: Hydrogen and oxygen isotope fractionation between ice and water, The Journal of Physical Chemistry, 72, 3683-3684, 10.1021/j100856a060, 1968.

Pazouki, P., Prevost, M., McQuaid, N., Barbeau, B., de Boutray, M. L., Zamyadi, A., and Dorner, S.: Breakthrough of cyanobacteria in bank filtration, Water Res, 102, 170-179, 10.1016/j.watres.2016.06.037, 2016.

Rosen, M. R.: The Influence of Hydrology on Lacustrine Sediment Contaminant Records, in: Environmental Contaminants: Using natural archives to track sources

and long-term trends of pollution, edited by: Blais, J. M., Rosen, M. R., and Smol, J. P., Springer Netherlands, Dordrecht, 5-33, 2015.

Song, X., Zhang, J., Zhan, C., Xuan, Y., Ye, M., and Xu, C.: Global sensitivity analysis in hydrological modeling: Review of concepts, methods, theoretical framework, and applications, Journal of Hydrology, 523, 739-757, 10.1016/j.jhydrol.2015.02.013, 2015.
* * *
[Figure]

**Fig. 1.** Revised version of Figure 1.

[Figure]

**Fig. 2.** Revised version of Figure 2.

[Figure]

**Fig. 3.** Isotopic fractionnation due to ice-forming in a lake (solid black line) and the theoretical isotopic signature of the residual water in Lake A (blue circle).

[Figure]

**Fig. 4.** Geochemical signature of Lake A, Lake B, flood-water and the regional groundwater (GW).

[Figure]

Fig. 5. Relationship between the lake volume and the lake water level for bank slopes of 25 degree and 90 degree.

---

## Author Comment (AC2) · 22 Jul 2020

article [utf8]inputenc

graphicx xcolor ragged2e textcomp

On one hand, I consider this study as a stimulating contribution to the ongoing effort of improving the reliability of isotope-based modeling for the prediction of lakes hydrological behavior. Based on an attractive, although complex, case study, it presents a commendable attempt to assess quantitatively the sensitivity of the model results to the variations of the parameters. On the other hand, reading the manuscript

is a bit frustrating due to a number of weaknesses in the data set and hypotheses, and also in the description of the model algorithm.

**RC1-1a.** As for the data, the confidence that we can have on the robustness of the authors' conclusions clearly suffers from the lack of continuity of the lake level monitoring, and from the lack of documentation of the lake stratification. Due to the ice cover (line 124) and logger failure (line 140), the level of the lake is available only for a part of the flooding period. In particular, we miss the comparison between lake A and lake DM for summer and fall 2017, when lake DM was seemingly above the threshold, and should thus have overflown again into lake A (Figure 2).

**Answer: Clarification.** It is true that the data is limited, but we believe that it is sufficient and reliable to perform an isotopic mass balance model. We did not initially orchestrate the sampling campaigns and monitoring program to perform such study. However, a 100-year flood occurred during springtime 2017 and we took advantage of the importance of this hydrological event to assess the partition of the water mass balance under extreme conditions.

Considering Quebec's meteorological context, surface water level monitoring and surface water sampling can be hard to achieve during specific periods of the year. For instance, during springtime, it can be dangerous to perform in-lake water sampling due to ice-cover melting and flooding. Moreover, access to the study site was limited during the peak of the flood event. Ice-cover also constitutes a challenge for the lake water level measurements. Although attempts were made at installing loggers throughout late autumn and wintertime, drifting ice can damage the equipment and lead to its loss. Still, near shore lake water samples could have been collected in early May and provide valuable data for the characterization of the evolution of the isotopic signature of the lake. Also, punctual measurements of Lake A water level have been carried throughout the study period and provide complementary data to understand the evolution of the water levels (see submitted Fig. 1 with this response).

**RC1-1b.** More importantly, the lack of temperature and isotope data during the peak of the flooding period, and around the fall overturning, casts a strong doubt on the justification of the assumption of lake homogeneity. This weakness is acknowledged by the authors at several places (line 260, 269-270, 311, 433, 519). However, the reader is left with the impression that a two-layer model would definitely be needed, and that the model results might be much more sensitive to the seasonal stratification than to the other parameters tested in the study.

**Answer: Clarification.** In a recent paper, Gibson et al. (2017) studied the impact of sampling strategies on the water yield (or depth-equivalent run-off) estimations for the Turkey Lake (32 m deep), Ontario (Canada) under stratified and well-mixed conditions. They reported $18\%$ difference on the water yield when performing grab sampling (i.e., 1 sample at 1 m depth) and bulk sampling (i.e., assessment of the whole lake water column). The difference was less important (i.e., $11\%$) when comparing bulk sampling to integrated sampling for epilimnion, metalimnion and hypolimnion. They also reported discrepancies up to $20\%$ for the water yield at the same lake according to the timing of the lake water sampling (3 times between Oct 27 and Nov 30). This last result shows that temporal shifts may induce more important bias than the uncertainty related to the lake stratification. Considering the above, we consider that the developed model yields a valuable estimation of Lake A water balance, because we accounted for the lake stratification by using a depth-average isotopic signature to represent $\delta_L$.

**RC1-2.** Another source of worry arises from a possibly spurious choice for the isotope composition of the groundwater inflow end member. As soon as the authors show that one of the two main supplies of lake A, i.e. the flood water overflowing from lake DM, shows an isotope composition which is below the lake evaporation line, one would expect the other main source, i.e. groundwater, to lie above that line, and not on it (line 290). This needs to be thoroughly discussed, all the more as the authors emphasize that ïAËŽd'G is among the most stringent parameters. More generally,

some explanation is needed about the origin of the isotopic difference between the groundwater and flood water. Would it be possible to collect samples from the aquifer, away from the influence from the lake? How does the aquifer composition compare with the amount-weighted average of the rainfall seasonal variation?

**Answer: Clarification.** There is a scientific consensus concerning the isotopic composition of groundwater contributing to a lake (Edwards et al., 2004). The isotopic composition of the total inputs ($\delta_I$) to a lake is resulting from the mixing between the isotopic composition of groundwater inflow ($\delta_G$), surface water inflow ($\delta_{Is}$) and precipitations ($\delta_P$) and can be defined by:

$$\delta_I = (\delta_{Is} * I_s + \delta_G * I_G + \delta_P * P)/I$$

where $I_S$, $I_G$ and P are the surface water inputs, groundwater inputs and precipitations, respectively. The total inputs (I) are described as $I = I_S + I_G + P$. Conceptually, $\delta_G$, $\delta_{Is}$ and $\delta_P$ all plot along the LMWL, so does $\delta_I$ (given no influence of evaporation). Then, the position of $\delta_I$ on the LMWL is controlled by the relative proportions of $I_S$, $I_G$ and P. In general, $\delta_I$ is not significantly influenced by $\delta_P$, because P $<<$ I.

In cases where there is no surface water input (i.e., $I_s = 0$), the $\delta_I \approx \delta_G$ and the isotopic signature of the lake ($\delta_L$) will evolve along the LEL from $\delta_G$. In other words, the intersection between the LMWL and the LEL corresponds to the isotopic signature of the lake if $E = 0$ and is a good estimate of $\delta_G$, when $I_s = 0$. In the case of our study site, the yearly recurring flood events are affecting the isotopic composition of the local groundwater contributing to Lake A. In fact, the hydraulic connection between Lake DM and Lake A and the high water levels result in an enhanced springtime recharge of relatively depleted water. Hence, the isotopic composition of local groundwater was conceptualized as a mixture between flood-water and isotopic composition of regional groundwater and, therefore, more depleted than the amount-weighted average of $\delta_P$, which is $-10.2‰$ for $\delta^{18}O$ and $-68‰$ for $\delta^2H$ (calculated for the year 2016). While the

partition between $I_S$ and $I_G$ was not known a priori, it is possible to infer that the intersection between the LMWL and the LEL would correspond to the most depleted isotopic signature the local groundwater could have. Computing the isotopic mass balance model with such estimation of the $\delta_G$ is conservative because it yields to a lower limit of the estimation of groundwater exchange.

Furthermore, estimating the $\delta_G$ from the intersection between the LMWL and the LEL was considered more adequate than measuring it from groundwater samples. Indeed, in hydrogeological contexts where groundwater-surface water interactions are important, the chemistry and isotopic signature of groundwater typically bear some local heterogeneity. Hence, the representativity from a groundwater sample can be hard to understand. In the context of this study, it was preferable to estimate $\delta_L$ from the intersection between the lake's LEL and the LMWL, as it represents the mean isotopic signature of the local groundwater contributing to the lake.

Note that this issue was also addressed by M. Rosen (see response to comment SC1-22a).

**RC1-3.** On another aspect, I have some reservation about the way the model is described. A luxury of details is given on several very classical aspects already extensively described in previous works (i.e. Craig & Gordon's approach of the isotopic budgets), which could thus be better placed in appendix, while the description takes shortcuts on other key linkage in the modeling procedure. For instance, the reader should not have to wait until line 357 (results section) to learn the hypotheses leading to the outflow estimate! Another example is the emphasis put on the volume-dependent modeling (line 49 and 54, 200-210). If the authors want to convince the reader that this is important, they have to better explain equation (6) to (10), which are quite cryptic, and compare with the same equation based on a constant volume approximation. This should also be discussed when looking at the results. This whole section should be written again, as a true instruction manual for

anyone willing to apply such a model to another case-study.

**Answer: Done.** We agree that the more common aspects of the model could be easily described in the appendix and that we would benefit from highlighting the details concerning site-specific points. Hence, we propose placing the description of $\delta_E$ (L159-L198) calculations in Supplementary Materials, as well as the results of the evaporative fluxes calculation with the Penman equation (i.e., L240-L249 and L259-L270) and the comparison with two other models (i.e., L250-L258).

**RC1-4.** In general, I have the feeling that the overall structure of the manuscript is a bit messy. I would recommend giving first all the information that can be deduced from the lake level variations (i.e. line 330-349), in order to introduce properly the aims of using isotopes to better unravel the contribution of the different sources.

**Answer: Done.** We made an effort at reorganizing the sections of the manuscript to make a better use of the "4 Results" and the "5 Discussion" sections. Below is the proposed structure (key figures and tables in italic):

4 Results

4.1 Water fluxes and isotopic framework (Figure 3, Figure 4 and Table 1)

4.2 Evaluation of the water budget

4.2.1 Volume-dependent isotopic mass balance (Figure 6 and Table 3)

4.2.2 Sensitivity analysis (Table 2)

4.3 Temporal variability in the water balance partition (Figure 7)

5 Discussion

5.1 Local flood-water marked groundwater

5.2 Importance of flood-water inputs in the water balance partition

5.3 Resilience of Lake A to surface water pollution (Figure 9)
Note that the original subsection "4.2.1 Insights from net water fluxes at Lake A" is not reporting a specific objective of the study, but rather provides a "reality check" by describing the net water fluxes and Lake A volume variation. Hence, we opt to merge it with the newly proposed "4.1 Water fluxes and isotopic framework" section. That being said, we conceive that the results illustrated in Figure 5 could simply be summarized in the text, and we propose including Figure 5 in the Supplementary materials in the revised version of the manuscript. Besides, we believe that it could ease the reading of the article to use a modified version of Figure 8 (see Fig. 2 submitted with this response) as a schematic to illustrate our conceptual model and the site-specific considerations. This could be added to a new subsection, "2.3 Conceptualization of the groundwater-surface water interactions".

Specific comments, in addition to those already pointed out by the other reviewer:

**RC1-5.** Line 220-239: "Outflow fluctuations were derived from water level variations at Lake A using linear interpolation between adjusted daily minimum and maximum outflow. Daily inflow into Lake A was calculated to compensate for the adjusted outflow, as the net water fluxes are required to be equal to the lake's daily volume variation." I still do not understand what is done exactly on this key point. This needs to be written in equations and related to the main unknowns in equations (1) to (10).

**Answer: Done.** To facilitate the reading and avoid confusion, we suggest a reformulation of Line 226-229:

> *"Considering the above, it was assumed that the daily outflow flux from Lake A varied linearly according to the lake water level; the minimum and maximum outflow corresponding to the minimum and maximum water level, respectively. The outflow range (i.e., minimum and maximum values) was adjusted to obtain best fit between the calculated and observed $\delta_L$.*

*Daily inflow into Lake A compensates for the adjusted daily outflow and
daily lake volume difference."*

**RC1-6.** Line 275: "Interpolation was used to simulate the dP on a daily-time step."
This suggests that the rain data show a smooth evolution through time along the
season. Is this really the case?

**Answer: Clarification.** Reviewer 1 is correct. Sampling for precipitations was done
on a monthly time step (approximately). Therefore, we did not analyze the isotopic
signature of every single precipitation event and interpolation between the monthly
samples was necessary to compute the model at a daily time step. When computing
the evolution of the isotopic signature of the lake, precipitations are mixed
instantaneously with the whole lake volume. As the daily precipitations are much
smaller than the whole lake volume (and the other inputs), the bias caused by the
interpolation of the isotopic signature of precipitations is not expected to significantly
affect the results of the model. There would have been no gain on the accuracy of the
model in sampling precipitations at a smaller time step.

**RC1-7.** Line 284: same evaporation slope for lake waters and flood water. Is this not
surprising, as this slope depends on the climate parameters of Craig & Gordon's
equation, while flooding and evaporation do not occur at the same period of the year?

**Answer: Clarification.** The slope of the LEL is strongly influenced by the relative
humidity, and to a less extent by the temperature and the lake water balance (Gibson
et al., 2015). Given the density of surface water bodies in Canada, the relative
humidity is almost constant throughout the year and is roughly $80\%$. Hence, the LEL
slope variations are expected to be very small for a specific location.

**RC1-8.** Line 333: lake elevation assessed from well VP. Unclear what is meant by this
statement as the difference of elevation of the water level between lake A and well VP

is supposed to change with time along with the recharge/discharge alternation. (already pointed out by the other reviewer).

**Answer: Clarification.** There is in fact a water level difference between Lake A and the observation well VP. Note that the water level at VP is always lower than at Lake A, due to the pumping at the neighbouring bank filtration site.

When computing the model, the absolute lake water level is not important. The equations of the model are dependent on f, which is the remaining fraction of lake water:

$f = V/V_0$

where $V_0$ is the initial lake volume (at the beginning of the time step).

As the time series for Lake A water level was not covering the entire study period, a proxy was needed. The correlation coefficient between Lake A and VP is 0.9885 for all the available data (from 2017 to 2020, which spans both high and low water periods). Hence, while the water levels at Lake A and VP are not identical, the daily variations are expected to be similar. We thus conceive that VP is a good surrogate for Lake A.

**RC1-9.** Line 357: "the outflow fluxes are proportional to the lake's water level. We adjusted minimum and maximum outflow fluxes (Q) so that the latter respectively correspond to the minimum and maximum water levels." Again, (see comment above), I do not understand what this means.

**Answer: Done.** A reformulation of Line 226-229 was proposed. See response to comment RC1-5.

**RC1-10.** Line 368 and figure 6: The results obtained from dAЁŻdD are strictly redundant to those from dAЁŻd18O. What is really missing in this figure is some data at the beginning of May!

**Answer: Clarification.** The use of dual isotopes (i.e., $\delta^{18}O$ and $\delta^2 H$) is helpful to

perform adequate parametrization of $\delta_A$, especially in seasonal climates. However, this aspect was not one of the main goal of our study and a comprehensive study on this topic was already published by Yi et al. (2008).

Concerning the apparent lack of data in early May, we reiterate that in-lake water sampling can be dangerous to achieve in certain climates (see our response to RC1-1). In fact, ice-melting and limited access to the study site during the flood event prevented us from performing bulk water sampling along the water column. Despite these field conditions, we were able to perform near-shore lake water sampling. Although these samples are only representative of the surface-most part of the lake, they are still valuable for our understanding of the lake's dynamics. Furthermore, reference scenarios A and B were used to address this issue specifically (see comment RC1-11).

**RC1-11.** Line 434-437: scenarios A and B are supposed to compensate for the lack of data at the peak and end of the flood period. However, just mentioned like this without description, and sent back to Appendix C leaves a disastrous impression on the reader.

**Answer: Done.** We thank Reviewer 1 for this suggestion. The scenarios A and B could be both presented in Figure 6 and we would benefit from bringing the comparison between the two scenarios to the forefront.

**RC1-12**. Line 452: "The isotopic mass balance model revealed it was necessary to allow for significant groundwater outflow from Lake A during springtime to correctly reproduce the observed dL ". A best illustration of this conclusion would have been to compare the results of the model with and without the groundwater outflow.

**Answer: Moot.** From our point of view, performing a simulation without any groundwater outflow would not be representative of any realistic scenario. However, it could be interesting to quantify the lake water level increase that would be needed to

simulate the same $\delta_L$ with and without outflows from the lake. It would underline the importance of considering the groundwater-surface water interactions when studying the water balances of flood-affected lakes.

**RC1-13.** Line 485-487: confusion between tG and tf. (already pointed out by the other reviewer).

**Answer: Done.** It is $t_f$. Thank you for pointing that out.

**RC1-14.** Line 503-504: confusion on "increase" ?

**Answer: Done.** Reviewer 1 is right. It should be written "decrease".

**References**

Edwards, T. W. D., Wolfe, B. B., Gibson, J. J., and Hammarlund, D.: Use of water isotope tracers in high latitude hydrology and paleohydrology, in: Long-term environmental change in Arctic and Antarctic Lakes, developments in paleoenvironmental research, edited by: Pienitz, R., Douglas, M., and Smol, J. P., Springer, Dordrecht, Netherlands, 187-207, 2004.

Gibson, J. J., Birks, S. J., and Yi, Y.: Stable isotope mass balance of lakes: a contemporary perspective, Quaternary Science Reviews, 131, 316-328, 10.1016/j.quascirev.2015.04.013, 2015.

Gibson, J. J., Birks, S. J., Jeffries, D., and Yi, Y.: Regional trends in evaporation loss and water yield based on stable isotope mass balance of lakes: The Ontario Precambrian Shield surveys, Journal of Hydrology, 544, 500-510, 10.1016/j.jhydrol.2016.11.016, 2017.

Yi, Y., Brock, B. E., Falcone, M. D., Wolfe, B. B., and Edwards, T. W. D.: A coupled isotope tracer method to characterize input water to lakes, Journal of Hydrology, 350, 1-13, 10.1016/j.jhydrol.2007.11.008, 2008.

[Figure]

[Figure]

[Figure]

**Fig. 1.** Revised version of Figure 1.

[Figure]

[Figure]

**Fig. 2.** Original version of Figure 8.

---

## Author Comment (AC3) · 22 Jul 2020

article [utf8]inputenc

graphicx xcolor ragged2e textcomp

In this manuscript entitled "Quantifying flood-water impacts on a lake water budget via volume-dependent transient stable isotope mass balance", the authors focus on an artificial lake and justify their study by stating that "[understanding] the relative importance of the hydrological processes in lakes can also help to depict the vulnerability and/or resilience of a lake to pollution". They aim to develop a predictive

model of both atmospheric and water balance controls on isotopic enrichment, quantify of flood-water inputs to the lake, and conduct a model sensitivity analysis was conducted to evaluate potential sources of uncertainty. Overall, the manuscript is of appropriate length and well written. Figures and tables are also of good quality and rich in information without being too crowded. While I enjoyed reading this manuscript, I think that the authors need to make a strong case for the broader relevance, impact and transferability of their methods or conclusions, in addition to revisiting the structure of manuscript. My most major criticisms are as follows:

**RC2-1.** ** In its present state, the manuscript pretty much reads like a case study report. There is nothing wrong with case studies per se, as the uniqueness of place makes the conclusions of many papers inherently site-specific. That being said, I think that the authors should try to extrapolate their conclusions (or speculate about how their conclusions might extend) to other lakes (artificial or not) in Canada, North America and around the World. What makes Lake A and Lake DM different (or not) than other lakes where similar isotope mass balance approaches have been used in the past? In other words, what makes the present study novel? What are the really key contributions that represent an advancement to the science – and that may be relevant beyond the particular site that the authors focused on? Can the results be extrapolated to depressional wetlands which are affected by flooding as well? And if results and conclusions cannot be extrapolated, what about some of the methods applied in the current manuscript?

My asking those questions is not my way to say that there are no novel contributions in this manuscript, but rather to say the authors have not explicitly identified them and should highlight them better.

**Answer: Done and clarifications.** First, we are grateful to Reviewer 2 for this valuable comment. First, the following material could be added to the introduction at L43 to better highlight the broad relevant of such study:

*"Knowledge concerning the surface water-groundwater interactions in flooded environments are of global importance, as there is an increasing interest worldwide for undamming rivers and floodplain restoration as management tools for flood risk and/or enhancing ecosystem services (Dixon et al., 2016)."*

Another original aspect of our study is related to the approach, i.e., the use of isotopic tools to cope with the impossibility (or difficulty) to perform direct measurements of surficial water fluxes. The following material could be added to the introduction at L51:

*"Furthermore, most isotopic mass balance models are applied to contexts where there are no surface water inputs and/or the surface water inputs are quantified by direct measurements (i.e., river stage). However, in floodplains, direct measurement of surface water inflow is difficult (or nearly impossible), while isotopic mass balance models can be useful in providing estimation of the water balance partition with a minimum sampling and monitoring effort."*

Concerning the applicability of the method to other environments, please see response to RC2-2.

Besides, we think it is relevant to add the following material to the introduction at L58 to better explain the outcomes of this study:

*"The study period spanned a 100-year flood, an event which may no longer occur on dammed rivers (Bednarek, 2001). The present case study is thus a precursor to future research on the impact of restoring the river corridors, as it provides an example of the importance of flood-water inputs on the water balance of a lake in an urban area and during a 100-year flood. This*

[Figure]

*study also provides insights on the usefulness of isotopic data from lakes to complement the hydrogeomorphic methodology for the delineation of the flooding space of rivers. This concept is an important aspect of the recently developed methodology for determining the freedom space for river by Biron et al. (2014) and which was proven to be an economically viable river management approach (Buffin-Bélanger et al., 2015)."*

**RC2-2.** ** The introduction lacks an overarching goal or research question for the study, as well as specific research objectives or questions. Instead, the last paragraph of the intro just states that the present study builds upon two other studies. The only sentence of the introduction that could be seen as a research goal is the one that reads as: "The main purpose of this study was initially to expand our understanding of flood-affected lake dynamics in the context of a seasonal climate". It is quite vague, though, so I suggest that the authors include some more specific objectives or questions at the end of the introduction. This should also help highlight the novel contributions that the present study intends to make.

**Answer: Done.** Based on this comment and other suggestions below, we propose the following reformulation of the general and specific objectives:

General objective – The main purpose of this study is to evaluate the importance of flood-water inputs on the annual water budget of a lake located in a floodplain, in order to depict the resilience and sensitivity to changes in the water balance partition and flood-water and/or groundwater quality

Specific objective 1 – Investigate the isotopic framework for the local water cycle

Specific objective 2 – Evaluate the water budget considering reference scenarios (A and B)

Specific objective 3 – Analyze the temporal variability of the groundwater inputs and the sensitivity of the lake to flood-water driven pollution

Specific objective 4 – Demonstrate the implications of flood-water storage on the water balance partition

We could read the following in the introduction (in replacement of L52-58 and L76-78):

*"The main objective of this study is to evaluate the importance of flood-water inputs, relatively to groundwater inputs, on the annual water budget of lakes located in a floodplain, in order to depict the resilience and sensitivity of flood-affected lakes to changes in the water balance partition and flood-water and/or groundwater quality. To do so, we first aim at investigating the isotopic framework for the local water cycle to verify the applicability of an isotopic mass balance to the study site. Secondly, we quantify the water budget according to two reference scenarios (A and B) to grasp the impact of site-specific uncertainties on the computed results. Then, we analyze the temporal variability of the groundwater inputs and the sensitivity of the lake to flood-water driven pollution. Finally, we demonstrate the implications of flood-water storage on the water balance partition. The water balance is computed via a volume-dependent transient isotopic mass balance model, which is applied to predict the daily isotopic response of an artificial lake in Canada that is ephemerally connected to a $150,000 km^2$ watershed during spring freshet and other periods of flooding. During flood events, the surficial water fluxes towards the study lake are not constrained in a river or canal but occur over a 1 km wide area."*

The following material should be added to the conclusion (in replacement of L513-514), to underline the applicability of this method to other environments:

*"Flood-water originated from a large watershed, extending roughly 300 km northwards from the study site and, thus, draining snowmelt and rain*

*waters with relatively more depleted isotopic signatures than the local waters. Given the contrasted isotopic signature of the flood-water, the isotopic mass balance model was efficiently applied at the study site. Such isotopic framework is far from being unique. Indeed, most flood events occur in lowlands, while flood-waters originate from highlands (i.e., mountainous environments). Hence, our approach can be applied in many contexts, as flood-waters have a marked isotopic signature.*

**RC2-3.**** There seem to be a lot of "results" that are listed in Section 3, which most readers would equate to a Methods section (and not a results section). I would suggest that the authors try to reorganize their text a bit, so that section 3 only focuses on methods while section 4 summarizes results.

**Answer: Done.** We propose moving all section "3.3.2 Isotopic framework" to section "4 Results". Also, we think it is best to move the results of the evaporative fluxes calculation with the Penman equation (i.e., L240-L249 and L259-L270) and the comparison with two other models (i.e., L250-L258) to the Supplementary materials, as this topic is not a specific objective of our research.

**RC2-4.**** Following up on the previous point: Section 4 does not seem to focus on "plain results" only, as it includes several interpretations, discussions and linkages to the literature. Section 4 therefore reads as a combined Results and Discussion section, which is a bit surprising as there is a separate discussion section later. I suggest that the authors try to better distribute methodological aspects, results and discussions/ interpretation into distinct sections (and sub-sections).

**Answer: Done.** We made an effort at reorganizing the sections of the manuscript to make a better use of the "4 Results" and the "5 Discussion" sections. Below is the proposed structure (key figures and tables in italic):

4 Results

4.1 Water fluxes and isotopic framework (Figure 3, Figure 4 and Table 1)

4.2 Evaluation of the water budget

4.2.1 Volume-dependent isotopic mass balance (Figure 6 and Table 3)

4.2.2 Sensitivity analysis (Table 2)

4.3 Temporal variability in the water balance partition (Figure 7)

5 Discussion

5.1 Local flood-water marked groundwater

5.2 Importance of flood-water inputs in the water balance partition

5.3 Resilience of Lake A to surface water pollution (Figure 9)
Note that the original subsection "4.2.1 Insights from net water fluxes at Lake A" is not
reporting a specific objective of the study, but rather provides a "reality check" by
describing the net water fluxes and Lake A volume variation. Hence, we opt to merge
it with the newly proposed "4.1 Water fluxes and isotopic framework" section. That
being said, we conceive that the results illustrated in Figure 5 could simply be
summarized in the text, and we propose including Figure 5 in the Supplementary
materials in the revised version of the manuscript.

Also, as requested by Reviewer 1, we propose to add subsection "2.3
Conceptualization of the groundwater-surface water interactions" to illustrate our
conceptual model of the study site, which can be depicted with a slightly modified
version of Figure 8 (see comment RC1-4).

**RC2-5.**** Along the same lines as the two previous points, Section 5 is a bit confusing.
There are completely new results (e.g., Table 3, Figure 7) first reported on in this
section. Conversely, there are not a lot of literature references (none in subsection
5.1, and only 1 literature reference, as far as I can see, in subsection 5.2). So, a lot of
the text listed under the "5 - Discussion" header does not really read like a typical

discussion section, in the sense that there is very little confrontation of the present study results with the existing literature. The authors should rectify that as much as possible.

**Answer: Done.** In the proposed new structure of the manuscript, Table 3 and Figure 7 now appear in the "4 Results" section. Furthermore, to confront our results with existing literature, we suggest providing the following material for each sub-section:

**5.1 Local flood-water marked groundwater**

Comparison of the isotopic signature of groundwater ($\delta_G$) with the amount-average isotopic signature of precipitations in the south of Quebec (Canada). Such data is available from the GNIP database.

We propose to add the following material:

> *"The estimated isotopic signature of $\delta_G$ is more depleted than the St-Bruno amount weighted mean of $\delta_P$ ($-10.2$‰ for $\delta^{18}O$ and $-68$‰ for $\delta^2H$; calculated from the IRRES database for the year 2016). The latter compares well with the GNIP database long-term Ottawa amount-weighted mean ($-10.9$‰ for $\delta^{18}O$ and $-75$‰ for $\delta^2H$) (IAEA/WMO, 2018)."*

**5.2 Importance of flood-water inputs in the water balance partition**

Falcone (2007) studied the hydrological processes influencing the water balance of lakes in the Peace-Athabasca Delta, Alberta (Canada) using water isotope tracers. They reported that flooded lakes were replenished at $88\%$ in average, whereas the snowmelt-dominated lakes (i.e., not flooded) were replenished at only $31\%$ in average.

We propose to add the following material at L448:

> *"Total flood-water inputs summed at $4.82x10^6 m^3$, which is nearly equal to the lake initial volume (i.e., $4.70x10^6 m^3$). Similar results were obtained by*

*Falcone (2007) who studied the hydrological processes influencing the water balance of lakes in the Peace-Athabasca Delta, Alberta (Canada) using water isotope tracers. They reported that the springtime freshet (in 2003) did replenish the flooded lakes from $68\%$ to $> 100\%$ ($88\%$ in average)".*

**5.3 Resilience of Lake A to surface water pollution**

The resilience of kettle lakes in Quebec (Canada) was studied by Arnoux et al. (2017) and this data could be added to Figure 9 for a comparative purpose. A discussion addressing the similitudes and differences between the two datasets (i.e., Arnoux et al. (2017) vs this study) is relevant.

We propose to add the following material (in replacement of L489-493):

*"Arnoux et al. (2017) suggested that lakes with G-index $> 50\%$ and $t_f < 5$ years are considered sensitive to groundwater changes, but resilient to surface water pollution. From all the kettle lakes (n = 21) studied by Arnoux et al. (2017), $> 25\%$ were also found to have a G-index $< 50\%$ and a $t_f < 1$ year and were thus characterized as highly sensitive to groundwater quantity and quality changes. Concerning Lake A, all studied scenarios (i.e., reference scenarios A and B and the sensitivity analysis) also yielded to a high sensitivity to groundwater changes. As opposed to the kettle lakes studied by Arnoux et al. (2017), Lake A is located in a floodplain and the flood-water inputs were demonstrated to be highly important over the annual water budget due to a prolonged contribution from flood-water bank storage. Hence, a change in the intensity and/or duration of the yearly recurrent flood events can impact the replenishment of the aquifer (i.e., bank storage) and result in a groundwater quantity and/or quality change."*

**RC2-6a.** \*\* Subsection 5.3 is a bit confusing. The authors provided a list of physical water quality parameters + ions earlier in their manuscript. Based on their introduction, I expected those physical water quality parameters + ions to be used to support the "surface water pollution" aspects of the manuscript. However, in subsection 5.3, there is no reference to those parameters/ions, and the assessment of resilience to water pollution is solely based on mean flushing time. Why were the parameters/ions described earlier not used?

**Answer: Clarification.** It was deemed preferable not to discuss the physico-chemical parameters and the major ion data. This decision was made in order to limit the manuscript length. However, we agree that section "5.3 Resilience to surface water pollution" could benefit from an additional discussion concerning the geochemical data. We suggest that the geochemical signature of Lake A is compared to the one of Lake DM, Lake B and regional groundwater (i.e., observed at piezometers upstream of Lake B). Below is the proposed additional figure (Fig. 4 submitted with this response) and interpretation to add to section "4.1 Water fluxes and isotopic framework":

*"Water samples were additionally collected at the surface and at various depths within Lake A (n = 23) and at Lake DM (n = 1) during the study period and analyzed for major ions, as detailed in section 3.2. The geochemical facies of Lake A and Lake DM samples are illustrated in Figure 5 by the means of a Piper diagram. Both Lake A and flood-water were found to be Ca-$HCO_3$ types, which is typical for precipitation- and snowmelt-dominated waters (Clark, 2015). The geochemistry of Lake A is relatively constant throughout the year and confirms depth-wise homogeneity. Water samples were also collected for comparison at the surface and depth of Lake B (n = 42) and at observation well Z16 (n = 11), which is upstream of Lake B and, thus, representative of the local groundwater contributing to the latter (Ageos, 2016). The geochemistry of Lake B is significantly distinct from Lake A and appears to be influenced by*

[Figure]

*regional groundwater of Na-Cl water type."*

Also, we propose to add the following material to section "5.3 Resilience of Lake A to surface water pollution" (at L509):

*"Considering the Ca-$HCO_3$ water type of Lake A (see section 4.1) and the importance of the flood-water inputs to the annual lake water budget (see sections 5.1 and 5.2), the flood-water contribution is likely to dampen intra-annual geochemical variations at Lake A and is controlling the low-mineralization levels compared to the neighboring lake (i.e., Lake B). If a reduction in the flood-water inputs was to occur, the geochemistry of Lake A could potentially shift towards that of Lake B. In such a case, an increase of the salinity and in the concentration of $Na^+$, $Ca^{2+}$, $SO_4^{2+}$ and $Cl^-$ would be expected for Lake A.*

Note that this issue was also highlighted by Reviewer 2 (see SC1-12c).

**RC2-6b.** And is it adequate to use the mean flushing time as a proxy measure for a lake's resilience to "all" surface water pollution, regardless of the reactivity/sorption coefficients of the chemical determinants under consideration? This last question may be out of scope for the present manuscript, but a clarification sentence would help manage readers' expectations.

**Answer: Clarification.** Reviewer 2 is correct; the fate of the contaminant is also to be considered when assessing for the sensitivity/resilience of a lake to a specific surface water pollution. In the submitted version of the manuscript, we simply aimed at demonstrating a broader scenario by depicting the mean flushing time by groundwater, i.e., a key parameter for the resilience to surface water pollution. This reflects the global sensitivity of a water body and is to be adapted for each specific

contaminant. That being said, a brief addition to the discussion could be helpful. Below is the proposed additional material:

> *"Initially proposed by Arnoux et al. (2017), this interpretation framework allows unraveling the response time of a lake to changes in groundwater and/or surface water quantity and/or quality. It depicts a general case, applicable to various surface water pollution, regardless of the fate of the contaminants. Hence, care should be taken when interpreting the sensitivity to contaminants which are subject to attenuation processes, such as degradation and sorption."*

**RC2-7.**\*\* The first sentence of the conclusion states reiterates that the "goal" of the present study was to "develop a volume-dependent transient isotopic mass balance model, assuming well-mixed conditions, in order to better understand the dynamics of the hydrological processes at a flood-affected lake in southern Canada". As commented upon above, I find this to be rather vague. After reading through the details of the manuscript, it seems like the authors specifically want to address questions related to the relative importance of groundwater for Lake A on an aggregated annual scale as well as through different seasonal/wetness conditions (what they refer to as temporal variability of hydrological processes). The authors also dedicated a fair amount of time/manuscript space to discuss many different elements, e.g.: the peculiarities of lake dynamics under flooding conditions, uncertainties associated with their isotope mass balance model (those uncertainties are multiple in nature, i.e., input data uncertainty, structural data uncertainty, output data uncertainty, even maybe model parameter uncertainty), and the application of pollution resilience assessment framework. It is quite difficult, from the whole manuscript, to figure out which of those elements are primary versus secondary targets/goals/objectives of the manuscript and how they relate (or not) to one another. I think that there is a nice science story in the manuscript, and I hope the authors will see my comments as

suggestions for strengthening it and making it interesting to the broad readership of HESS.

**Answer: Done.** We made an effort at addressing this issue based on the comments and suggestions above. See response to comment RC2-2 and RC2-4 for clarifications concerning the general and specific objectives and a proposed structure for the revised manuscript.

\* SPECIFIC COMMENTS ABOUT SOME TEXT SECTIONS OR FIGURES

See "sticky notes" and yellow highlights in the pdf proofs

**Answer:** See below.

\* TYPOS AND EDITORIAL SUGGESTIONS

See "sticky notes" and yellow highlights in the pdf proofs

**Answer:** See below.

**RC2-8.** L24: Refer to "Lake DM". What is this? Not previously defined in the abstract

**Answer: Done.** Reviewer 2 is correct; it must be defined first. It should be written "Lake Deux-Montagnes (DM)". This needs to be corrected.

**RC2-9.** L36-37: Refer to "Lerner and Harris, 2009;Cunha et al.,2016;Scanlon et al., 2005". Space missing

**Answer: Done.** We used the EndNote® Output Style File from Copernicus (downloaded from https://www.hydrology-and-earth-system-sciences.net). We corrected the downloaded referencing style by changing the multiple citation separator to '; ' (i.e., semi-colon and space). All the references were automatically updated.

**RC2-10.** L46: Spaces missing in-between successive in-text citations. That tends to
happen throughout the text and needs to be rectified

**Answer: Done.** See response to comment RC2-9 (above).

**RC2-11.** L56: Refer to "long". Do the authors mean "long" or "wide:?

**Answer: Done.** We meant 1 km "wide" area. This is to be corrected in the manuscript.

**RC2-12.** L59: Refer to "connectivity". Connectivity between what and what?

**Answer: Done.** We meant the "hydraulic connectivity between Lake A and Lake DM". This needs to be specified.

**RC2-13.** L89: What does "it" refer to, here?

**Answer: Done.** We refer to S1. This needs to be specified.

**RC2-14.** L96: Refer to "(Deux-Montagnes)". This should probably have been specified earlier, i.e., the first time that "Lake DM" is mentioned in this section (see previous page)

**Answer: Done.** Reviewer 2 is correct. It needs to be specified at L91.

**RC2-15.** L97: "...drains via the St. Lawrence River..." By using the term "via", do the authors mean "to" or "toward" the St. Lawrence River?

**Answer:** Done. We meant "to".

**RC2-16.** L99-100: Refer to "...it is likely that no or very limited subsurface hydraulic connection between Lake A and Lake DM exist." The authors likely need to expand on this hypothesis a bit. Has this been verified in the field, or is it an assumption/hypothesis solely based on surficial deposits information?

**Answer: Done and clarifications.** Lake A and Lake B were created by the sand dredging activities. This sand deposit was described by Ageos (2010) as a buried valley, which extends in the NE-SW direction and was carved into the Champlain Sea clay. We propose to add a new figure (see Fig. 2 submitted with this response) and the following text to L97:

*"While alluvial sands were mapped in the area between Lake A and Lake DM (see Figure 1b), stratigraphic data (i.e., well logs) confirms that only a thin layer (few centimeters to roughly 2 meters) of alluvial sands are deposited on top the clayey sediments in the area between Lake A and Lake DM (see Figure 1d)."*

**RC2-17.** L151: Refer to "... August 17,2017...." Space missing.

**Answer: Done.** Indeed, there is a space missing. Thank you.

**RC2-18.** L153: Physico-chemical parameters and ions do not seem to have been used at all by the authors, i.e., they are not presented in any result table or figure. Why are the sampling procedures related to them presented here, then?

**Answer:** See response to comment RC2-6a and RC2-6b.

**RC2-19.** L164: Refer to "Stable isotopes of water". Oxygen and hydrogen.

**Answer: Done.** We are grateful to Reviewer 2 for pointing that out. It is indeed more appropriate to use "stable isotopes of oxygen and hydrogen". This also needs to be corrected at L304 (see comment RC2-23b).

**RC2-20.** L271: There seem to be a lot of "results" that are listed in section 3, which most readers would equate to a Methods section (and not a results section). I would

suggest that the authors try to reorganize their text a bit, so that section 3 only focuses on methods while section 4 summarizes results

**Answer:** See response to comment RC2-6a and RC2-6b.

**RC2-21.** L281: Refer to "are". Grammar issue. Subject singular subject, verb plural

**Answer: Done.** It needs to be corrected to "is".

**RC2-22.** L282: Refer to "They". Flood-water samples?

**Answer: Done.** We must specify "The flood-water samples" instead of "They".

**RC2-23a.** L304: Refer to "Deciphering surface and groundwater inputs". Was that a specific research objective/question for the present study?

**Answer: Done.** We propose to merge the subsection "Deciphering surface and groundwater inputs" with the revised subsection "4.1 Water fluxes and isotopic framework". We believe that it can facilitate the reading as observations in both Figure 3 and Figure 4 are complementary.

**RC2-23b.** Refer to "stable isotopes of water". I know that this is a phrase that is used a lot (including by myself, sometimes, mistakingly) but I suggest that the authors rephrase it, as often as is appropriate, in their manuscript. Technically, water does not have any isotopes, but oxygen and hydrogen do.

**Answer: Done.** See response to comment RC2-19.

**RC2-24.** L322: Refer to "…isotopic composition Lake A…". Change to "…isotopic composition of Lake A…"

**Answer: Done.** Thank you.

**RC2-25.** L329: Refer to "Quantification of flood-water inputs into Lake A". Is that

another specific research objective/question in the present study?

**Answer: Done.** The importance of the flood-water inputs on the water budget is in fact the main objective of our study. See the response to comment RC2-7.

**RC2-26.** L335: Refer to "...and daily net...". Change to "... and the daily net...".

**Answer: Done.** Thank you.

**RC2-27.** L340: Refer to "On early August..." Change to "In early August..."

**Answer: Done.** Thank you.

**RC2-28.** L385: Refer to "Sensitivity analysis". Why was this done and how novel (i.e., different from what others have done) is this? Is that another specific research question/objective targeted by the present study?

**Answer: Clarifications.** The sensitivity analysis was not a specific objective, but it was needed to grasp the relative impact of the input parameter's uncertainties on the model outputs. In order words, our objective was to assess the reliability of the model outputs against a range of possible input values. As the model outputs remained comparable to the reference scenario, we concluded that the model was representative of the local hydrological processes.

In order to limit the length of the manuscript, we propose to move Table 2 to the "Supplementary material" section. Note that the results of the sensitivity analysis are also depicted in Figure 7 and Figure 9.

**RC2-29.** L411: Refer to "Negligible". Change to "A negligible".

**Answer: Done.** Thank you.

**RC2-30.** L413: Refer to "...the value...". Change to "...the values..."

**Answer: Done.** Thank you.

**RC2-31.** L416: Refer to "...only small...". Change to "...only a small..."

**Answer: Done.** Thank you.

**RC2-32.** L420: Refer to "Importance of groundwater on the annual lake budget". So the reader should assume that quantifying this was a major goal of the present study?

**Answer: Clarifications.** The main objective of the study was to evaluate the importance of flood-water, relatively to the groundwater inputs, on the annual lake budget. Please see response to comment RC2-7 for the reformulation of the general and specific objectives of this study. To avoid any confusion regarding the objectives, we proposed a modification to the structure of the manuscript (see response to RC2-4).

**RC2-33.** L438: Refer to "Temporal variability of the hydrological processes". Specific research objective of this study?

**Answer: Clarifications.** We consider that this a specific objective of the study. We think it is important to underline the temporal variability of the hydrological processes, because the flood events generally occur during a specific time of the year.

**RC2-34.** L464: Refer to "...Flood-water...". Change to "...flood-water..."

**Answer: Done.** Thank you.

**RC2-35.** L469: Refer to "...that water quality...". Change to "...that the water quality..."

**Answer: Done.** Thank you.

**RC2-36.** L483: Refer to "Resilience". Since the word resilience can have very different meaning in different sub-fields or sub-disciplines of ecology, hydrology and

ecohydrology, I strongly suggest that the authors provide their adopted definition for that term.

**Answer: Done.** We are grateful to Reviewer 2 for this suggestion. We propose to add the following material to L484:

> *"Resilience of a system has been defined as its capacity to cope with perturbations (i.e., internal and/or external changes) while maintain its state (Cumming et al., 2005). In the case of a lake, perturbations can manifest as a change in the water quantity and quality contributing to the water balance."*

**RC2-37.** L495: About the x-axis label: should we read tG or tf? I am a bit confused...

**Answer: Done.** Thank you. This was also pointed out by the other reviewers. Indeed, it needs to be corrected to $t_f$ (the mean flushing time by groundwater).

**RC2-38.** L499: Refer to "Arnoux et al, 2017a". This likely warrants more explanation in the text, so that the readers can get a good idea of where that representation/framework is coming from and what its underlying rationale is without having to go back to the 2017 paper.

**Answer: Clarifications.** Arnoux et al. (2017) performed isotopic mass balances over of 21 kettle lakes in Quebec (Canada) and speculated about their response to a perturbation by comparing the G-index and $t_f$. Considering these two indices, they proposed an interpretative framework to discuss the resilience of the lakes to surface water pollution, depicted on a plot of G-Index vs $1/t_f$, as we did in Figure 9.

We suggest adding a description of the interpretative framework developed by Arnoux et al. (2017) at L488 and to compare our results to the ones of Arnoux et al. (2017) in Figure 9 and discuss the similarities/differences and implications (see response to comment RC2-5 for the proposed additional material).

**RC2-39.** L500-509: This paragraph comes a bit out of nowhere. It should be better linked with previous text, as well as specific research questions or objectives (which are currently missing from the manuscript)

**Answer: Done.** We propose to add the following material at L500:

> *"Considering the above, it is possible to speculate about the potential future impacts of climate change on Lake A."*

Additionally, it is relevant to add a discussion concerning the impact of the flood-water inputs on the geochemistry of Lake A (see response to comment RC2-6a).

**RC2-40.** Figure B1: The caption refers to blue hollow symbols and a solid blue line by my version of the manuscript includes a black and white figure, not a color figure.

**Answer: Done.** Thank you. Indeed, the caption needs to be consistent with the figure. It is to be corrected to "black".

**References**

Ageos: Drinking water supply: Application for an authorization under Section 31 of Groundwater Catchment Regulation: Hydrogeological expert report, AGEOS, Brossard, QC, Canada2010-723, volume 1 de 2, 2010.

Ageos: Drinking water supply: Monitoring of piezometric fluctuations in the water table and lake levels: Period from April 27, 2012 to December 17, 2015: Annual Report 2015, AGEOS, Brossard, QC, Canada, 42, 2016.

Arnoux, M., Barbecot, F., Gibert-Brunet, E., Gibson, J., Rosa, E., Noret, A., and Monvoisin, G.: Geochemical and isotopic mass balances of kettle lakes in southern Quebec (Canada) as tools to document variations in groundwater quantity and quality, Environmental Earth Sciences, 76, 106, 10.1007/s12665-017-6410-6, 2017.

Bednarek, A. T.: Undamming Rivers: A Review of the Ecological Impacts of Dam Removal, Environmental Management, 27, 803-814, 10.1007/s002670010189, 2001.

Biron, P. M., Buffin-Bélanger, T., Larocque, M., Choné, G., Cloutier, C.-A., Ouellet, M.-A., Demers, S., Olsen, T., Desjarlais, C., and Eyquem, J.: Freedom Space for Rivers: A Sustainable Management Approach to Enhance River Resilience, Environmental Management, 54, 1056-1073, 10.1007/s00267-014-0366-z, 2014.

Buffin-Bélanger, T., Biron, P. M., Larocque, M., Demers, S., Olsen, T., Choné, G., Ouellet, M.-A., Cloutier, C.-A., Desjarlais, C., and Eyquem, J.: Freedom space for rivers: An economically viable river management concept in a changing climate, Geomorphology, 251, 137-148, 10.1016/j.geomorph.2015.05.013, 2015.

Clark, I.: Groundwater Geochemistry and Isotopes, Boca Raton, FL, 2015.

Cumming, G. S., Barnes, G., Perz, S., Schmink, M., Sieving, K. E., Southworth, J., Binford, M., Holt, R. D., Stickler, C., and Van Holt, T.: An Exploratory Framework for the Empirical Measurement of Resilience, Ecosystems, 8, 975-987, 10.1007/s10021-005-0129-z, 2005.

Dixon, S. J., Sear, D. A., Odoni, N. A., Sykes, T., and Lane, S. N.: The effects of river restoration on catchment scale flood risk and flood hydrology, Earth Surface Processes and Landforms, 41, 997-1008, 10.1002/esp.3919, 2016.

Falcone, M.: Assessing hydrological processes controlling the water balance of lakes in the Peace-Athabasca Delta, Alberta, Canada using water isotope tracers, UWSpace, 2007.
* * *
[Figure]

**Fig. 1.** Geochemical signature of Lake A, Lake B, flood-water and the regional groundwater (GW).

[Figure]

**Fig. 2.** Schematic representation of the A-A' cross-section.

---

## Author Response (AR1)

Dear Pr Genevieve Ali,

First, we would like to thank the two reviewers (Bruno Hamelin and anonymous referee) and the participant in the interactive discussion (Michael Rosen) for their detailed reading of the manuscript and for their valuable recommendations. All of these efforts have contributed significantly to improving our work. Please see our point-by-point response to the reviewers' comments for the related corrections in the manuscript.

In addition, we made further improvements to better address the key-comments that you highlighted. Indeed, as efforts were made at better structuring the manuscript, we seized the opportunity to better explain the context of our study. We find this significantly helps to state the research objectives more clearly (editor's key comment 1). Also, we provide a number of clarifications concerning the data quality and the modelling framework throughout the manuscript (editor's key comment 2). Lastly, we improve the discussion by further comparing our results to the literature. This has allowed us to open the discussion to a broader readership (editor's key comment 3) by demonstrating the importance of the bank storage effect when performing water mass balance assessments on flood-affected lakes to correctly depict their resilience.

All the above-mentioned improvements are further explained below. Please note that a number of grammatical errors were also corrected in the revised manuscript (all marked in red in the attached version).

**EC-1. The absence of specific research objectives or questions**

Efforts were made at better highlighting the new perspectives concerning the application of isotopic approaches in ungauged basins (see introduction). It helped to reformulate our research objectives. Below is the revised context and main objective of the study:

"Isotopic mass balance models are typically applied to contexts where there are no surface water inputs (Sacks et al., 2014; Arnoux et al., 2017b) and/or the surface water inputs are quantified by stream gauging (Stets et al., 2010). In remote environments, such as in northern Canada, application of isotopic methods is particularly convenient, as direct measurement of surface water inflow is difficult or nearly impossible (Turner et al., 2010; Brock et al., 2007). Recently, Haig et al. (2020) opened up new perspectives, as they reported excellent agreement between results obtained via isotopic mass balance and gauging techniques when assessing the water budget of connected lakes in Saskatchewan (Canada). They highlighted that the isotopic approach was efficient for characterizing the impacts of floods and droughts, and that a broad application can contribute to water resources management in providing information to understand the vulnerability of ungauged systems. As future climate change impacts are expected to include increases in flood magnitude and frequency (Aissia et al., 2012), flood-affected lake water budget assessments are of utmost importance.

The main objective of this study is to demonstrate the application of isotopic mass balance to flood-affected lakes, as this approach is particularly opportune in providing estimates of the water balances and insights on the dynamics of ungauged systems."

**EC-2. A lack of clarity in several instances, in relation to data quality/continuity or the chosen modelling framework**

**EC-2a.** In Sect. 3.3, we briefly mentioned that "the potential impacts of the ice-cover formation and melting are neglected, as the ice volume is likely to represent only a small fraction (<2%) of the entire water body." As further details concerning the modelling framework were requested, we added the following material:

"Moreover, considering the ice-water isotopic separation factor, i.e., 3.1 ‰ for  $\delta^{18}$ O and 19.3 ‰ for  $\delta^{2}$ H (O'Neil, 1968) and assuming well-mixed conditions, the lake water isotopic variation would be comprised within the analytical uncertainty."

**EC-2b.** In Sect. 4.2, we detail the isotopic framework and specify that  $\delta_G$  was estimated from the LMWL-LEL intersection. It was explained

In order to better illustrate the use of this method, we added the following material:

"It is noteworthy that estimating the  $\delta_G$  from direct sampling at observation wells in the vicinity of lakes may be misleading due to potential heterogeneity (i.e., mixing between groundwater and surface water in the hyporheic zones). This consideration is particularly important at flood-affected lakes, as surface watergroundwater interactions are expected. In this context, it is advocated to estimate  $\delta_G$  from the LMWL-LEL as it better represents the inflowing water to a lake."

**EC-2c.** In Sect. 4.3.1, we state that "three sampling campaigns (i.e., on February 9, 2017, August 17, 2017 and January 25, 2018) were conducted at Lake A in order to collect water samples for isotopic analyses". We opted to specify the reason of this sampling strategy by adding the following material (underlined part):

"...[we] collect[ed] water samples for isotopic analyses from the epilimnion, metalimnion and hypolimnion (Figure 5) to account for the vertical stratification of the isotopic signature (Gibson et al., 2017)."

**EC-2d.** In Sect. 5.3, we now provide insights to improve the effectiveness of our approach, and particularly concerning the sampling strategies. Such considerations are helpful to other researchers in order to perform future mass balance studies.

**EC-3. Relevance of the study beyond to a broader readership**

**EC-3a.** The discussion concerning bank storage was strengthened and widened to a broader readership.

In Sect. 5.1, we better compare the initial estimation of the G-Index to the one considering the bank storage. To do so, we proposed an idealized scenario for which all (or part of) outputs during the flood event eventually discharged back to the lake as groundwater inputs, but which originated from Lake DM (i.e., surface water).

In Sect. 5.2, we now compare our results to the other lakes (n = 21) in similar morpho-climatic contexts. Furthermore, we discuss the effect of flooding (and bank storage) on the resilience of lakes to groundwater and surface water changes. Consequently, the revised Fig. 7 and Fig. 8 better depict the bank storage considerations.

In Sect. 5.3, we provide a new perspective by stating that "water budget assessments at artificial lakes (such as Lake A) can also contribute to track human impact on the water cycle". Note that Arnoux et al. (2017a) previously demonstrated this application for natural lakes. Moreover, we discuss the interest for performing water mass balance at lakes with rapid flushing time, "as they are precursors of the evolution of nearby surface water bodies characterized by longer flushing times".

We hope that the revised paper now meets your requirements and expectations.

Sincerely,

Janie Masse-Dufresne PhD Student Department of Civil, Geological and Mining Engineering Polytechnique Montreal

On one hand, I consider this study as a stimulating contribution to the ongoing effort of improving the reliability of isotope-based modeling for the prediction of lakes hydrological behavior. Based on an attractive, although complex, case study, it presents a commendable attempt to assess quantitatively the sensitivity of the model results to the variations of the parameters. On the other hand, reading the manuscript is a bit frustrating due to a number of weaknesses in the data set and hypotheses, and also in the description of the model algorithm.

**RC1-1a.** As for the data, the confidence that we can have on the robustness of the authors' conclusions clearly suffers from the lack of continuity of the lake level monitoring, and from the lack of documentation of the lake stratification. Due to the ice cover (line 124) and logger failure (line 140), the level of the lake is available only for a part of the flooding period. In particular, we miss the comparison between lake A and lake DM for summer and fall 2017, when lake DM was seemingly above the threshold, and should thus have overflown again into lake A (Figure 2).

**Answer: Clarification.** It is true that the data is limited, but we believe that it is sufficient and reliable to perform an isotopic mass balance model. We did not initially orchestrate the sampling campaigns and monitoring program to perform such study. However, a 100-year flood occurred during springtime 2017 and we took advantage of the importance of this hydrological event to assess the partition of the water mass balance under extreme conditions.

Considering Quebec's meteorological context, surface water level monitoring and surface water sampling can be hard to achieve during specific periods of the year. For instance, during springtime, it can be dangerous to perform in-lake water sampling due to ice-cover melting and flooding. Moreover, access to the study site was limited during the peak of the flood event. Ice-cover also constitutes a challenge for the lake water level measurements. Although attempts were made at installing loggers throughout late autumn and wintertime, drifting ice can damage the equipment and lead to its loss. Still, near shore lake water samples were collected in early May and provide valuable data for the characterization of the evolution of the isotopic signature of the lake. Also, punctual measurements of Lake A water level have been carried throughout the study period and provide complementary data to understand the evolution of the water levels (see the revised version of Fig. 2 in the manuscript).

**RC1-1b.** More importantly, the lack of temperature and isotope data during the peak of the flooding period, and around the fall overturning, casts a strong doubt on the justification of the assumption of lake homogeneity. This weakness is acknowledged by the authors at several places (line 260, 269-270, 311, 433, 519). However, the reader is left with the impression that a two-layer model would definitely be needed, and that the model results might be much more sensitive to the seasonal stratification than to the other parameters tested in the study.

Answer: Clarification and done. We are grateful to the reviewer for this comment. We consider that the developed model yields a valuable estimation of Lake A water balance, because we accounted for the lake stratification by using a depth-average isotopic signature to represent  $\delta_{L}$ . In this regard, we propose to add a justification for the use of a well-mixed model in the revised manuscript (at the beginning of Sect. 3.3):

"Stable isotope mass balances can either be performed based on (i) a well-mixed single layer model or (ii) a depth resolved multi-layered model. In a recent study, Arnoux et al. (2017b) compared a well-mixed model and a depth-resolved multi-layer model. Both models yielded similar results and provided a general understanding of the groundwater-surface water interactions. The multi-layer model additionally allowed for the determination of groundwater flow with depth, but required a temporally- and depth-resolved sampling

in order to ensure a thorough understanding of the stability/mixing of the different layers. Such important sampling and monitoring efforts are however often unrealistic in remote and/or flood-affected contexts. Additionally, Gibson et al. (2017) studied the impact of sampling strategies on the water yield (i.e., the depth-equivalent runoff to the lake) estimations for the Turkey Lake (32 m deep) under stratified and well-mixed conditions. They reported 18% difference on the water yield when performing grab sampling (i.e., 1 sample at 1 m depth) and bulk sampling (i.e., assessment of the whole lake water column). The difference was less important (i.e., 11%) when comparing bulk sampling to integrated sampling for epilimnion, metalimnion and hypolimnion. They also reported discrepancies up to 20% for the water yield estimations at the same lake according to the timing of the lake water sampling. This last result shows that temporal shifts may induce greater bias than the uncertainty related to the lake stratification. For these reasons, we advocated the application of a well-mixed model."

**RC1-2.** Another source of worry arises from a possibly spurious choice for the isotope composition of the groundwater inflow end member. As soon as the authors show that one of the two main supplies of lake A, i.e. the flood water overflowing from lake DM, shows an isotope composition which is below the lake evaporation line, one would expect the other main source, i.e. groundwater, to lie above that line, and not on it (line 290). This needs to be thoroughly discussed, all the more as the authors emphasize that ïAcd'G is among the most stringent parameters. More generally, some explanation is needed about the origin of the isotopic difference between the groundwater and flood water. Would it be possible to collect samples from the aquifer, away from the influence from the lake? How does the aquifer composition compare with the amount-weighted average of the rainfall seasonal variation?

**Answer: Clarification.** There is a scientific consensus concerning the isotopic composition of groundwater contributing to a lake (Edwards et al., 2004). The isotopic composition of the total inputs ( $\delta_I$ ) to a lake is resulting from the mixing between the isotopic composition of groundwater inflow ( $\delta_G$ ), surface water inflow ( $\delta_{Is}$ ) and precipitations ( $\delta_P$ ) and can be defined by:

$$\delta_{I} = \frac{\delta_{IS}I_{S} + \delta_{G}I_{G} + \delta_{P}P}{I}$$

where  $I_s$ ,  $I_g$  and P are the surface water inputs, groundwater inputs and precipitations, respectively. The total inputs (I) are described as  $I = I_s + I_g + P$ .

Conceptually,  $\delta_G$ ,  $\delta_{Is}$  and  $\delta_P$  all plot along the LMWL, so does  $\delta_I$  (given no influence of evaporation). Then, the position of  $\delta_I$  on the LMWL is controlled by the relative proportions of  $I_s$ ,  $I_G$  and P. In general,  $\delta_I$  is not significantly influenced by  $\delta_P$ , because  $P \ll I$ .

In cases where there is no surface water input (i.e.,  $I_s = 0$ ), the  $\delta_I \approx \delta_G$  and the isotopic signature of the lake ( $\delta_L$ ) will evolve along the LEL from  $\delta_G$ . In other words, the intersection between the LMWL and the LEL corresponds to the isotopic signature of the lake if E=0 and is a good estimate of  $\delta_G$ , when  $I_s = 0$ .

In the case of our study site, the yearly recurring flood events are affecting the isotopic composition of the local groundwater contributing to Lake A. In fact, the hydraulic connection between Lake DM and Lake A and the high water levels result in an enhanced springtime recharge of relatively depleted water. Hence, the isotopic composition of local groundwater was conceptualized as a mixture between flood-water and isotopic composition of regional groundwater and, therefore, more depleted than the amount-weighted average of  $\delta_P$ , which is -10.2‰ for  $\delta^{18}$ O and -68‰ for  $\delta^{2}$ H (calculated from the IRRES database for the year 2016). The latter compares well with the GNIP database long-term Ottawa amount-weighted mean (-10.9‰ for  $\delta^{18}$ O and -75‰ for  $\delta^{2}$ H) (IAEA/WMO, 2018). While the partition between IS and IG was not known a priori, it is possible to infer that the intersection between the LMWL and the LEL would correspond to the most depleted isotopic signature the local groundwater could have. Computing the isotopic mass balance model with such estimation of the  $\delta_G$  is conservative because it yields to a lower limit of the estimation of groundwater exchange.

Furthermore, estimating the  $\delta_G$  from the intersection between the LMWL and the LEL was considered more adequate than measuring it from groundwater samples. Indeed, in hydrogeological contexts where groundwater-surface water interactions are important (due to floods and/or groundwater pumping), the chemistry and isotopic signature of groundwater typically bear some local heterogeneity. Hence, the representativity from a groundwater sample can be hard to understand. In the context of this study, it was preferable to estimate  $\delta_L$  from the intersection between the lake's LEL and the LMWL, as it represents the mean isotopic signature of the local groundwater contributing to the lake.

Lines 288 to 292 (in the original manuscript) were corrected to (now in Sect. 4.2 in the revised manuscript):

"It has been argued that the LMWL-LEL intersection is representative of the isotopic composition of the inflowing water to a lake and is thus commonly used to depict the isotopic signature of groundwater ( $\delta_G$ ) in isotopic mass balance applications (Gibson et al., 1993; Wolfe et al., 2007; Edwards et al., 2004). Concerning the study site, the intersection between the St-Bruno LMWL and Lake A LEL corresponds to -11.26 ‰ for  $\delta^{18}O$  and -77 ‰ for  $\delta^{2}H$ , and was used as an estimate of  $\delta_G$  in the isotopic mass balance model. It is noteworthy that estimating the  $\delta_G$  from direct sampling at observation wells in the vicinity of lakes may be misleading due to potential heterogeneity (i.e., mixing between groundwater and surface water in the hyporheic zones)."

Note that this issue was also addressed by M. Rosen (see response to comment SC1-22a).

**RC1-3.** On another aspect, I have some reservation about the way the model is described. A luxury of details is given on several very classical aspects already extensively described in previous works (i.e. Craig & Gordon's approach of the isotopic budgets), which could thus be better placed in appendix, while the description takes shortcuts on other key linkage in the modeling procedure. For instance, the reader should not have to wait until line 357 (results section) to learn the hypotheses leading to the outflow estimate! Another example is the emphasis put on the volume-dependent modeling (line 49 and 54, 200-210). If the authors want to convince the reader that this is important, they have to better explain equation (6) to (10), which are quite cryptic, and compare with the same equation based on a constant volume approximation. This should also be discussed when looking at the results. This whole section should be written again, as a true instruction manual for anyone willing to apply such a model to another case-study.

**Answer: Done.** We agree that the more common aspects of the model could be easily described in the appendix and that we would benefit from highlighting the details concerning site-specific points. Hence, we propose to present the detailed computation for f(u),  $\delta_E$ , m,  $\delta_S$ ,  $\delta^*$  and  $\delta_A$  in *Appendix A*.

**RC1-4.** In general, I have the feeling that the overall structure of the manuscript is a bit messy. I would recommend giving first all the information that can be deduced from the lake level variations (i.e. line 330-349), in order to introduce properly the aims of using isotopes to better unravel the contribution of the different sources.

**Answer: Done.** We made an effort at reorganizing the sections of the manuscript to make a better use of the "4 Results" and the "5 Discussion" sections. Below is the proposed structure:

1 Introduction 2 Study site 2.1 Geological and hydrological settings 2.2 Conceptualization of the groundwater-surface water interactions 3 Methods 3.1 Field measurements 3.2 Water sampling and analytical techniques 3.3 Stable isotope mass balance 3.4 Water fluxes 4 Results

4.1 Hydrodynamics of the flood event

4.2 Isotopic and geochemical framework

4.3 Evaluation of the water budget

4.3.1 Volume-dependent isotopic mass balance

4.3.2 Sensitivity analysis

4.4 Temporal variability in the water balance partition

**5 Discussion**

- 5.1 Importance of bank storage discharge on the water balance partition
- 5.2 Resilience of lakes to surface water and groundwater changes
- 5.3 Implications for water management

In the revised Sect. 2, we propose the addition of a subsection to better illustrate our conceptual model of the study site (i.e., "2.2 Conceptualization of the groundwater-surface water interactions"). A 3D schematic representation was added (i.e., revised Fig. 2).

Also, the original subsection "4.2.1 Insights from net water fluxes at Lake A" (cf. L330-349) was not reporting any specific objective of the study, but rather provides a "reality check" by describing the net water fluxes and Lake A volume variation. Hence, we opted to merge the results initially presented in this sub-section with the revised subsection "4.1 Hydrodynamics of the flood event". The original Fig. 5 is no longer appearing in the revised manuscript, as the revised Fig.3 also depicts these results.

Specific comments, in addition to those already pointed out by the other reviewer:

**RC1-5.** Line 220-239: "Outflow fluctuations were derived from water level variations at Lake A using linear interpolation between adjusted daily minimum and maximum outflow. Daily inflow into Lake A was calculated to compensate for the adjusted outflow, as the net water fluxes are required to be equal to the lake's daily volume variation." I still do not understand what is done exactly on this key point. This needs to be written in equations and related to the main unknowns in equations (1) to (10).

Answer: Done. To facilitate the reading and avoid confusion, we suggest a reformulation of Line 226-229:

"Considering the above, it was assumed that the daily outflow flux from Lake A varied linearly according to the lake water level; the minimum and maximum outflow ( $Q_{min}$  and  $Q_{max}$ ) corresponding to the minimum and maximum water level, respectively. The outflow range (i.e., minimum and maximum values) was adjusted to obtain best fit between the calculated and observed  $\delta_L$ .

Total daily inflow (sum of daily P,  $I_s$  and  $I_G$ ) into Lake A compensates for the adjusted daily outflow and daily lake volume difference."

**RC1-6.** Line 275: "Interpolation was used to simulate the \_P on a daily-time step." This suggests that the rain data show a smooth evolution through time along the season. Is this really the case?

**Answer: Clarification.** Reviewer 1 is correct. Sampling for precipitations was done on a monthly time step (approximately). Therefore, we did not analyze the isotopic signature of every single precipitation event and interpolation between the monthly samples was necessary to compute the model at a daily time step. When computing the evolution of the isotopic signature of the lake, precipitations are mixed instantaneously with the whole lake volume. As the daily precipitations are much smaller than the whole lake volume (and the other inputs), the bias caused by the interpolation of the isotopic signature of precipitations is not expected to significantly affect the results of the model. There would have been no gain on the accuracy of the model in sampling precipitations at a smaller time step.

**RC1-7.** Line 284: same evaporation slope for lake waters and flood water. Is this not surprising, as this slope depends on the climate parameters of Craig & Gordon's equation, while flooding and evaporation do not occur at the same period of the year?

**Answer: Clarification.** The slope of the LEL is strongly influenced by the relative humidity, and to a less extent by the temperature and the lake water balance (Gibson et al., 2015). Given the density of surface water bodies in Canada, the relative humidity is almost constant throughout the year and is roughly 80%. Hence, the LEL slope variations are expected to be very small for a specific location.

**RC1-8.** Line 333: lake elevation assessed from well VP. Unclear what is meant by this statement as the difference of elevation of the water level between lake A and well VP is supposed to change with time along with the recharge/discharge alternation. (already pointed out by the other reviewer).

**Answer: Clarification.** There is in fact a water level difference between Lake A and the observation well VP. Note that the water level at VP is always lower than at Lake A, due to the pumping at the neighbouring bank filtration site.

When computing the model, the absolute lake water level is not important. The equations of the model are dependent on f, which is the remaining fraction of lake water:

$$f = \frac{V}{V_0}$$

where  $V_0$  is the initial lake volume (at the beginning of the time step).

As the time series for Lake A water level was not covering the entire study period, a proxy was needed. The correlation coefficient between Lake A and VP is 0.9885 for all the available data (from 2017 to 2020, which spans both high and low water periods). Hence, while the water levels at Lake A and VP are not identical, the daily variations are expected to be similar. We thus conceive that VP is a good surrogate for Lake A.

L329-334 (in the original manuscript) was improved and moved to the revised Sect. 4.3.1):

"Lake A volume variations are estimated from water level records at Lake A and assuming a constant lake area. When not available, water levels at Lake DM or observation well VP are used as proxies. Water level of Lake DM is used when there is a hydraulic connection with Lake A (i.e., above the topographical threshold) and data from observation well VP is used otherwise. These approximations were deemed acceptable because the simulation of  $\delta_L$  depends on the remaining fraction of lake water (not the absolute water level), and daily variations of the water levels at Lake A, Lake DM and observation well VP were shown to be similar (see Section 4.1)."

**RC1-9.** Line 357: "the outflow fluxes are proportional to the lake's water level. We adjusted minimum and maximum outflow fluxes (Q) so that the latter respectively correspond to the minimum and maximum water levels." Again, (see comment above), I do not understand what this means.

**Answer: Done.** A reformulation of Line 226-229 was proposed. See response to comment RC1-5. Additionally, we added the labels " $Q_{min}$ " and " $Q_{max}$ " (i.e., minimum and maximum outflow fluxes) to the revised Fig. 3. This allows to illustrate the timing of " $Q_{min}$ " and " $Q_{max}$ ", i.e., synchronous to the minimum and maximum water levels (of Lake A).

**RC1-10.** Line 368 and figure 6: The results obtained from ïAcd'D are strictly redundant to those from ïAcd'18O. What is really missing in this figure is some data at the beginning of May!

**Answer: Clarification.** The use of dual isotopes (i.e.,  $\delta^{18}$ O and  $\delta^{2}$ H) is helpful to perform adequate parametrization of  $\delta_{A}$ , especially in seasonal climates. However, this aspect was not one of the main goals of our study and a comprehensive study on this topic was already published by Yi et al. (2008).

Concerning the apparent lack of data in early May, we reiterate that in-lake water sampling can be dangerous to achieve in certain climates (see our response to RC1-1). In fact, ice-melting and limited access to the study site during the flood event prevented us from performing bulk water sampling along the water column. Despite these field conditions, we were able to perform near-shore lake water sampling. Although these samples are seemingly representative of the surface-most part of the lake, they are still valuable for our understanding of the lake's dynamics, as the lake is expected to be fully mixed until early May due to a lack of density gradient (see below).

The following material was added to the revised manuscript (in Sect. 4.3.1):

"While depth-average  $\delta_{L}$  was not available at the end of the flood-water inputs period (i.e., in early May), water samples from the surface of Lake A provide relevant evidences to better constrain the model. Two scenarios, namely A and B, were considered. Until early May, the observed surface water temperature was < 5°C (see Figure C1), which translates to a limited density gradient along the water column and does not allow for the development of a thermal stratification. In this context, it is possible to assume that Lake A is fully mixed until early May and that the water samples from the surface of the lake are representative of the whole water body. Hence, the modeled  $\delta_{L}$  is additionally constrained at  $\delta^{18}O \approx -11.1\%$  and  $\delta^{2}H \approx -77\%$  (in early May) and at  $\delta^{18}O \approx -11.6\%$  and  $\delta^{2}H \approx -80\%$  (in late April) for scenarios A and B, respectively."

**RC1-11.** Line 434-437: scenarios A and B are supposed to compensate for the lack of data at the peak and end of the flood period. However, just mentioned like this without description, and sent back to Appendix C leaves a disastrous impression on the reader.

**Answer: Done.** In the revised manuscript, we took care to bring the comparison between the reference scenarios A and B to the forefront (in Sect. 4.3.1). We also added the following material to discuss the representativity of both scenarios:

"While the computed flows for scenario A are within a plausible range for the combination of surface and groundwater outflow processes, scenario B yielded less realistic results. As mentioned above, scenario B was constrained at  $\delta^{18}O \approx -11.6\%$  and  $\delta^2H \approx -80\%$  in late April (Fig. 6), based on a surface water sample which was taking during a temporary decreasing water level period (Fig. 3) and is thus likely less representative of the overall lake's dynamic compared to scenario A. This is demonstrating the limit of the approach and that it is important to correctly constrain the model during the flood events in order to perform precise estimations of the water balance."

**RC1-12.** Line 452: "The isotopic mass balance model revealed it was necessary to allow for significant groundwater outflow from Lake A during springtime to correctly reproduce the observed \_L ". A best illustration of this conclusion would have been to compare the results of the model with and without the groundwater outflow.

**Answer: Moot.** From our point of view, performing a simulation without any groundwater outflow would not be representative of any realistic scenario. By neglecting the evaporation and precipitations, a simulation of  $\delta_{L}$  without any groundwater outflow can be simplified to a binary mixing model, for which a flood-water volume is added to the initial lake water volume:

$$\delta_L = f_{Is} \times \delta_{Is} + f_L \times \delta_0$$
$$f_{Is} + f_L = 1$$

where  $f_{Is}$  and  $f_L$  are the relative proportion of flood-water and lake water at the end of the flood event.

Considering an initial isotopic signature of -10.15‰ for  $\delta^{18}$ O and  $\delta_{Is}$  = -12‰, the  $f_{Is}$  would be 0.4 to obtain  $\delta_{L}$  = -11.1‰ (i.e., the isotopic signature of the lake in early May). As the initial volume of the lake is 4.7 X 106 m3, the total floodwater input (Is) would be 7.8 x 106 m3. The latter roughly corresponds to an equivalent water level variation of 11 m, which is unrealistic. Hence, groundwater outflow was undoubtedly necessary to correctly reproduce the observed  $\delta_{L}$ .

**RC1-13.** Line 485-487: confusion between tG and tf. (already pointed out by the other reviewer). **Answer: Done.** It is tf and it was corrected.

**RC1-14.** Line 503-504: confusion on "increase" ? **Answer: Done.** Reviewer 1 is right. It should be written "decrease".

In this manuscript entitled "Quantifying flood-water impacts on a lake water budget via volume-dependent transient stable isotope mass balance", the authors focus on an artificial lake and justify their study by stating that "[understanding] the relative importance of the hydrological processes in lakes can also help to depict the vulnerability and/or resilience of a lake to pollution". They aim to develop a predictive model of both atmospheric and water balance controls on isotopic enrichment, quantify of flood-water inputs to the lake, and conduct a model sensitivity analysis was conducted to evaluate potential sources of uncertainty. Overall, the manuscript is of appropriate length and well written. Figures and tables are also of good quality and rich in information without being too crowded. While I enjoyed reading this manuscript, I think that the authors need to make a strong case for the broader relevance, impact and transferability of their methods or conclusions, in addition to revisiting the structure of manuscript. My most major criticisms are as follows:

**RC2-1.** \*\* In its present state, the manuscript pretty much reads like a case study report. There is nothing wrong with case studies per se, as the uniqueness of place makes the conclusions of many papers inherently site-specific. That being said, I think that the authors should try to extrapolate their conclusions (or speculate about how their conclusions might extend) to other lakes (artificial or not) in Canada, North America and around the World. What makes Lake A and Lake DM different (or not) than other lakes where similar isotope mass balance approaches have been used in the past? In other words, what makes the present study novel? What are the really key contributions that represent an advancement to the science – and that may be relevant beyond the particular site that the authors focused on? Can the results be extrapolated to depressional wetlands which are affected by flooding as well? And if results and conclusions cannot be extrapolated, what about some of the methods applied in the current manuscript?

My asking those questions is not my way to say that there are no novel contributions in this manuscript, but rather to say the authors have not explicitly identified them and should highlight them better.

Answer: Completed with clarifications. First, we are grateful to the reviewer for this valuable comment.

We improve the introduction to better highlight the broad relevance of such studies. The following material now appears in the introduction:

"Isotopic mass balance models are typically applied to contexts where there are no surface water inputs (Sacks et al., 2014; Arnoux et al., 2017b) and/or the surface water inputs are quantified by stream gauging (Stets et al., 2010). In remote environments, such as in northern Canada, application of isotopic methods is particularly convenient, as direct measurement of surface water inflow is difficult or nearly impossible (Turner et al., 2010; Brock et al., 2007). Recently, Haig et al. (2020) opened up new perspectives, as they reported excellent agreement between results obtained via isotopic mass balance and gauging techniques when assessing the water budget of connected lakes in Saskatchewan (Canada). They highlighted that the isotopic approach was efficient for characterizing the impacts of floods and droughts, and that a broad application can contribute to water resources management in providing information to understand the vulnerability of ungauged systems. As future climate change impacts are expected to include increases in flood magnitude and frequency (Aissia et al., 2012), flood-affected lake water budget assessments are of utmost importance. The main objective of this study is to demonstrate the application of isotopic mass balance to flood-affected lakes, as this approach is particularly opportune in providing estimates of the water balances and insights on the dynamics of ungauged systems."

Also, to the best of our knowledge, this is the first study to apply an isotopic approach to depict the water balance during an extreme flood event. The following was also added to the introduction:

"Our study period spans a 100-year flood, and the results of this study are therefore an example indicative of an extreme hydrological event."

Concerning the applicability of the method to other environments, please see response to RC2-2.

**RC2-2.** \*\* The introduction lacks an overarching goal or research question for the study, as well as specific research objectives or questions. Instead, the last paragraph of the intro just states that the present study builds upon two other studies. The only sentence of the introduction that could be seen as a research goal is the one that reads as: "The main purpose of this study was initially to expand our understanding of flood-affected lake dynamics in the context of a seasonal climate". It is quite vague, though, so I suggest that the authors include some more specific objectives or questions at the end of the introduction. This should also help highlight the novel contributions that the present study intends to make.

**Answer: Completed.** Based on this comment and other suggestions below, we propose the following reformulation of the general and specific objectives:

*General objective* – Demonstrate the application of isotopic mass balance to flood-affected lakes, as this approach is particularly opportune in providing estimates of the water balances and insights on the dynamics of ungauged systems

*Specific objective* 1 – Evaluate the importance of flood-water inputs on the annual water budget of a lake located in a floodplain in an urban area

*Specific objective 2* – Depict the resilience of flood-affected lakes to changes in water balance partitioning and flood-water and/or groundwater quality

In order to achieve these objectives, we used the following working steps:

Step 1 – Establish an isotopic framework based on the local water cycle

Step 2 – Evaluate the water budget considering reference scenarios (A and B)

Step 3 – Analyze the temporal variability of the groundwater inputs and the sensitivity of the lake to floodwater driven pollution

Step 4 – Demonstrate the implications of flood-water storage on the water balance partition

We can now read the following in the introduction (in replacement of L52-58 and L76-78):

"The main objective of this study is to demonstrate the application of isotopic mass balance to flood-affected lakes, as this approach is particularly opportune in providing estimates of the water balances and insights on the dynamics of ungauged systems. We thus evaluate the importance of flood-water inputs (and bank storage) on the annual water budget of a lake located in a floodplain in an urban area, in order to depict its resilience to changes in water balance partitioning and flood-water and/or groundwater quality. To do so, we first aim to establish an isotopic framework based on the local water cycle, to verify the applicability of isotopic mass balance in the present setting, as contrasting isotopic signatures are required between various water storages and fluxes, including flood-water inputs. Secondly, we quantify the water budget according to two reference scenarios (A and B) to grasp the impact of site-specific uncertainties on the computed results. Then, we analyze the temporal variability of the groundwater inputs and the sensitivity of the lake to flood-water driven pollution. Finally, we demonstrate the implications of flood-water storage on the water balance partition. It is hypothesized that the groundwater fluxes (inputs and outputs) through lake banks are unneglectable in lake water budgets, even for flood-affected lakes.

The water balance is computed via a volume-dependent transient isotopic mass balance model, which is applied to predict the daily isotopic response of an artificial lake in Canada that is ephemerally connected to a 150,000 km2 watershed during spring freshet and other periods of flooding. During these recurring perennial flood events, the surficial water fluxes entering the study lake are not constrained in a gaugeable river or canal but occur over a 1-km wide surficial flood area."

The following material was added to the conclusion (in replacement of L513-514), to underline the applicability of this method to other environments:

"Given the contrasting isotopic signature of the flood water, the isotopic mass balance model was effectively applied at the study site. We anticipate that the isotopic framework is likely to be transferable to other lake systems subject to periodic flooding including lowland lakes fed by mountain flood-waters, river deltas, wadis, or nival (snowmelt-dominated) regimes, the latter of which dominates the high latitude and high altitude coldregions including much of the Canadian landmass."

**RC2-3.**\*\* There seem to be a lot of "results" that are listed in Section 3, which most readers would equate to a Methods section (and not a results section). I would suggest that the authors try to reorganize their text a bit, so that section 3 only focuses on methods while section 4 summarizes results.

**Answer: Done.** We opted to place all section "3.3.2 Isotopic framework" in section "4 Results". Also, note that part of section 3 was moved to the Appendix A, as suggested by another reviewer's comment (RC1-3).

**RC2-4.**\*\* Following up on the previous point: Section 4 does not seem to focus on "plain results" only, as it includes several interpretations, discussions and linkages to the literature. Section 4 therefore reads as a combined Results and Discussion section, which is a bit surprising as there is a separate discussion section later. I suggest that the authors try to better distribute methodological aspects, results and discussions/ interpretation into distinct sections (and sub-sections).

**Answer: Completed.** We make an effort to reorganize the sections of the manuscript to make better use of the "4 Results" and the "5 Discussion" sections. Below is the revised structure:

1 Introduction

2 Study site

2.1 Geological and hydrological settings

2.2 Conceptualization of the groundwater-surface water interactions

3 Methods

- 3.1 Field measurements
- 3.2 Water sampling and analytical techniques
- 3.3 Stable isotope mass balance
- 3.4 Water fluxes

4 Results

- 4.1 Hydrodynamics of the flood event
- 4.2 Isotopic and geochemical framework
- 4.3 Evaluation of the water budget

4.3.1 Volume-dependent isotopic mass balance

4.3.2 Sensitivity analysis

4.4 Temporal variability in the water balance partition

**5** Discussion

5.1 Importance of bank storage discharge on the water balance partition

5.2 Resilience of lakes to surface water and groundwater changes5.3 Implications for water management

In the revised Sect. 2, we propose the addition of a subsection to better illustrate our conceptual model of the study site (i.e., "2.2 Conceptualization of the groundwater-surface water interactions"). A 3D schematic representation was added (i.e., revised Fig. 2).

Also, the original subsection "4.2.1 Insights from net water fluxes at Lake A" (cf. L330-349) was not reporting any specific objective of the study, but rather provides a "reality check" by describing the net water fluxes and Lake A volume variation. Hence, we opted to merge the results initially presented in this sub-section with the revised subsection "4.1 Hydrodynamics of the flood event". The original Fig. 5 is no longer appearing in the revised manuscript, as the revised Fig.3 also depicts these results.

**RC2-5.**\*\* Along the same lines as the two previous points, Section 5 is a bit confusing. There are completely new results (e.g., Table 3, Figure 7) first reported on in this section. Conversely, there are not a lot of literature references (none in subsection 5.1, and only 1 literature reference, as far as I can see, in subsection 5.2). So, a lot of the text listed under the "5 - Discussion" header does not really read like a typical discussion section, in the sense that there is very little confrontation of the present study results with the existing literature. The authors should rectify that as much as possible.

**Answer: Completed.** In the revised structure of the manuscript, Table 3 (i.e., revised Table 2) and Figure 7 now appear in the "4 Results" section. Furthermore, to confront our results with existing literature, we added the following material:

**Section 4.2:**

"The regional amount-weighted mean  $\delta_P$  is -10.2‰ for  $\delta^{18}O$  and -68‰ for  $\delta^{2}H$  (calculated from the IRRES database for the year 2016). The latter compares well with the GNIP database long-term Ottawa amount-weighted mean (-10.9‰ for  $\delta^{18}O$  and -75‰ for  $\delta^{2}H$ ) (IAEA/WMO, 2018)."

**Section 5.1:**

"The total flood-water volume summed to  $4.82 \times 10^6$  m3 (for scenario A), which is nearly equal to the lake initial volume (i.e.,  $4.70 \times 10^6$  m3). Similar results were obtained by (Falcone, 2007) who studied the hydrological processes influencing the water balance of lakes in the Peace-Athabasca Delta, Alberta (Canada) using water isotope tracers. They reported that a springtime freshet (in 2003) did replenish the flooded lakes from 68% to >100% (88% in average)."

**Section 5.2:**

We opted to compare the resilience/sensitivity of Lake A to other lakes in similar morpho-climatic contexts. Hence, added the results from Arnoux et al., (2017a) to the revised Figure 8.

**RC2-6a.** \*\* Subsection 5.3 is a bit confusing. The authors provided a list of physical water quality parameters + ions earlier in their manuscript. Based on their introduction, I expected those physical water quality parameters + ions to be used to support the "surface water pollution" aspects of the manuscript. However, in subsection 5.3, there is no reference to those parameters/ions, and the assessment of resilience to water pollution is solely based on mean flushing time. Why were the parameters/ions described earlier not used?

**Answer: Clarification.** It was initially deemed preferable not to discuss the physico-chemical parameters and the major ion data. This decision was made in order to limit the manuscript length. However, we agree that our work could benefit from discussing the geochemical data.

First, we propose to compare the geochemical signature of Lake A to the one of Lake DM, Lake B and regional groundwater (i.e., observed at piezometers upstream of Lake B). Below is the proposed additional interpretation which was added to section "4.1 Water fluxes and isotopic framework" (with revised Figure 5):

"The geochemical facies of Lake A and Lake DM samples are illustrated in Figure 6 by the means of a Piper diagram. Mean values for Lake B and regional groundwater (GW) geochemical facies are also plotted for comparison purpose. Both Lake A and flood-water were found to be Ca-HCO3 types, which is typical for precipitation- and snowmelt-dominated waters (Clark, 2015). The geochemistry of Lake A is relatively constant throughout the year and reveals a depth-wise homogeneity. The geochemistry of Lake B is significantly distinct from Lake A and appears to be influenced by a regional groundwater characterized by a Na-Cl water type."

Then, the following discussion (in Section 5.2) was supplemented by additional geochemical considerations :

"Concerning Lake A, all studied scenarios (i.e., reference scenarios A and the sensitivity analysis) yielded values for G-Index>50% and  $t_f<1$  year, i.e., highly sensitive to groundwater changes, but resilient to surface water pollution. Nevertheless, it was shown that bank recharge, storage and discharge to lakes is crucial to correctly represent the G-Index by accounting for the origin of water fluxes (Fig.7; Sect. 5.1). While bank storage impacts the G-Index, the total water inputs (and the  $t_f$ ) remain unchanged (see orange arrow in Fig. 8). Therefore, the studied lake thus receives a reduced groundwater contribution relatively to the initial estimated apportionment when not accounting for bank storage, while it benefits from having a rapid flushing time. This implies that flood-affected lakes are more likely to be characterized by an intermediate condition and, thus, are relatively resilient to both surface water and groundwater quantity and quality changes. The geochemical data (Sect. 4.2) is in accordance with this interpretation. Indeed, a low-mineralization and Ca-HCO3 water type at Lake A is coherent with the significant flood-water contributions (to the lake and aquifer). In comparison, the neighboring lake (i.e., Lake B) does not undergo yearly recurrent flooding and was shown to be more mineralized with a Na-Cl water type, likely originating from road-salt contamination of regional groundwater (Pazouki et al., 2016). Biehler et al. (2020) similarly reported hydrological control on the geochemistry of a shallow aquifer in an hyporheic zone, where river stage influenced the mixing ratio between river water and the deeper aquifer."

"In such a case, the geochemistry of Lake A could potentially shift towards that of Lake B, and an increase of the salinity and in the concentration of  $Na^+$ ,  $Ca^{2+}$ ,  $SO_4^{2+}$  and  $Cl^-$  would be expected for Lake A."

Note that this issue was also highlighted by another reviewer (see SC1-12c).

**RC2-6b.** And is it adequate to use the mean flushing time as a proxy measure for a lake's resilience to "all" surface water pollution, regardless of the reactivity/sorption coefficients of the chemical determinants under consideration? This last question may be out of scope for the present manuscript, but a clarification sentence would help manage readers' expectations.

**Answer: Clarification.** Reviewer 2 is correct; the fate of the contaminant is also to be considered when assessing for the sensitivity/resilience of a lake to a specific surface water pollution. In the submitted version of the manuscript, we simply aimed at demonstrating a broader scenario by depicting the mean flushing time by groundwater, i.e., a key parameter for the resilience to surface water pollution. This reflects the global sensitivity of a water body and is to be adapted for each specific contaminant. That being said, a brief clarification is needed. Below is the added additional material:

"[Arnoux et a., (2017a)] proposed an interpretation framework which relates the response time of a lake to changes in groundwater and/or surface water quantity and/or quality thereby linking the G-Index with  $t_f$  (Fig. 8). They depict a general case, applicable to surface water pollutions in general, regardless of reactivity or

**fate of contaminants. Hence, care should be taken when interpreting the sensitivity to specific contaminants which are subject to attenuation processes, such as degradation and sorption."**

**RC2-7.\*\*** The first sentence of the conclusion states reiterates that the "goal" of the present study was to "develop a volume-dependent transient isotopic mass balance model, assuming well-mixed conditions, in order to better understand the dynamics of the hydrological processes at a flood-affected lake in southern Canada". As commented upon above, I find this to be rather vague. After reading through the details of the manuscript, it seems like the authors specifically want to address questions related to the relative importance of groundwater for Lake A on an aggregated annual scale as well as through different seasonal/wetness conditions (what they refer to as temporal variability of hydrological processes). The authors also dedicated a fair amount of time/manuscript space to discuss many different elements, e.g.: the peculiarities of lake dynamics under flooding conditions, uncertainties associated with their isotope mass balance model (those uncertainties are multiple in nature, i.e., input data uncertainty, structural data uncertainty, output data uncertainty, even maybe model parameter uncertainty), and the application of pollution resilience assessment framework. It is quite difficult, from the whole manuscript, to figure out which of those elements are primary versus secondary targets/goals/objectives of the manuscript and how they relate (or not) to one another. I think that there is a nice science story in the manuscript, and I hope the authors will see my comments as suggestions for strengthening it and making it interesting to the broad readership of HESS.

**Answer: Completed.** We made an effort at addressing this issue based on the comments and suggestions above. See response to comment RC2-2 and RC2-4 for clarifications concerning the general and specific objectives and a revised structure of the manuscript. The beginning of the conclusion (where we reiterate the goal of the study) was reformulated accordingly and can now read as:

"In this study, we demonstrated application of isotopic mass balance to flood-affected lakes. A volumedependent transient isotopic mass balance model was developed and applied to a flood-affected lake in an ungauged basin in southern Quebec (Canada). This allowed for better understanding of the resilience of a flood-affected lake to changes in the surface/groundwater water balance partition, to understand the role of flood-water, and to predict resilience of groundwater quantity and quality for a local water supply."

\* SPECIFIC COMMENTS ABOUT SOME TEXT SECTIONS OR FIGURES

See "sticky notes" and yellow highlights in the pdf proofs **Answer:** See below.

**\* TYPOS AND EDITORIAL SUGGESTIONS**

See "sticky notes" and yellow highlights in the pdf proofs **Answer:** See below.

Please also note the supplement to this comment:

https://www.hydrol-earth-syst-sci-discuss.net/hess-2020-101/hess-2020-101-RC2-

supplement.pdf

**RC2-8.** L24: Refer to "Lake DM". What is this? Not previously defined in the abstract **Answer: Done.** Reviewer 2 is correct; it must be defined first. It should be written "Lake Deux-Montagnes (DM)". This was corrected.

**RC2-9.** L36-37: Refer to "Lerner and Harris, 2009;Cunha et al.,2016;Scanlon et al., 2005". Space missing **Answer: Done.** We used the *EndNote® Output Style File* from Copernicus (downloaded from https://www.hydrology-and-earth-system-sciences.net/for\_authors/manuscript\_preparation.html). We corrected the downloaded referencing style by changing the multiple citation separator to '; ' (i.e., semi-colon and space). All the references were automatically updated.

**RC2-10.** L46: Spaces missing in-between successive in-text citations. That tends to happen throughout the text and needs to be rectified

Answer: Done. See response to comment RC2-9 (above).

**RC2-11.** L56: Refer to "long". Do the authors mean "long" or "wide:? **Answer: Done.** We meant 1 km "wide" area. This was corrected in the manuscript.

**RC2-12.** L59: Refer to "connectivity". Connectivity between what and what? **Answer: Done.** We meant the "hydraulic connectivity between Lake A and Lake DM". This is now specified.

**RC2-13.** L89: What does "it" refer to, here? **Answer: Done.** We refer to S1. This is now specified.

**RC2-14.** L96: Refer to "(Deux-Montagnes)". This should probably have been specified earlier, i.e., the first time that "Lake DM" is mentioned in this section (see previous page) **Answer: Done.** Reviewer 2 is correct. It is now specified at L91.

**RC2-15.** L97: "...drains via the St. Lawrence River..." By using the term "via", do the authors mean "to" or "toward" the St. Lawrence River?

Answer: Done. We meant "to" and corrected the revised manuscript.

**RC2-16.** L99-100: Refer to "...it is likely that no or very limited subsurface hydraulic connection between Lake A and Lake DM exist." The authors likely need to expand on this hypothesis a bit. Has this been verified in the field, or is it an assumption/hypothesis solely based on surficial deposits information?

**Answer: Done with clarifications.** Lake A and Lake B were created by the sand dredging activities. This sand deposit was described by Ageos (2010) as a buried valley, which extends in the NE-SW direction and was carved into the Champlain Sea clay. We added a geological cross-section in the revised Figure 1. Also, we added the following material to section 2.1:

"While alluvial sands were mapped in the area between Lake A and Lake DM (Figure 1b), stratigraphic data (i.e., well logs) confirms that only a thin layer (few centimeters to roughly 2 meters) of alluvial sands are deposited on top the clayey sediments in the area between Lake A and Lake DM (see Figure 1c). Hence, it is likely that little or no subsurface hydraulic connection exists between Lake A and Lake DM."

**RC2-17.** L151: Refer to "... August 17,2017..." Space missing. **Answer: Done.** Indeed, there was a space missing.

**RC2-18.** L153: Physico-chemical parameters and ions do not seem to have been used at all by the authors, i.e., they are not presented in any result table or figure. Why are the sampling procedures related to them presented here, then?

Answer: See response to comment RC2-6a and RC2-6b.

RC2-19. L164: Refer to "Stable isotopes of water". Oxygen and hydrogen.

**Answer: Done.** We are grateful to Reviewer 2 for pointing that out. It is indeed more appropriate to use "stable isotopes of oxygen and hydrogen". This was also corrected at L304 (see comment RC2-23b).

**RC2-20.** L271: There seem to be a lot of "results" that are listed in section 3, which most readers would equate to a Methods section (and not a results section). I would suggest that the authors try to reorganize their text a bit, so that section 3 only focuses on methods while section 4 summarizes results **Answer:** See response to comment RC2-6a and RC2-6b.

**RC2-21.** L281: Refer to "are". Grammar issue. Subject singular subject, verb plural **Answer: Done.** It was corrected to "is".

**RC2-22.** L282: Refer to "They". Flood-water samples?

Answer: Done. We specified "The flood-water samples" instead of "They".

**RC2-23a.** L304: Refer to "Deciphering surface and groundwater inputs". Was that a specific research objective/question for the present study?

**Answer: Done.** No, it was a working step which was necessary to achieve the main and specific objectives. As we made efforts at revising the structure of the manuscript, we opted to merge the data illustrated in the original Figure 4 with the revised Figure 6. By doing so, the original section "4.1 Deciphering surface and groundwater inputs via stable isotopes of water" was incorporated in the revised section "4.3.1 Volume dependent isotopic mass balance model".

**RC2-23b.** Refer to "stable isotopes of water". I know that this is a phrase that is used a lot (including by myself, sometimes, mistakingly) but I suggest that the authors rephrase it, as often as is appropriate, in their manuscript. Technically, water does not have any isotopes, but oxygen and hydrogen do. Answer: Done. See response to comment RC2-19.

**RC2-24.** L322: Refer to "...isotopic composition Lake A...". Change to "...isotopic composition of Lake A..." Answer: Done. It was corrected.

**RC2-25.** L329: Refer to "Quantification of flood-water inputs into Lake A". Is that another specific research objective/question in the present study?

**Answer: Done.** The evaluation of the water budget corresponds to a working step in order to achieve the main and specific objectives. Please see response to comment RC2-2.

**RC2-26.** L335: Refer to "...and daily net...". Change to "... and the daily net...". Answer: Done. It was corrected.

**RC2-27.** L340: Refer to "On early August..." Change to "In early August..." Answer: Done. It was corrected.

**RC2-28.** L385: Refer to "Sensitivity analysis". Why was this done and how novel (i.e., different from what others have done) is this? Is that another specific research question/objective targeted by the present study? **Answer: Clarifications.** The sensitivity analysis was not a specific objective, but it was needed to grasp the relative

impact of the input parameter's uncertainties on the model outputs. In order words, our objective was to assess the reliability of the model outputs against a range of possible input values. As the model outputs remained comparable to the reference scenario, we concluded that the model was representative of the local hydrological processes.

In order to limit the length of the manuscript, we placed all the sensitivity analysis results (i.e., the tables for scenarios A and B) in the Appendix D. Note that the results of the sensitivity analysis (for scenario A) are also depicted in the revised Figure 7 and Figure 8.

**RC2-29.** L411: Refer to "Negligible". Change to "A negligible". **Answer: Done.** It was corrected.

**RC2-30.** L413: Refer to "...the value...". Change to "...the values..." Answer: Done. It was corrected.

**RC2-31.** L416: Refer to "...only small...". Change to "...only a small..." Answer: Done. It was corrected.

**RC2-32.** L420: Refer to "Importance of groundwater on the annual lake budget". So the reader should assume that quantifying this was a major goal of the present study?

**Answer: Clarifications.** Efforts were made at reviewing the structure of the manuscript. Hence, the sub-section was reformulated (see RC2-2). Evaluating the importance of flood-water inputs (and bank storage) on the annual water budget was a specific objective of the study.

**RC2-33.** L438: Refer to "Temporal variability of the hydrological processes". Specific research objective of this study? **Answer: Clarifications.** Analyzing the temporal variability of the groundwater inputs was a working step in order to achieve the main and specific objectives (see RC2-2). We think it was important to underline the temporal variability of the hydrological processes, because the flood events generally occur during a specific time of the year.

**RC2-34.** L464: Refer to "...Flood-water...". Change to "...flood-water..." Answer: Done. It was corrected.

**RC2-35.** L469: Refer to "...that water quality...". Change to "...that the water quality..." Answer: Done. It was corrected.

**RC2-36.** L483: Refer to "Resilience". Since the word resilience can have very different meaning in different sub-fields or sub-disciplines of ecology, hydrology and ecohydrology, I strongly suggest that the authors provide their adopted definition for that term.

**Answer: Done.** We are grateful to Reviewer 2 for this suggestion. We opted to add the following material in the revised section 5.2:

"Resilience of a system has been defined as its capacity to cope with perturbations (i.e., internal and/or external changes) while maintaining its state (Cumming et al., 2005). In the case of a lake, perturbations can manifest as a change in the water quantity and quality contributing to the water balance. According to Arnoux et al. (2017a), the impact of a perturbation to a lake is not only dependent on the relative importance of water budget fluxes, but also on the residence time of water in the lake."

RC2-37. L495: About the x-axis label: should we read tG or tf? I am a bit confused...

**Answer: Done.** This was also pointed out by the other reviewers. Indeed, it was corrected to tf (the mean flushing time by groundwater).

**RC2-38.** L499: Refer to "Arnoux et al, 2017a". This likely warrants more explanation in the text, so that the readers can get a good idea of where that representation/framework is coming from and what its underlying rationale is without having to go back to the 2017 paper.

**Answer: Clarifications.** Arnoux et al. (2017) performed isotopic mass balances over of 21 kettle lakes in Quebec (Canada) and speculated about their response to a perturbation by comparing the G-index and Tf. Considering these two indices, they proposed an interpretative framework to discuss the resilience of the lakes to surface water and groundwater changes, depicted on a plot of G-Index vs 1/Tf, as we did in Figure 9.

To avoid any confusion, we added a description of the interpretative framework developed by Arnoux et al. (2017) and summarized their results. This additionally allowed to compare our results to the ones of Arnoux et al. (2017) in Figure 9 and discuss the similarities/differences and implications (see response to comment RC2-5 for the proposed additional material).

**RC2-39.** L500-509: This paragraph comes a bit out of nowhere. It should be better linked with previous text, as well as specific research questions or objectives (which are currently missing from the manuscript) **Answer: Done.** We added the following material:

"Considering the above, it is possible to speculate about the potential future impacts of climate change on Lake A."

Additionally, we proposed a discussion concerning the impact of the flood-water inputs on the geochemistry of Lake A and the potential evolution in front of climate changes (see response to comment RC2-6a).

**RC2-40.** Figure B1: The caption refers to blue hollow symbols and a solid blue line by my version of the manuscript includes a black and white figure, not a color figure.

**Answer: Done.** The caption and the figure were both revised. We added color on the figure in order to facilitate the reading.

Answer: Done. The suggested reference was added.

SC1-3. Line 56: "...but occur over a 1 km long area." Do you mean 1 km "wide" area? The length of the river or canal is of no importance, it is the width that will make it hard to measure flux. Please change to "wide".Answer: Done. The reviewer is right. We meant 1 km "wide" area. This was corrected in the manuscript.

**SC1-4.** Line 58: "The democratization of isotope mass balances in Quebec..." What does the "democratization" of isotope mass balance mean? Was this auto corrected from the original word to be used. I hope so, as I had no idea that isotopes were political! Should the word be "demonstration"? Not really sure what is going on here.

**Answer: Clarification.** This is a concept we translated from French, but the Reviewer's comment made us realize that the meaning is not accurate in English. We wanted to convey the fact that isotopic methods are not widely used in Quebec (Canada) yet and that we could benefit from their application. As this is only a local implication, we opt to withdraw this sentence from the manuscript. In replacement, we opted for the following:

"...isotopic approach was efficient for characterizing the impacts of floods and droughts, and that a broad application can contribute to water resources management in providing information to understand the vulnerability of ungauged systems."

**SC1-5.** Line 60-70: It would be good to include Herczeg et al (2003) here as well as they determined changes in isotopic composition due to groundwater pumping, this also shows how transient changes can affect the isotopic composition of lakes.

**Answer: Done.** This publication is indeed very interesting and is related to our study. Herczeg et al. (2003) demonstrated the impact of various forcings, such as the rainfall variability, land-use changes and increased pumping rates, on the water residence time in the Blue Lake, in Australia. Note that the pumping corresponds to direct surface water abstraction (Lamontagne and Herczeg, 2002), and not groundwater pumping. Nonetheless, Herczeg et al. (2003) showed the importance of studying lake water budgets in order to identify potential governing forcing in order to secure water quantity and quality overtime, as lakes are important water resources for the production of drinking water. In that sense, the following material was added to the introduction:

"...and thus secure water quantity and quality over time for drinking water production purposes (Herczeg et al., 2003)."

**SC1-6.** Line 78: There is no hypothesis indicated in this manuscript. The objectives are clear but there is no indication of what mechanisms they propose may be important. A hypothesis should be added. **Answer: Done.** We added the hypothesis in the introduction (after the main and specific objectives):

"It is hypothesized that the groundwater fluxes (inputs and outputs) through lake banks are unneglectable in lake water budgets, even for flood-affected lakes."

Additionally, we added a new sub-section (2.2 Conceptualization of the groundwater-surface water interactions) to better illustrate the conceptual model (illustrated in the revised Figure 2). In this section, we briefly present the conceptual hypotheses underlying the model development are explained in this section (i.e., 2.2) and can read as:

"During the flood-water input period, we hypothesize that the surface water inputs (IS) and precipitations (P) represent the total water inputs to Lake A (Fig. 2a). High-water levels at Lake A impose a hydraulic gradient at the lake-aquifer interface which inhibits groundwater inflows (IG). Contrastingly, it is assumed that IG constitutes the main water input to Lake A during the normal periods, while IS is neglectable (Fig. 2b). In fact, as the flood-water inputs stop, the water level at Lake A lowers and the hydraulic gradient at the lake-aquifer interface is reversed and allows for IG to flow to the lake."

**SC1-7a.** Figure 1. Water courses shown don't match up with the description. There is supposed to be one inlet and outlet to Lake A, but at least two inlets are shown (or outlets). Flow directions are needed on the other streams (canals?) shown.

**Answer: Done.** A description of the or canal (S3) in the southwestern part of Figure 1b was added to sub-section 2.1. As the topography in the study area is nearly flat, the flow direction can evolve according to Lake DM elevation, similarly to S2. We proposed to correct Line 91-92 to:

"Two channelized outlet streams (S2 and S3) allow water to exit Lake A and flow towards Lake DM. The flow direction at S2 and S3 can be temporally reversed (Fig. 1b) when the water level of Lake DM is above the topographic threshold of 22.12 m.a.s.l. (Ageos, 2010)."

SC1-7b. The reorienting of the North arrow is somewhat confusing, probably needed.

**Answer: Done.** We agree with the reviewer, but the original Figure 1c no longer appears in the revised version of Figure 1 (due to further modifications).

**SC1-7c.** All sampling is done in one corner of the lake, why was this done? Presumably the lake is well mixed? Samples taken near the shore, like LB-S1 could have some evaporation signature in them. Was this accounted for? **Answer: Clarification.** Lake A is an actively mined sand pit and access to the Lake was limited to this area due to logistical and safety issues. In this study, the horizontal variability of the waterbody was not assessed. However, Pazouki et al. (2016) showed that temperature, dissolved oxygen, pH and turbidity are very similar along four 10-meter vertical profiles (one in each corner) in Lake A. We added the following material to section 3.2:

"Additional field campaigns were conducted on February 9, 2017, August 17, 2017 and January 25, 2018 in order to perform vertical profile measurements and water sampling at various depths (e.g. 2 m, 4 m, 8 m, 12 m and 15 m) at LA-P1 to LA-P4. Lake water sampling was performed in the northern part of the lake for logistical reasons and due to ease of accessibility. As horizontal homogeneity has been previously demonstrated by Pazouki et al. (2016), the water samples were deemed representative of the whole waterbody."

**SC1-7d.** If Lake DM really has a name (Deux-Montagnes), then the name should be put on the map with the (DM) in parentheses. One might also argue to use the French for the whole name, Lac des Deux Montagnes. Lake B also appears to be called Lac Val des Sables in google earth, is this not correct?

**Answer: Moot.** The Reviewer is correct; Lake B is called "Lac Val-des-Sables". The names "Lake A", "Lake B" and "Lake DM" were used to keep the location of the study site as anonymous as possible, due to an agreement with the Town. However, this comment clearly shows that it is easy to figure it out. Still, we prefer to use the names "Lake A", "Lake B" and "Lake B" and "Lake DM" to facilitate the reading.

Considering all the above (i.e., SC1-7a to SCI1-7d) and other reviewer's comments, a revised version of Figure 1was added to the manuscript.

**SC1-8.** Line 116: "All water levels are reported relative to a reference water levels measured on February 9, 2017" One reference water level or many? Please fix, this is a combination of both.

**Answer: Done.** We meant that there is a reference water level (z=0) for each well starting on February 9, 2017. In order to avoid any confusion and considering comment SC1-10, we opted to revise Figure 2 and to simply illustrate the water level evolution relatively to a known common datum (i.e., meters above sea level).

**SC1-9.** Line 118 change to "...over the Ottawa River watershed..." Answer: Done. It was corrected.

**SC1-10.** Line 125-126: "...synchronous with those of Lake DM (Fig. 2) from late February to late July 2017". How do you know water levels are synchronous from Late February when water level measurements weren't begun until April? This can't be known. Given the sparse data in figure 2, and the non-synchronous relation between the observation well and Lake DM in the autumn, this can't be conclusively known. In addition, some of the well peaks appear to actually occur before the lake level rises, which is a bit strange. In any case, more information is needed to be able to say this. It may be true for the flood period, but that would be expected. The low flow period doesn't appear to be completely synchronous. While it may be true that Lake DM controls Lake A water level during flood periods and/or high water periods, there is no data presented that shows that Lake A water levels are synchronous with Lake DM during low flow or low water levels. Clearly the groundwater is not synchronous during September to November.

**Answer: Clarification and done.** A hydraulic connection between Lake DM and Lake A only occurs when a topographic threshold (at 22.12 m.a.s.l.) is exceeded. During autumn, the water levels of Lake DM and VP are below this threshold, which explains the discrepancy between Lake DM and VP water levels during this period. This information is not explicitly shown on the original version of Figure 2. Considering the above and comment SC1-8, we revised Figure 2 (see response to comment SC1-8) in which the water levels are now illustrated relatively to a known common datum (i.e., meters above sea level). By doing so, it is possible to graphically represent the periods of hydraulic connection between Lake A and Lake DM. In our opinion, this will facilitate the reading and help understanding of the hydrodynamic context. Furthermore, we added four manual measurements of the water level at Lake A. These observations confirm that the water level of Lake DM is not governing the one of Lake A during autumn (September to November).

Note that the absolute water level at the pumping well P5 is not known (missing information concerning the positioning of the Town's pressure logger). The water level at P5 could thus not be represented on the revised version of Figure 2.

**SC1-11.** Section 3.1 Field measurements section. There is no mention of calibration for the water level loggers. Without calibration how do you know they were synchronous or that they water levels were the same? Please give all the calibrations that were done on instruments and isotopic analyses.

Answer: Done. The following material was added to complement original Line 140.

"All the level loggers' clocks were synchronized with the computer's clock when launching automatic measurements for a 3-month period. This procedure was done via the Diver-Office 2018.2 software. Manual measurements of the water level were regularly performed to calibrate (relatively to a reference datum) and validate the automatic water level measurements."

**SC1-12a.** Section 3.2 Water sampling and analytical techniques: Other than ion balances, was any other QA/QC done? This needs to be stated. In addition, were isotope measurements compared with standard mass spectroscopy? Ring cavity measurements have been shown to be in error in some cases and should be viewed with some skepticism unless comparison is made to standard mass spectroscopy or the methodologies listed below have been followed. See Wassaner et al (2014) and Sengupta (2014) for examples. Perhaps more detail about replicates and comparisons to mass spectroscopy measurements can be done to alleviate concerns over the accuracy of the ring cavity measurements. References: Sengupta, S., 2014, Pros and Cons of Laser Based Isotope Measurements of Water and Real Time Vapour Samples: A User's Perspective. Gond. Geol. Mag., V. 29 (1 and 2), pp.45-51 Wassenaar, L.L., Coplen T.B., Aggarwal, P.K., 2014 Approaches for Achieving Long-Term Accuracy and Precision of \_180 and \_2H for Waters Analyzed using Laser Absorption Spectrometers. Environ. Sci. Technol. 2014, 48, 2, 1123-1131.

**Answer: Clarification and done**. We agree that it is worth to provide more details on the analytical procedures. Lines 166-168 were corrected to:

"1 ml of water was pipetted in a 2ml vial and closed with a septum cap. Each sample was injected (1 microliter) and measured 10 times. The first 2 injections of each sample were rejected to limit memory effects. Three internal reference waters ( $\delta^{18}O=0.23\pm0.06\%$ ,  $-13.74\pm0.07\%$  &  $-20.35\pm0.10\%$ ;  $\delta^{2}H=1.28\pm0.27\%$ , - $98.89\pm1,12\%$  &  $-155.66\pm0.69\%$ ;  $\delta^{17}O=0.03\pm0.04\%$ ,  $-7.32\pm0.06\%$  &  $-10.80\pm0.06\%$ ) were used to normalize the results on the VSMOW-SLAP scale. A 4th reference water ( $\delta^{18}O=-4.31\pm0.08\%$ ;  $\delta^{2}H=-25.19\pm0.83\%$ ;  $\delta^{17}O=-2.31\pm0.04\%$ ) was analyzed as an unknown to assess the exactness of the normalization. The overall analytical uncertainty (1  $\sigma$ ) is better than  $\pm0.1\%$  for  $\delta^{18}O$ ,  $\pm1.0\%$  for  $\delta^{2}H$  and  $\pm0.1\%$  for  $\delta^{17}O$ . This uncertainty is based on the long-term measurement of the 4th reference water and does not include the homogeneity nor the representativity of the sample (Light stable isotope geochemistry laboratory of Geotop-Uqam)."

**SC1-12b.** Also, were any samples taken under ice? Ice will fractionate the isotopic composition and make the mass balance different. Has this been accounted for?

**Answer: Clarification**. We opted to neglect the ice fractionation in the isotopic model, because assumptions concerning the forming (or melting) rate and isotopic signature of the ice would have been needed. To support this assumption, we calculated the isotopic signature of the residual water (i.e., lake water), which is described by:

$$\delta \approx \delta_0 + \varepsilon \ln (f_{residual})$$

where  $\delta_0$  is the initial water isotopic signature,  $\epsilon$  is the ice-water isotopic separation factor and  $f_{residual}$  is the residual water fraction.

If the ice thickness is 0.35 m (observed on March 4, 2019) over a surface of 2.79 x  $10^5$  m2 and the lake volume is 4.7 x  $10^6$  m3, the ice volume would be 8.7 x  $10^4$  m3 and f would be 0.98. Considering an ice-water isotopic separation factor ( $\epsilon$ ) of 3.1 for  $\delta^{18}$ O and 19.3 for  $\delta^{2}$ H (O'Neil, 1968) and a  $\delta_0$  of -10‰ for  $\delta^{18}$ O and -71 ‰ for  $\delta^{2}$ H, the isotopic signature of the residual water ( $\delta$ ) would be -10.06 ‰ for  $\delta^{18}$ O and -71.39 ‰ for  $\delta^{2}$ H. Such variation falls within the

analytical uncertainty (i.e.,  $\pm 0.1$  ‰ for  $\delta^{18}$ O,  $\pm 1.0$  ‰ for  $\delta^{2}$ H). Note that a well-mixed lake is assumed for this calculation.

The following material was added to sub-section 3.3:

"In this study, the potential impacts of the ice-cover formation and melting are neglected, as the ice volume is likely to represent only a small fraction (<2%) of the entire water body. Moreover, considering the ice-water isotopic separation factor, i.e., 3.1 ‰ for  $\delta^{18}$ O and 19.3 ‰ for  $\delta^{2}$ H (O'Neil, 1968) and assuming well-mixed conditions, the lake water isotopic variation would be comprised within the analytical uncertainty. Also, floodwater inputs from Lake DM were expected to be much more important and occurring simultaneously with icemelt during the freshet period."

**SC1-12c.** It also isn't clear why methods are included for water quality sampling. These data don't appear to have been used in this manuscript, so it simply takes up space. Please remove the methods for chemical sampling and concentrate only on the isotopic measurements.

**Answer: Clarification.** It was initially deemed preferable not to discuss the physico-chemical parameters and the major ion data. This decision was made in order to limit the manuscript length. However, we agree that our work could benefit from discussing the geochemical data.

First, we proposed to compare the geochemical signature of Lake A to the one of Lake DM, Lake B and regional groundwater (i.e., observed at piezometers upstream of Lake B). Below is the proposed additional interpretation which was added to section "4.1 Water fluxes and isotopic framework" (with revised Figure 5):

"The geochemical facies of Lake A and Lake DM samples are illustrated in Fig. 5 by the means of a Piper diagram. Mean values for Lake B and regional groundwater (GW) geochemical facies are also plotted for comparison purpose. Both Lake A and flood-water were found to be Ca-HCO3 types, which is typical for precipitation- and snowmelt-dominated waters (Clark, 2015). The geochemistry of Lake A is relatively constant throughout the year and reveals a depth-wise homogeneity. The geochemistry of Lake B is significantly distinct from Lake A and appears to be influenced by a regional groundwater characterized by a Na-Cl water type."

Then, the following discussion (in Section 5.2) was supplemented by additional geochemical considerations :

"Concerning Lake A, all studied scenarios (i.e., reference scenarios A and the sensitivity analysis) yielded values for G-Index>50% and  $t_f<1$  year, i.e., highly sensitive to groundwater changes, but resilient to surface water pollution. Nevertheless, it was shown that bank recharge, storage and discharge to lakes is crucial to correctly represent the G-Index by accounting for the origin of water fluxes (Fig.7; Sect. 5.1). While bank storage impacts the G-Index, the total water inputs (and the  $t_f$ ) remain unchanged (see orange arrow in Fig. 8). Therefore, the studied lake thus receives a reduced groundwater contribution relatively to the initial estimated apportionment when not accounting for bank storage, while it benefits from having a rapid flushing time. This implies that flood-affected lakes are more likely to be characterized by an intermediate condition and, thus, are relatively resilient to both surface water and groundwater quantity and quality changes. The geochemical data (Sect. 4.2) is in accordance with this interpretation. Indeed, a low-mineralization and Ca-HCO3 water type at Lake A is coherent with the significant flood-water contributions (to the lake and aquifer). In comparison, the neighboring lake (i.e., Lake B) does not undergo yearly recurrent flooding and was shown to be more mineralized with a Na-Cl water type, likely originating from road-salt contamination of regional groundwater (Pazouki et al., 2016). Biehler et al. (2020) similarly reported hydrological control on the geochemistry of a shallow aquifer in an hyporheic zone, where river stage influenced the mixing ratio between river water and the deeper aquifer."

"In such a case, the geochemistry of Lake A could potentially shift towards that of Lake B, and an increase of the salinity and in the concentration of  $Na^+$ ,  $Ca^{2+}$ ,  $SO_4^{2+}$  and  $Cl^-$  would be expected for Lake A."

Note that this issue was also highlighted by Reviewer 2 (see RC2-6a).

**SC1-13.** Line 170: "The water and stable isotope mass balance of a well-mixed lake can be described..." The authors haven't actually demonstrated that the lake is well mixed. A figure showing the lake profiles should be presented. **Answer: Done.** We added a justification for the use of a well-mixed model in sub-section 3.3:

"Stable isotope mass balances can either be performed based on (i) a well-mixed single layer model or (ii) a depth resolved multi-layered model. In a recent study, Arnoux et al. (2017b) compared a well-mixed model and a depth-resolved multi-layer model. Both models yielded similar results and provided a general understanding of the groundwater-surface water interactions. The multi-layer model additionally allowed for the determination of groundwater flow with depth, but required a temporally- and depth-resolved sampling in order to ensure a thorough understanding of the stability/mixing of the different layers. Such important sampling and monitoring efforts are however often unrealistic in remote and/or flood-affected contexts. Additionally, Gibson et al. (2017) studied the impact of sampling strategies on the water yield (i.e., the depthequivalent runoff to the lake) estimations for the Turkey Lake (32 m deep) under stratified and well-mixed conditions. They reported 18% difference on the water yield when performing grab sampling (i.e., 1 sample at 1 m depth) and bulk sampling (i.e., assessment of the whole lake water column). The difference was less important (i.e., 11%) when comparing bulk sampling to integrated sampling for epilimnion, metalimnion and hypolimnion. They also reported discrepancies up to 20% for the water yield estimations at the same lake according to the timing of the lake water sampling. This last result shows that temporal shifts may induce greater bias than the uncertainty related to the lake stratification. For these reasons, we advocated the application of a well-mixed model."

The isotopic composition of the lake water at different depth are illustrated in the revised Figure 6. While the exact depth of each sample is not specified, we marked the  $\leq 2$  m depth vs the >2 m depth samples. It shows that the lake is well mixed on February 2017 and January 2018, and that a stratification developed during summertime. Based on the results of Gibson et al. (2017), it is appropriate to fit the modeled isotopic signature of the lake  $(\delta_L)$  on the depth-average observed  $\delta_L$  in order to take into account the stratification. Fitting the modeled  $\delta_L$  on samples from the epilimnion (i.e.,  $\leq 2$  m depth) would have been misleading.

**SC1-14.** Line 198: So, evaporation was held constant for the entire month. Particularly in the spring, that is a brave assumption. This seems to be the coarsest time step. Why was this needed?

**Answer: Clarification**. In the model, the evaporation is specified at a daily time step. At Line 198, we are referring to the input parameters (i.e.,  $\delta_P$ ,  $\epsilon^+$ ,  $\alpha^+$  and  $\epsilon_K$ ) for the calculation of the atmospheric moisture ( $\delta_A$ ). In this calculation, the  $\delta_P$ ,  $\epsilon^+$ ,  $\alpha^+$  and  $\epsilon_K$  are evaporation flux-weighted. Given daily evaporation rate time series,  $\delta_A$  can only be estimated at a monthly time step. For more details, see Gibson et al. (2015).

**SC1-15.** Line 208: limiting isotopic composition (Gibson et al., 2015). This is not a common term. Although this can be found in the reference listed, it should be detailed more here.

**Answer: Done**. Gibson et al. (2015) states that " $\delta^*$  is the isotopic composition that a desiccating water body would approach under non-steady-state conditions as it dries up (i.e.  $V \rightarrow 0$ )."

It was corrected to:

"... $\delta^*$  is the isotopic composition that the lake would approach as V  $\rightarrow$  0 (Gibson et al., 2015)."

Note that we opted to place this calculation in Appendix A in the revised manuscript following other reviewers' comments.

**SC1-16.** Line 216: "The above-mentioned equations are computed on a daily time step to calculate the isotopic composition of the lake (\_L)." Yet, some parameters have monthly time steps. How do you reconcile that? Does this

mean the monthly time steps aren't that important, or should it all be done monthly? This seems like a limitation to the daily time step.

Answer: Clarification. Please see response to comment SC1-14.

**SC1-17.** Line 218: It has been stated a few times that the lake is well mixed, but this has not been demonstrated with any measurements. The reader needs evidence that the lake is well mixed, particularly over the time period of measurement, which is over the springtime period, when mixing may not be complete.

**Answer: Clarification**. Please see response to comment SC1-13.

**SC1-18.** Line 223: "Assuming homogenous hydraulic conductivity of the sediments" This is a big assumption and likely not accurate overall, but in a sandy aquifer, might be a reasonable assumption given other errors in the model. This should be explained more.

**Answer: Clarification**. We are grateful to the reviewer for questioning this point. As it was pointed out, this assumption might be a reasonable in the context of this study, but care should be taken when stating so. This comment made us reanalyze our reasoning leading to this assumption. We did not state the correct hypothesis. In the context of our study, we should simply hypothesize the following:

"The outflows from the lake are thus roughly proportional to the lake water level, ..."

**SC1-19.** Line 259 and 263: This is a pet peeve of mine, but "since' is a time word and shouldn't be used to replace "because", please change to "because" everywhere in the manuscript when it is not used as a temporal term. **Answer: Done.** It was corrected.

**SC1-20.** Line 269: change to "...lead to overestimation...:" **Answer: Done**. It was corrected.

**SC1-21.** Line 270, so is the potential underestimation of groundwater exchange underestimated here? Or was something done to account for this. Please explain.

**Answer: Clarification**. A sensitivity analysis was performed over the evaporative fluxes (E) to address this point specifically (i.e., the overestimation of E, leading to a potential underestimation of the groundwater exchange). When considering E - 20%, the model yields to total annual outputs of  $1.44 \times 10^7 \text{ m}^3$ , while the reference scenario yields  $1.72 \times 10^7 \text{ m}^3$ . This translates to a 16% decrease of the total annual outputs (and inputs). The evaporation represents roughly 2% of the total outputs in both scenarios. The remaining 98% corresponds to the total groundwater and surface water outputs (Q). Also, when considering E - 20%, the surface water inputs (Is) correspond to 68%, which remains very similar to the partition of the reference scenario (i.e., 71%). While E was found to be one of the most stringent parameters, the water balance partition remains similar for both scenarios over an annual basis. Hence, an overestimation of E is not misleading.

**SC1-22a.** Line 290-295: Why not just measure the GW input? Why does it need to be estimated from the intersection with the LEL?

**Answer: Clarification**. In hydrogeological contexts where groundwater-surface water interactions are important, the chemistry and isotopic signature of groundwater typically bear some local heterogeneity. Hence, the representativity from a groundwater sample can be hard to understand. In the context of this study, it was preferable to estimate  $\delta_L$  from the intersection between the lake's LEL and the LMWL, as it represents the mean isotopic signature of the local groundwater contributing to the lake.

For further details, please see response to comment RC1-2.

**SC1-22b.** Also, although the evaporation process is the same between flood water and lake water (having the same slope) that is not unusual. What is unusual is that they don't intersect at the same place, so the floodwater is a different source from the recharge from GW or rainfall.

**Answer: Clarification**. The flood-water and groundwater have different sources. The flood-water is mainly composed of springtime rainwater and snowmelt water and is originating from a large watershed which extends to the North. The isotopic signature of the flood-water was thus expected to be more depleted relatively to the groundwater. The local groundwater is conceptualized as a mixture between local precipitations and flood-water (due to the yearly recurrent flooding of the study area).

**SC1-22c.** There do appear to be five lake values (one of which appears to be unevaporated floodwater) that fall on the floodwater line, so there is some influence from floodwater on the isotopic composition of the lake. This should be address more fully.

**Answer: Clarification**. Indeed, some surface water samples from Lake A are plotting near the flood-water LEL and one might hypothesize that this is suggesting an influence of the flood-water on the isotopic composition of the lake. However, of the five Lake A ( $\leq 2$  m depth) water samples that appear to plot near the flood-water LEL, only the most depleted sample was collected during the flooding event in 2017. The other samples were taken in April 2016, June 2016, December 2017 and January 2018. Considering the timing of the sampling dates, these four surface water samples are more likely suggesting mixing between lake water and precipitations.

**SC1-23a.** Line 331-332: The authors say: "Lake A volume variations are estimated from water level records assuming a constant lake area. When not available, the surface elevation of Lake A is assumed to be equal to the water level at other observation points." I don't understand what this means. Unless this is a pit lake with perfectly straight vertical sides, the Lake area will increase as elevation increases and it will take more water for fill shallower stage heights as the lake gets bigger. Please explain if this is not true for this lake.

**Answer: Clarification**. The Reviewer is correct. Net water fluxes were calculated considering constant area (i.e., perfectly vertical banks). This assumption was made because of the relatively flat topography (outside of the lake's banks). Attempting to delineate the lake's contour with a Digital Elevation Model (DEM) would have led to unrealistic results. Therefore, we opted to neglect the surface variations. This assumption is not likely to have a significant impact on the model outputs. In fact, the lake water level variations extend over a 2.9 m range only (from 21.87 m.a.s.l. to 24.77 m.a.s.l.), which is relatively small compared to the maximum depth. Hence, the calculated lake volumes are very similar when considering 25° slopes or 90° slopes over the range of water level variations. However, for the calculation of the isotopic signature of the lake (i.e.,  $\delta_L$ ), assuming vertical banks would have led to less representative values. We assumed 25° slopes, to calculate a depth-average  $\delta_L$ . In this case, the lower depths have less impact than the shallower parts of the lake on the estimation of a depth-average  $\delta_L$ .

**SC1-23b.** Furthermore, water levels in a well cannot be used unless there is no GW flow to the lake. If the groundwater level is the same as the lake level, then there will be no flow to the lake and the flow is stagnant. Has this been observed? If not, this GW elevation should not be used as a surrogate for lake level. **Answer: Moot.** Please see response to comment SC1-24.

**SC1-24.** Figure 2 actually show that Lake A water level is at no time equal to Observation well VP, and is generally higher than the well elevation, except in late summer, suggesting the lake is losing water to the well except when precipitation slows down and the lake level lowers. Lake DM, which is a possible surrogate for Lake A elevation, is also never equal to the elevation of well VP, except on the rising limb of the floodwater. Therefore, the well VP elevation is not a good surrogate for lake A elevation and should not be used as such, unless a better explanation can be given.

**Answer: Moot**. The Reviewer is correct, there is groundwater flow between Lake A and VP. We know that the pumping wells induce a hydraulic gradient, which forces Lake A water to infiltrate the sandy bank (year-round). However, the isotopic composition of the lake  $\delta_L$  is iteratively solved at each time step and is dependent on f, which is the remaining fraction of lake water. The model is thus based on the water level difference between two time-steps (not the absolute water level). From August to November, the daily water level variations at VP are expected to be of the same range as the ones of Lake A. Moreover, the water level of VP is a better approximation than Lake

DM during the period of no hydraulic connection (i.e., from August to November). Considering the above, the observed water level at VP can be used as a surrogate for Lake A from August to November 2017.

The following material was added to section 4.3.1:

"Lake A volume variations are estimated from water level records at Lake A and assuming a constant lake area. When not available, water levels at Lake DM or observation well VP are used as proxies. Water level of Lake DM is used when there is a hydraulic connection with Lake A (i.e., above the topographical threshold) and data from observation well VP is used otherwise. These approximations were deemed acceptable because the simulation of  $\delta_L$  depends on the remaining fraction of lake water (not the absolute water level), and daily variations of the water levels at Lake A, Lake DM and observation well VP were shown to be similar (see Section 4.1)."

This issue was also addressed in response to comment RC1-8.

**SC1-25.** Line 338-340: This also a time of groundwater input (at least following the Lake DM elevation compared to Well VP). Is this considered in the fluxes?

**Answer: Clarification**. At Line 338-340, we refer to the net water fluxes, which include all inputs and outputs. During the flood period (i.e., February 23, 2017 to May 8, 2017), the high water level at Lake A was very likely to impose a hydraulic gradient towards the aquifer, which led to very limited contribution of groundwater to the lake's water balance in comparison to the floodwater inputs (from Lake DM). Hence, we developed the water balance model assuming that the groundwater inputs were null during the flooding period. Contrastingly, surface water inputs were neglected for the rest of the simulated period, while groundwater inputs were expected to play a major role in the water balance partition.

The above is now better illustrated in sub-section "2.2 Conceptualization of the groundwater-surface water interactions".

**SC1-26.** Line 359: So, here the vertical profiles are volume-weighted, which suggests the sides of the lake are not vertical, if they were then you wouldn't need to volume-weight them. But above you say you use a constant lake area to get the volume. Which is it?

Answer: Clarification. Please see response to comment SC1-23a.

**SC1-27a.** Line 382-384: you do have 3 vertical profiles; you could have at least estimated how big a difference using a stratified model using some max and min values for the isotopes.

**Answer: Clarification**. The development of a multi-layer model was beyond the scope of this study (see response to comment SC1-17).

**SC1-27b.** It also isn't clear from the discussion above this if the direction of groundwater low, in or out of the lake is considered, as the water level data suggests in changes through the modeling period. **Answer: Clarification.** Please see comment SC1-25.

**SC1-28a.** Table 2: A small point, but I'm not sure why commas are used in this table. Scientific notation usually uses a period even for large numbers. Europeans use commas for decimals and then periods for large numbers, so I'm not sure what style is being used here. I would prefer these to all be periods not commas. **Answer: Done.** Indeed, all the commas are to be replaced by periods.

**SC1-28b.** A larger point for this table is that the sensitivity analysis doesn't appear to use very wide values to check how sensitive the variables are. A change of 0.5 per mil for oxygen is not that far outside the error of the measurement. It looks like most of the differences looked at are between 10 and 20 percent. Is that reasonable, what is the variability of the rainfall amounts over time. Granted E isn't likely to have a large range, but some of the variable could have larger ranges than are estimated here.

Answer: Clarification. Sensitivity analysis aims at identifying the input parameters that most affect the robustness of a model and can help in the model parameterization, calibration, optimization, and uncertainty quantification (Song et al., 2015). Depending on the complexity of the hydrological model and the authors' objectives, different methods can be employed. In this study, a one-at-a-time (OAT) sensitivity analysis was performed to grasp the relative impact of the input parameter's uncertainties on the model outputs. In order words, our objective was to assess the reliability of the model outputs against a range of possible input values. As the model outputs remained comparable to the reference scenario, we concluded that the model was representative of the local hydrological processes. The selected range of input variables was carefully chosen. Concerning the isotopic framework, a change of  $\pm 0.5\%$  for  $\delta^{18}$ O was considered adequate to depict the potential bias introduced by sampling and analytical methods. Note that the overall analytical uncertainty (1  $\sigma$ ) is ±0.1‰ for  $\delta^{18}$ O. For the meteorological parameters, a range of ±10% was selected to represent the potential spatial variability (not temporal variability), as the data was retrieved from offsite meteorological stations. Furthermore, a range of  $\pm 20\%$  for the evaporative fluxes (E) was deemed necessary because it was calculated from a selected evaporation model (i.e., Penman-48 equation), which is dependent on numerous meteorological parameters. A comparison with two other evaporation models (i.e., Linacre-OW and openwater simplified version of Penman-48) revealed adequation between the estimations from April to August, but discrepancies during late summer and autumn.

**SC1-29.** Line 414: What about groundwater influx at this time? Ok, I see discussed in the next section. **Answer: Done**. Ok.

**SC1-30.** Line 440: Table 3 provides the relative importance of the hydrological processes for that year that was measured, not for an annual timescale. Measurements for all parameters weren't done for the whole year as well. This should be modified.

Answer: Done. It was corrected.

**SC1-31.** Line 485: tG the mean flushing time by groundwater isn't included in equation 13 and is instead written as tf, which I assume is the time of flushing (by groundwater). This needs to either be explained better, if I don't understand this, or the notation needs to be corrected. Everywhere else it is tf.

Answer: Done. Indeed, it was corrected to tf (the mean flushing time by groundwater).

**SC1-32.** Figure 9. The caption also has reference to tG is this a different variable or is it tf? **Answer: Done**. It is tf and it was corrected.

**SC1-33a.** The climate change part of this paper is somewhat of a throw away suggestion. There is really no data or simulations that support either conclusion and the modeling doesn't appear to help either. Given the possibility of either more or less flooding the conclusions seem pretty obvious. **Answer: Done.** We added the following material:

"Considering the above, it is possible to speculate about the potential future impacts of climate change on Lake A."

Additionally, we proposed a discussion concerning the impact of the flood-water inputs on the geochemistry of Lake A and the potential evolution in front of climate changes (see response to comment SC1-12c).

**SC1-33b.** While the model and the system are relatively well characterized it isn't clear what this gives other scientists other than a look at a local system. How can this be used in other lake systems and can a lake with fewer measurements or larger area or volume be characterized using this model? It would be good if some bigger questions were answered rather than just the local questions that have no real interest to scientists or the public outside of the area.

Answer: Done and clarifications. We are grateful to the reviewer for this valuable comment.

We did improve the introduction to better highlight the broad relevance of such study. The following material now appears in the introduction:

"Isotopic mass balance models are typically applied to contexts where there are no surface water inputs (Sacks et al., 2014; Arnoux et al., 2017b) and/or the surface water inputs are quantified by stream gauging (Stets et al., 2010). In remote environments, such as in northern Canada, application of isotopic methods is particularly convenient, as direct measurement of surface water inflow is difficult or nearly impossible (Turner et al., 2010; Brock et al., 2007). Recently, Haig et al. (2020) opened up new perspectives, as they reported excellent agreement between results obtained via isotopic mass balance and gauging techniques when assessing the water budget of connected lakes in Saskatchewan (Canada). They highlighted that the isotopic approach was efficient for characterizing the impacts of floods and droughts, and that a broad application can contribute to water resources management in providing information to understand the vulnerability of ungauged systems. As future climate change impacts are expected to include increases in flood magnitude and frequency (Aissia et al., 2012), flood-affected lake water budget assessments are of utmost importance.

The main objective of this study is to demonstrate the application of isotopic mass balance to flood-affected lakes, as this approach is particularly opportune in providing estimates of the water balances and insights on the dynamics of ungauged systems."

Also, to the best of our knowledge, this is the first study to apply an isotopic approach to depict the water balance during an extreme flood event. The following was also added to the introduction:

"Our study period spans a 100-year flood, and the results of this study are therefore an example indicative of an extreme hydrological event."

The following material was added to the conclusion (in replacement of L513-514), to underline the applicability of this method to other environments:

[revised manuscript text omitted]

- 35 ecosystem services often depends on the water quality of the lake (Mueller et al., 2016). Globally, the quantity and quality of groundwater and surface water resources are known to be affected by land-use (Lerner and Harris, 2009; Cunha et al., 2016; Scanlon et al., 2005) and climate changes (Delpla et al., 2009). As both surface water and groundwater contribute to lake water balances (Rosenberry et al., 2015), changes that affect the surface water/groundwater apportionment can potentially modify or threaten lake water quality (Jeppesen et al., 2014). Understanding the relative importance of the hydrological processes in
- 40 lakes can also help to depict the vulnerability and/or resilience of a lake to pollution (Rosen, 2015) as well as to invasive species (Walsh et al., 2016) and thus secure water quantity and quality over time for drinking water production purposes (Herczeg et al., 2003). In Quebec (Canada), there are an important number of municipal wells that receive contributions from surface water resources (i.e., lakes or rivers) and are thus performing unintentional (Patenaude et al., 2020) or intentional (Masse-Dufresne et al., 2019; Masse-Dufresne et al., 2020) bank filtration.
- 45 Over the past few decades, significant developments have been made in application of isotope mass balance models for assessing the spatial and temporal variability of hydrological processes in lakes; most notably, the quantification of groundwater and evaporative fluxes (Herczeg et al., 2003; Bocanegra et al., 2013; Gibson et al., 2016; Arnoux et al., 2017a). Isotopic mass balance models are typically applied to contexts where there are no surface water inputs (Sacks et al., 2014; Arnoux et al., 2017b) and/or the surface water inputs are quantified by stream gauging (Stets et al., 2010). In remote
- 50 environments, such as in northern Canada, application of isotopic methods is particularly convenient, as direct measurement of surface water inflow is difficult or nearly impossible (Turner et al., 2010; Brock et al., 2007). Recently, Haig et al. (2020) opened up new perspectives, as they reported excellent agreement between results obtained via isotopic mass balance and gauging techniques when assessing the water budget of connected lakes in Saskatchewan (Canada). They highlighted that the isotopic approach was efficient for characterizing the impacts of floods and droughts, and that a broad application can
- 55 contribute to water resources management in providing information to understand the vulnerability of ungauged systems. As future climate change impacts are expected to include increases in flood magnitude and frequency (Aissia et al., 2012), floodaffected lake water budget assessments are of utmost importance.

The main objective of this study is to demonstrate the application of isotopic mass balance to flood-affected lakes, as this approach is particularly opportune in providing estimates of the water balances and insights on the dynamics of ungauged

60 systems. We thus evaluate the importance of flood-water inputs (and bank storage) on the annual water budget of a lake located in a floodplain in an urban area, in order to depict its resilience to changes in water balance partitioning and flood-water and/or groundwater quality. To do so, we first aim to establish an isotopic framework based on the local water cycle, to verify the applicability of isotopic mass balance in the present setting, as contrasting isotopic signatures are required between various water storages and fluxes, including flood-water inputs. Secondly, we quantify the water budget according to two reference

- 65 scenarios (A and B) to grasp the impact of site-specific uncertainties on the computed results. Then, we analyze the temporal variability of the groundwater inputs and the sensitivity of the lake to flood-water driven pollution. Finally, we demonstrate the implications of flood-water storage on the water balance partition. It is hypothesized that the groundwater fluxes (inputs and outputs) through lake banks are unneglectable in lake water budgets, even for flood-affected lakes.
- The water balance is computed via a volume-dependent transient isotopic mass balance model, which is applied to predict the daily isotopic response of an artificial lake in Canada that is ephemerally connected to a 150,000 km2 watershed during spring freshet and other periods of flooding. During these recurring perennial flood events, the surficial water fluxes entering the study lake are not constrained in a gaugeable river or canal but occur over a 1-km wide surficial flood area. Our study period spans a 100-year flood, and the results of this study are therefore an example indicative of an extreme hydrological event.
- outflow, evaporation, and residence times for two young artificial groundwater lakes near Heidelberg, Germany, although these lakes had no surface water connections, and volumetric changes were considered negligible. Zimmermann (1979) showed that the lakes were actively exchanging with groundwater, which controlled the long-term rate of isotopic enrichment to isotopic steady state, but the lakes also responded to seasonal cycling in the magnitude of water balance processes. While informative, Zimmermann (1979) did not attempt to build a predictive isotope mass balance model, but rather used a best-fit

A previous study by Zimmermann (1979) similarly used a transient isotope balance to estimate groundwater inflow and

- 80 approach to obtain a solitary long-term estimate of water balance partitioning for each lake. Petermann et al. (2018) also constrained groundwater connectivity for an artificial lake near Leipzig, Germany, with no surface inlet nor outlet. By comparing groundwater inflow rates obtained via stable isotope and radon mass balances on a monthly time-step, Petermann et al. (2018) highlighted the need to consider seasonal variability when conducting lake water budget studies. Our approach builds on that of Zimmermann (1979) and Petermann et al. (2018), developing a predictive model of both atmospheric and
- 85 water balance controls on isotopic enrichment, and accounting for volumetric changes on a daily time step.

**2 Study site**

**2.1 Geological and hydrological settings**

- Located in southern Quebec, Canada, Lake A is a small artificial lake created by sand dredging activities with a maximum observed depth of 20 m (Fig. 1a). The lake constitutes the main water resource for a bank filtration system (Masse-Dufresne et al., 2019) which is designed to supply drinking water for up to 18000 people (Ageos, 2010). The lake volume (4.70 x 106 m3) was estimated based on its surface area (2.79 x 105 m2 in October 2016, measured on *Google Earth Pro*), maximum observed depth, and assuming lake bank slopes of 25 degrees (Holtz and Kovacs, 1981). An assessment of the impact of uncertainty regarding the lake geometry on the model calculation is provided in Sect. 4.3.2. The lake was excavated within alluvial sands
- 95 which were deposited in a paleo valley carved into the Champlain Sea Clays (Ageos, 2010). Lake A receives inflow from a

small stream (S1) with a mean and maximum annual discharge of 0.32 m3 s-1 and 1.19 m3 s-1, respectively. Maximum discharge typically occurs during the month of April as S1 drains snowmelt water from a small watershed (14.4 km2) (Centre d'Expertise Hydrique du Québec, 2019), whereas low to no flow is recorded for the rest of the hydrological year. Two channelized outlet streams (S2 and S3) allow water to exit Lake A and flow towards Lake Deux-Montagnes (DM). The

- 100 flow direction at S2 and S3 can be temporally reversed (Fig. 1b) when the water level of Lake DM is above the topographic threshold of 22.12 m.a.s.l. (Ageos, 2010). This process typically occurs during springtime (from April to May) and, to a lesser extent, during autumn (from October to December) and results in the inundation of the area between Lake A and Lake DM. Thus, during these flood events, the surficial water fluxes towards Lake A are not constrained in S2 and S3 but occur over a 1 km wide area. While alluvial sands were mapped in the area between Lake A and Lake DM (Fig. 1b), stratigraphic data (i.e.,
- 105 well logs) confirms that only a thin layer (few centimeters to roughly 2 meters) of alluvial sands are deposited on top the clayey sediments in the area between Lake A and Lake DM (see Fig. 1c). Hence, it is likely that little or no subsurface hydraulic connection exists between Lake A and Lake DM.

Significantly, Lake DM is the receiving waters for the Ottawa River, which drains a large watershed of approximately 150000 km2 (MDDELCC, 2015) and in turn drains to the St. Lawrence River (Fig. 1a), which is an important drinking water

110 supply for the Cities of Montreal and Quebec.

---

## Referee Report (RR1)

**Quantifying flood-water impacts on a lake water budget via volume-dependent transient stable isotope mass balance**

Janie Masse-Dufresne et al.

This study uses field sampling and a mass balance model of stable isotopes $^2$H and $^{18}$O to estimate the relative contributions of flood water and groundwater to an artificial lake over one year. The article further illustrates that while groundwater input is an important component of the lake water balance, considering temporary storage of flood water input in the subsurface reduces the magnitude of regional groundwater contribution. The authors have improved the manuscript by incorporating their responses to reviewer comments on the previous version. However, further work is required to clearly draw out the objectives of the study, and clarify the conceptual model of the lake water balance and effects on the calculated water balance.

General comments:

1. The conceptual model of the lake water balance is not clearly described or consistently applied in the manuscript. For example, the lake is the water source for a bank filtration system (S2.1), but this information is not included in the conceptualization of groundwater – surface water interactions (S2.2) or the identification of water fluxes (S3.4). Presumably, the presence of this system influences both the rate and timing of water loss from the lake, at least at this boundary.
2. Given the significant limitations to the data and modelling effort, it may be advisable to emphasise throughout the manuscript that the values produced are first-order estimates.
3. The manuscript requires thorough editing in order to improve clarity. I fully appreciate the challenges associated with working in multiple languages. Thorough editing will help draw out the science that is currently getting lost here.
4. I suggest a further check to ensure that all changes indicated in responses to reviewers have been incorporated, as some vital information has been either lost or not included, inclusion of which would strengthen the manuscript (see line by line points below).

*Abstract*

Line 13-16: The main objective of this study…water supply. Suggest generalizing. Remove "important".

Line 17: The lake typically receives… Replace "important" with "substantial". Perennial connection indicates that the connection is always present. The manuscript indicates instead that the connection is established only when lake a topographic threshold is exceeded. Suggest revisiting abstract after revision. Streamline and clearly indicate the uncertainty of water budget estimates.

*S1 Introduction*

Line 34: What do you mean by "outcome"?

Line 51-56: Recently, Haig …. ungauged systems. Condense. Impacts of floods and droughts on what?

Line 58: This objective is not specific enough. Currently, it reads like a case study. Rephrase to specify what is new about your study.

Line 60: "(and bank storage)". Including bank storage? Although it is unclear whether bank storage is the correct term to apply here. Why do you believe that the process in question is bank storage and not floodplain recharge (or a combination of the two?).

Line 61: Is it in an urban area? This is not indicated on Figure 1.

Line 67: Generally, the hypothesis comes first. Is this actually an assumption?

Line 68: "unneglectable". Replace with either important or not negligible, depending on your meaning. Given the presence of a bank filtration system adjacent to the lake, wouldn't you expect the groundwater flux out of the lake to be important?

Line 74 to end of S1: The introduction may make more sense if you move this section to before Line 58. This would allow you to clearly show what is novel about your study.

Line 71: Delete "recurring perennial".

Line 73: Delete "indicative". Check that a 100-year flood (more correctly, a flood with an Average Recurrence Interval of 100 years) is generally considered an extreme event. With changing climates, floods that were formerly considered to have ARIs of 100 years are being re-classified to higher frequencies.

*S2 Study Site*

Line 89: Was the lake created by sand dredging, or is the sand-dredging on-going as suggested in the responses to reviewers? The latter has implications for mixing in parts of the lake.

Line 90: How do you know that Lake A is the main water source for the bank filtration system, and not Lake B? Do you have an estimate of the volumes of water extracted from the bank filtration system? How do they compare to the lake volume and the flood-water input volume? Also, please include Lake B in this section. Currently it is not mentioned.

Line 93: An assessment of the impact of uncertainty… Perhaps you could delete this sentence and replace it with an error estimate after the lake volume e.g., $4.7 \times 10^6$ m$^3$ ± ? m$^3$

Line 98: Unless I have missed it, this stream is not mentioned further in the manuscript. Please add some words here to indicate this.

Line 99-104: Please reword this section to clarify the surface water flows. My interpretation is that water flows from Lake A to Lake D-M through S2 and S3 for 7 months of the year, and for the other 5 months, it flows from Lake D-M to Lake A across the floodplain and in the streams. These time periods are fairly equal, so it seems odd to describe the reversal as temporary.

Line 105-106: There is no evidence of wells outside the paleochannel in Figure 1c. Where does the evidence come from that this layer is thin? Also, the topographic threshold show on Figure 3 appears to be within the clay as depicted in Figure 1c. Please clarify.

Line 107-110: Why is this significant for this study? Does the Ottawa River flow through Lake D-M? Clarify wording. Unclear why it matters that the St Lawrence River is a drinking water source for Montreal and Quebec in the context of this study.

Line 115: Clarify that water level monitoring at VP is groundwater level. Suggest shortening caption by using correct references for data sources and including them in the data list.

Line 124: To me this section reads more as a conceptual model of the Lake A water balance. Consider revising the title.

Line 126: Suggest arranging so that the condition that comes first in the figure also comes first in the sentence – switch figure order or sentence order. I also suggest coming up with a different descriptor than "normal" – as previously mentioned, the time difference between he two conditions appears to be small.

Line 129: Fig 2b?

Line 130: replace "Contrastingly" with "In contrast".

Line 131: Replace "neglectable" with "negligible".

Line 128-133: It would be helpful to rewrite this section to clarify your conceptual model of the lake water balance under the two conditions. My current understanding is as follows:

1. Flood-water input. The level of Lake DM rises quickly due to inputs from its larger catchment. Inputs to Lake A include surface water inputs (Is) by overland flow and streamflow (S2 and S3), and precipitation (P). The resulting water level in Lake A is assumed to be higher than the surrounding groundwater; this hydraulic gradient precludes groundwater input (Ig), but increases groundwater output (Qg) above that which occurs due to the bank filtration system. Water is also lost through evaporation (E). *Please clarify if there are streamflow losses during this period as indicated on Line 134, and if so, where.*

2. Otherwise: Without flood-water inputs, the water level of Lake A falls due to outputs to E and surface water outflows (Qs) through S2 and S3, and Qg due to bank filtration system. The water level in Lake A falls more quickly than the surrounding groundwater and so the hydraulic gradient between Lake A and groundwater switches, and groundwater flows into Lake A (Ig).

With the above clarifications, this seems like a fairly reasonable conceptual model. However, it does not quite place the lake in its full hydrological context. It may be reasonable to assume that lake water flows as groundwater NE from Lake A to the bank filtration system, but what is occurring at the other lake boundaries? Are there other areas where lake discharge occurs to the groundwater when flooding is not occurring? Also, given the situation described, it seems reasonable that overland flow infiltrates into the ground on the Lake DM side (SE?) of Lake A as well as being pushed out from Lake A as bank storage. With repeated flooding events, does this not have the ability to create groundwater with very different chemistry and isotopic signature to regional groundwater/groundwater on other sides of the lake that are not subjected to flooding? Not that the bank storage and floodplain (albeit small) "groundwaters" will have essentially the same isotopic signature, and so are indistinguishable using the tools in this study.

*S3 Methods*

Line 141: Are the level loggers pressure transducers? State the start and end time of the measurement periods rather than just the start.

Line 149: What are the further computations you are referring to? Atmospheric pressure corrections?

Line 154: "close to the surface near the lake edge". State the approximate depth and distance. Also include the timeframe for sampling.

Line 163: Does this mean that the direction of regional groundwater flow is from NE to SW? Do you assume that this "regional groundwater" also contributes to Lake A, or is the bank filtration system a complete barrier?

Line 179-182: Is it necessary to include $^{17}O$? Results for this isotope are not reported in this manuscript.

Line 186: for lakes?

Line 187: Did you perform computations with both types of models? If not, suggest rewording and adding a reference that shows that both models yield similar results. Do they provide an understanding of groundwater-surface water interactions or estimates of the sources of lake inputs and output flow paths?

Line 193: Clarify the term water yield in your study context.

Line 197. Change "advocated" to "selected". This supports the general point that any quantities are only first-order estimates.

Line 205-206: "during the ice-free period". How do you justify applying it over the whole year then? The manuscript mentions that the lake freezes over. I have not further reviewed the isotopic model development.

Line 236: Which observed values?

Line 237: More correctly, the outflow from the lake will be proportional to the difference between the lake water level and the adjacent groundwater level. It would be useful to plot the difference between Lake A water level and the groundwater level to test whether this linear assumption is justified. It is rather unfortunate that the Lake A level data is not available for a longer period, as after high flows there is clearly a difference in the lake water levels, even if they follow a similar pattern. Is there any groundwater level data available on other sides of Lake A? This would also help determine if the Ig and Qg fluxes varied around the lake. Depending on the pumping volume of the bank filtration system, it is plausible that using the groundwater level at VP will overestimate the hydraulic gradient and hence the groundwater flux out of the lake. Again, what role is the bank filtration system playing here? How is the system operated? Continuously?

Line 241: Qmin on Figure 3 corresponds to the lowest water level at VP, not the lowest water level measured in Lake A. The lowest level for Lake DM was in November. Clarify.

Line 245: It may be helpful here somewhere to simply state the water balance equations for the two conditions.

Line 250: Unless the source water for S1 also comes from a large catchment at similar latitude, this is a bold statement. I suggest deleting, and leaving the reason for excluding S1 to the fact that the flows are tiny by comparison. If you state this on Line 98, there is no reason to mention this stream here.

Line 256: Suggest revising this statement to say that major flooding occurs as a result of springtime snowmelt and minor flooding due to fall precipitation.

Line 259: Figure 3?

Line 263-265: Are the water level variations synchronous (happening at exactly the same time) or following the same pattern, but with a small lag that could be expected due to travel time? Amend to be clear that data is not available for Lake A up to late July. Does the water level measured at VP ever exceed ground level?

Line 267: What is this natural threshold? It needs to be clearly explained earlier in the manuscript.

Line 273: What I take from the manual water level measurements is that 1) the water level of Lake A is always higher than the groundwater level measured at VP and 2) Lake A and Lake DM do not always have similar water levels, regardless of whether the water level is above or below the topographic threshold. It would be useful to explain these aspects in the context of the Lake A water balance developed in S2.

Line 275: Isn't the actual volume increasing?

Line 281: Was a manual measurement of Lake A water level taken at the start of the study period? It isn't shown on Fig 3.

Line 282-286: Unclear the importance of the lake being dredged. Unless there is evidence of a low hydraulic conductivity layer around the edges of the lake, wouldn't you expect hydraulic connection when the lake is situated in alluvial sands? Perhaps rephrase to indicate that gross water fluxes are likely to exceed net water fluxes due to the surrounding geology and measured hydraulic gradients. A critical point that is missing here (and from the lake water balance conceptual model) is where the water goes when it drains from Lake A, whether surface or subsurface. The levels indicated on Fig 3 suggest that, at least in the early stages, the water is not draining to Lake DM.

Line 289: It is misleading to say that the shaded area represents flood water inputs. Won't surface water inputs from Lake DM to Lake A be received at all times that the water level of Lake DM is above the topographic threshold? Clarify throughout document and in the figure.

Line 298: Why 2016? The rest of the sampling was conducted in 2017.

Line 309: What is the justification for this? Do the three sample indicate temporal change in the flood-water inputs? How would this affect the calculated water budget?

Line 310: Similar to what? Unclear why this is relevant.

Line 313-Line 320: Suggest considering Jasechko et al. (2017) (doi: 10.1002/hyp.11175) and Welch et al. (2018) (doi: 10.1002/hyp.11396), which demonstrate widespread cold season bias to groundwater recharge in similar climates. Do you have any isotopic data from either of the observation wells?

Line 330: Did you perform significance testing to determine this? If yes, present the results. If not, change the wording.

Line 337: What are these scenarios (A and B) and what do they represent? It is difficult to interpret the following results without understanding this.

Line 369: Why do you consider these have stopped? Fig 3 indicates that the level in Lake DM remains higher than Lake A until Lake A measurements cease.

Line 375: How do you reconcile this with the fact that inspection of Fig 6 indicates that the isotopic match is closer for Scenario B?

Line 391: Consider moving the definition of mean flushing time to the methods. It is referred to extensively in the following pages and is hard to find here buried in a paragraph. The flushing time for Scenario B is approximately 30% lower than for Scenario A – is this not a fairly large difference?

Line 401: Suggest including explanation provided to reviewer that this sensitivity analysis was conducted OAT.

Line 404: This range of δIs does not cover the range of observed flood water inputs.

Line 408: Given the heat capacity of water, it seems unusual to use this as the only boundary for varying water temperature.

Line 416: As previously mentioned, this is not the only time when a hydraulic connection appears to form. How does amending this assumption affect the results?

Line 420: It would be helpful to state specifics for the change in LMWL and δIs/g and water balance.

Line 421: What are the impacts on the water budget of holding δIs/g constant? This at least needs to be discussed as a study limitation. Similarly, the assumption of a well-mixed lake needs to be discussed, given the obvious isotopic stratification.

Line 435: I suggest deleting this reference to the discussion and including analysis of the likely temporal change in groundwater inputs here in the results (Lines 448-470). Perhaps you could consider modifying your model to include two different groundwater inputs – one that reflects the regional groundwater, and one that reflects the mixture between flood-water inputs through the floodplain or bank storage and this regional groundwater. Consider also doing a simple calculation to estimate the potential volume of flood water that could be stored in the subsurface using values for alluvial sands available in the literature (if there are no local measurements) and the maximum depth that the groundwater level lowers in the dry season.

*S5 Discussion*

Line 509: "expected increases in water levels…" Which water levels?

Line 521: It is unclear how this study tracks human impacts on the water cycle.

Line 535: Another big missing piece in the data currently presented is the lack of water level measurements around the lake and through time that support the hypothesis that groundwater discharges into the lake. See also comment on Line 421.

*S6 Conclusions*

This section can be made more succinct. Focus on the major findings. No need to repeat volumes.

---

## Referee Report (RR2)

Hello,

This manuscript has vastly improved with the reorganisation and clarification provided. I have two general concerns remaining:

1. The link to bank storage is weak in the modelling and results, but prominent in the findings. The available hydraulic data do not provide supporting evidence (the groundwater level is not recorded above the water level of Lake A). I understand there were technical issues with the hydraulic measurements, so perhaps more use could be made of the isotopic model, or a clearer link provided to the hypothetical scenarios presented for the G-index. For example, can you get a good fit to the isotopic data with a groundwater input that has an isotopic composition intermediate to flood water and groundwater in the May – August time period?
2. The isotopic data presented in Appendix E (and Figure 6) clearly indicates that the lake is not well-mixed in August (the only depth profiles outside of winter). Arguments supporting the well-mixed model choice are provided; however, I do not think the first-order nature of the estimate of water fluxes is clearly emphasized throughout the manuscript (for example, it is not mentioned in the abstract).

Other minor comments are provided below.

Line 21: Lake A water budget?

Line 22: This is contradicted un the discussion.

Line 48: Delete "nearly impossible". The reference doesn't say it's impossible. Just time-consuming, expensive, and difficult.

Line 54: This paragraph would benefit from a topic sentence before diving into the specifics of previous studies.

Line 66: I remain unconvinced that bank storage is the correct term – perhaps a more general delayed (in subsurface) and direct flood water inputs?

Line 72: Flood water storage – surface (in the lake) or subsurface?

Line 81: "a" not "an"

Line 100: Rephrase: The direction of the surface water flux in S2 reverses when the water level in Lake DM exceeds… Does flow reversal also occur in S3?

Line 124: "Tin Lake A" – rephrase or define.

Line 125: Define the observed period here at first mention.

Line 127: Suggest rephrasing - It remains an assumption that Lake DM controls the water level variation at obs well VP. Perhaps "the data indicates". To the reader it is unclear why the Lake A water level is not used – state here that the logger broke.

Line 131: Suggest rephrasing to say that Lake A water level is not controlled by Lake DM in this period.

Line 137: Figure 2 indicates the water level in Lake A is below Lake DM in mid Dec (even considering error bars shown) – why wouldn't the water flow into it? Maybe it only happens for a short time (which could explain the lower correlation)

Line 151: Could there not also be Qg out the southern end of Lake A? Ie in the direction of regional groundwater flow.

Line 157: Rephrase to put Lake DM immediately after Qs otherwise this sentence reads as if Qg is possible going to Lake DM, which contradicts Line 150.

Line 161: If the water causing the water level increases is not coming from Lake DM, then where is it coming from? Direct precipitation? S1?

Line 274: The manuscript first argues that there is not a strong correlation, and then use Lake DM as representative? I get that the absolute level doesn't matter, but why not just use the obs well the whole time? Does this change the results?

Line 283: Are there 2 Penman-48 methods? Suggest rephrasing as it is unclear which is used in the model, the one that underestimates or the one that doesn't. If it's the one that doesn't underestimate in late summer-fall, the results are unlikely to be affected. Otherwise: Line 284: how did you resolve this? Specific heat capacity of water. Does this markedly affect the isotopic balance and interpretation?

Line 291: "The outflows of the lake are thus…" I still find these comments misleading. I suggest rewording to explain that the change in outflows from the lake is roughly proportional to the change in water level.

Line 302: Suggest inserting "it is assumed that" immediately before "the rising water level". Groundwater may still be entering lower in the lake.

Line 344: Does this use of the LMWL-LEL account for flood waters? This assumption will likely only hold where groundwater is sourced from the local precipitation, and is not subjected to different rates of evaporation during infiltration. It is curious to me that no isotopic data collected from the groundwater wells at the site is presented to support this choice.

Line 348: Where is Saint-Telesphore and why is it considered a useful comparison?

Line 377: depth-average"d" – the "d" is missing

Line 381-384: Unclear where the dates and isotopic values come from until reading the caption to Table 1. It would help the reader if the scenarios were more clearly presented and their purpose more clearly articulated – ie is it to compare lengths of flood-water control?

Section 2.2 Sensitivity analysis. Should this be Section 4.2.2? Given what the paper is aiming to do, this section would benefit from linking the effects of these changes to the different water fluxes.

Section 4.4 I'm struggling to reconcile this with the isotopic modelling – if the water discharging back to the lake is flood water, then wouldn't it have the isotopic signature of flood water? My understanding is that deltaG is held constant in the model. To test this, why not alter deltaG over this time period?

Line 577: This contradicts line 522 which categorises the lake as relatively resilient – clarify.

---

## Author Response (AR2)

**Report #1**

Submitted on 02 Jan 2021
Referee #3: Chani Welch, cwelch@okanagan.bc.ca

**Suggestions for revision or reasons for rejection (will be published if the paper is accepted for final publication):**

This study uses field sampling and a mass balance model of stable isotopes 2H and 18O to estimate the relative contributions of flood water and groundwater to an artificial lake over one year. The article further illustrates that while groundwater input is an important component of the lake water balance, considering temporary storage of flood water input in the subsurface reduces the magnitude of regional groundwater contribution. The authors have improved the manuscript by incorporating their responses to reviewer comments on the previous version. However, further work is required to clearly draw out the objectives of the study, and clarify the conceptual model of the lake water balance and effects on the calculated water balance.

***General comments:***
**RC3-1.** The conceptual model of the lake water balance is not clearly described or consistently applied in the manuscript. For example, the lake is the water source for a bank filtration system (S2.1), but this information is not included in the conceptualization of groundwater – surface water interactions (S2.2) or the identification of water fluxes (S3.4). Presumably, the presence of this system influences both the rate and timing of water loss from the lake, at least at this boundary.
**Answer: Done.** We improved the description of the conceptual model. To better introduce the conceptual model, we now present the hydrodynamics of the flood event in 2.2 (the conceptual model is thus in 2.3). Also, the impact of the bank filtration system on the location of the groundwater outflows are now specified.

**RC3-2.** Given the significant limitations to the data and modelling effort, it may be advisable to emphasise throughout the manuscript that the values produced are first-order estimates.
**Answer: Done.** We emphasized the fact our model yield first-order estimates in Sect. 3.3 and in the conclusion. See response to comment RC3-39 and RC3-74.

**RC3-3.** The manuscript requires thorough editing in order to improve clarity. I fully appreciate the challenges associated with working in multiple languages. Thorough editing will help draw out the science that is currently getting lost here.
**Answer: Done.** A thorough editing of the manuscript was done by an English-speaking colleague. Minor corrections were made and marked in red in the manuscript.

**RC3-4.** I suggest a further check to ensure that all changes indicated in responses to reviewers have been incorporated, as some vital information has been either lost or not included, inclusion of which would strengthen the manuscript (see line by line points below).
**Answer: Done.** We checked that all responses to reviewers have been incorporated. The following specific comments helped at further improving the manuscript.

***Abstract***
**RC3-5.** Line 13-16: The main objective of this study…water supply. Suggest generalizing. Remove "important".
**Answer: Done.** We generalized the objective of the study. L13-16 can now read as:

*"In this study, we demonstrate that isotopic mass balance modelling can be used to provide evidence of the relative importance of bank storage and direct flood-water inputs at ungauged lake systems."*

**RC3-6a.** Line 17: The lake typically receives… Replace "important" with "substantial".
**Answer: Done.** It was corrected to "substantial".

**RC3-6b.** Perennial connection indicates that the connection is always present. The manuscript indicates instead that the connection is established only when lake a topographic threshold is exceeded.
**Answer: Done.** It was corrected to "ephemeral".

**RC3-6c.** Suggest revisiting abstract after revision. Streamline and clearly indicate the uncertainty of water budget estimates.
**Answer: Done.** The abstract was revised at the end of the revision process. The objective was reformulated (see response to comment RC3-5). We now consider that it describes the article correctly.

**S1 Introduction**
**RC3-7.** Line 34: What do you mean by "outcome"?
**Answer: Done.** We were referring to the actual value of the ecosystem services. To clarify, we modified L33-35, which can now read as:
*"In fact, lacustrine ecosystems can provide a number of ecosystem services, such as biodiversity, water supply, recreation and tourism, fisheries and sequestration of nutrients (Schallenberg et al., 2013). The actual benefits that can be provided by lakes depend on the water quality, and poor resilience to water quality changes can lead to benefit losses (Mueller et al., 2016)."*

**RC3-8.** Line 51-56: Recently, Haig …. ungauged systems. Condense. Impacts of floods and droughts on what?
**Answer: Done.** We meant that the isotopic mass balance models can be applied to ungauged lake systems and can efficiently characterize the impacts of floods and droughts on water apportionment. To condense L51-56, we opted to improve the link with the previous sentence. Also, we took the opportunity to better contextualize our work relatively to the previous studies in ungauged systems. We thus rephrased L49-57 as:
*"In remote environments, such as in northern Canada, application of isotopic methods is particularly convenient, as direct measurements of surface water and groundwater fluxes is difficult or nearly impossible (Welch et al., 2018). Isotopic mass balance models can notably be applied to ungauged lake systems to efficiently characterize the impacts of floods on water apportionment (Haig et al., 2020). While isotopic frameworks were successfully used to assess the relative importance of flood-water inputs to lakes (Turner et al., 2010; Brock et al., 2007), no attempt was made at evaluating the timing of the flood-water inputs and to differentiating between the role of direct flood-water inputs and indirect delayed inputs from flood-water bank storage on a lake's annual water budget."*

**RC3-9.** Line 58: This objective is not specific enough. Currently, it reads like a case study. Rephrase to specify what is new about your study.
**Answer: Done.** We rephrased the objective of the study to emphasize the originality of our work. It can now read as:

> *"The main objective of this study is to provide evidence of the relative importance of bank storage and direct flood-water inputs at ungauged lake systems using an isotopic mass balance model."*

**RC3-10a.** Line 60: "(and bank storage)". Including bank storage?
**Answer: Done.** By rephrasing the objective of the study, the focus is now on both the direct flood-water inputs and the indirect flood-water inputs as bank storage (see response to RC3-9).

**RC3-10b.** Although it is unclear whether bank storage is the correct term to apply here. Why do you believe that the process in question is bank storage and not floodplain recharge (or a combination of the two?).
**Answer: Clarification.** In this study, we aim at quantifying the indirect flood-water inputs as bank storage. To do so, we consider a theoretical scenario in which all the outflowing water from the lake during the flood event eventually discharge back to the lake due to bank storage process. We do not aim at quantifying the impact of floodplain recharge on the annual water budget of the lake. However, the floodplain recharge process is implicitly taken into account in our model by the way we define the isotopic signature of groundwater ($\delta_G$), i.e., the intersection between the LMWL and the lake's LEL. In that sense, we did not make further correction to the manuscript at L60.

**RC3-11.** Line 61: Is it in an urban area? This is not indicated on Figure 1.
**Answer: Clarifications and done.** The study site is located in the metropolitan region of Montréal, as defined by the 2016 census of population (Statistics Canada). This information was added to Figure 1. It is noteworthy that L61 was reformulated in an effort to clarify the objective of the study and to condense the introduction. The fact that the study site is located in an urban area is now stated in section 2.1.

**RC3-12.** Line 67: Generally, the hypothesis comes first. Is this actually an assumption?
**Answer: Done.** See response to comment RC3-13.

**RC3-13.** Line 68: "unneglectable". Replace with either important or not negligible, depending on your meaning. Given the presence of a bank filtration system adjacent to the lake, wouldn't you expect the groundwater flux out of the lake to be important?
**Answer: Done.** As pointed out by the reviewer, this statement seems trivial. We opted to remove L67-68.

**RC3-14.** Line 74 to end of S1: The introduction may make more sense if you move this section to before Line 58. This would allow you to clearly show what is novel about your study.
**Answer: Done.** The paragraph was moved to L58.

**RC3-15.** Line 71: Delete "recurring perennial".
**Answer: Done.** It was deleted.

**RC3-16.** Line 73: Delete "indicative". Check that a 100-year flood (more correctly, a flood with an Average Recurrence Interval of 100 years) is generally considered an extreme event**.** With changing climates, floods that were formerly considered to have ARIs of 100 years are being re-classified to higher frequencies.
**Answer: Done.** The term "indicative" was deleted. Concerning the designation of the flood event, we agree with the reviewer that "100-year flood" should be replaced by a more adequate term, such as "a flood with an average recurrence interval of 100 years". Similarly, the USGS recommends using the term "1-percent annual exceedance probability (AEP) flood" (https://www.usgs.gov/special-topic/water-science-school/science/floods-and-recurrenceintervals?qt-science_center_objects=0#qt-science_center_objects). The Government of Quebec classifies the 2017 major flood event as "a flood exceeding the recurrence interval of 100 years". Considering all the above, we corrected L73 to: "Our study period spans a flood event, exceeding the recurrence interval of 100 years, and is therefore an example of the response of the system to a major hydrological events."

**S2 Study Site**
**RC3-17.** Line 89: Was the lake created by sand dredging, or is the sand-dredging on-going as suggested in the responses to reviewers? The latter has implications for mixing in parts of the lake.

**Answer: Clarifications and done.** We corrected the manuscript to specify that the sand-dredging is still on-going. Also, we agree that the dredging process might contribute to the mixing of the water column in some parts of the lake. However, we expect the dredging to have a small impact on an annual time scale, and we made no modification to the manuscript regarding this concern. In fact, the dredging only takes place during the weekdays and between June to October - no dredging is done during the night, the weekends, and the ice-cover period. Additionally, on-field observations let us believe that the mobilized water volumes by the dredging process are small compared to the lake volume. Although no data is available to quantify the impact of dredging on mixing in part of the lake, horizontal homogeneity across the lake was previously demonstrated by Pazouki et al. (2016). In their study, the authors reported good similarity between depth-resolved physico-chemical profiles ($n = 4$) performed in different zones of the lake during summertime. This information is mentioned in the manuscript (see section 3.2).

**RC3-18a.** Line 90: How do you know that Lake A is the main water source for the bank filtration system, and not Lake B? Do you have an estimate of the volumes of water extracted from the bank filtration system? How do they compare to the lake volume and the flood-water input volume?

**Answer: Clarifications.** The evaluation of the water origin at the bank filtration system was the purpose of a previous work (Masse-Dufresne et al., 2019). The pumped volume is about $2 \times 10^6$ $m^3$ over the study period (i.e., from February 2017 to January 2018), which corresponds to 43% of the lake volume and 42% of the total flood-water input (in scenario A).

**RC3-18b.** Also, please include Lake B in this section. Currently it is not mentioned.

**Answer: Done.** As description of Lake B was added to the section 2.1. We also made an effort at reorganizing the information presented in this section in order to ease the reading and make it more concise. The Ottawa river and Lake DM are now presented first. Then, we focus on Lake A and Lake B and the bank filtration system. We moved all information that is specific to the 2017 flooding event and to our conceptual model in the section 2.2.

**RC3-19.** Line 93: An assessment of the impact of uncertainty… Perhaps you could delete this sentence and replace it with an error estimate after the lake volume e.g., 4.7 x 106 m3 ± ? m3

**Answer: Clarifications.** To estimate the initial volume of the lake ($4.7 \times 10^6$ $m^3$), we made an assumption regarding the slope of the banks. In this context, we designed the sensitivity analysis to consider a realistic range for the bank slopes, rather than an error estimate on the initial lake volume (which is difficult to evaluate). By varying the bank slopes from 20 degrees to 30 degrees, the initial lake volume ranges from $4.84 \times 10^6$ $m^3$ (+3%) and $4.32 \times 10^6$ $m^3$ (-8%). It would be confusing to present these values as error estimate of the lake volume at L93. Hence, we opted to modify L93 and to only present the lake surface area and depth in section 2.1, as this is the available data. The estimation of the initial Lake A volume in now presented in section 3.4, i.e., where we define the model parametrization. This can read as:

"The initial lake volume ($4.7 \times 10^6$ m$^3$) was estimated from the observed lake surface area ($2.79 \times 10^5$ m$^2$) and the maximal depth (20 m) and assuming bank slopes of 25 degrees. Assuming bank slopes of 20 degrees or 30 degrees, a typical range for saturated sands (Holtz and Kovacs, 1981), would result in an estimated initial lake volume of $4.84 \times 10^6$ m$^3$ (+3%) and $4.32 \times 10^6$ m$^3$ (-8%)."

**RC3-20.** Line 98: Unless I have missed it, this stream is not mentioned further in the manuscript. Please add some words here to indicate this.
**Answer: Clarifications.** It was mentioned in section 3.4 (L250-253) that the potential contribution from S1 is neglected in this study. To clarify, we emphasized this assumption at section 2.1.

**RC3-21.** Line 99-104: Please reword this section to clarify the surface water flows. My interpretation is that water flows from Lake A to Lake D-M through S2 and S3 for 7 months of the year, and for the other 5 months, it flows from Lake D-M to Lake A across the floodplain and in the streams. These time periods are fairly equal, so it seems odd to describe the reversal as temporary.
**Answer: Done.** We agree with the reviewer that the term "temporary" may be misleading and was thus removed. Additionally, we would like to mention that we simplified the information initially presented at L99-104. All information related to the specific hydrological context of the 2017 major flooding is now presented in Section 2.3, i.e., where the conceptual model is detailed.

**RC3-22.** Line 105-106: There is no evidence of wells outside the paleochannel in Figure 1c. Where does the evidence come from that this layer is thin? Also, the topographic threshold show on Figure 3 appears to be within the clay as depicted in Figure 1c. Please clarify.
**Answer: Clarifications and done.** The locations of additional well logs were added to the figure. Note that the Figure 1c was moved in appendix, as modifications to figure were made to clarify the context of the study and better contextualize the study site relatively to the Ottawa River watershed. Also, the topographic threshold was determined from a land survey and corresponds to the maximum elevation along S2 streambed. This information was added to the manuscript in Section 2.1.

**RC3-23.** Line 107-110: Why is this significant for this study? Does the Ottawa River flow through Lake D-M? Clarify wording. Unclear why it matters that the St Lawrence River is a drinking water source for Montreal and Quebec in the context of this study.
**Answer: Clarification.** The Lake DM is an enlargement of the Ottawa River at the confluence with St. Lawrence River. We clarified this information in section 2.1. It is important to contextualize Lake DM relatively to the Ottawa River watershed in this study, because the 2017 major flood event was caused by the combination of intense precipitations and snowpack melting over the Ottawa River watershed (Teufel et al., 2019). The latter was clarified in the manuscript.

Teufel, B., Sushama, L., Huziy, O., Diro, G. T., Jeong, D. I., Winger, K., . . . Nguyen, V. T. V. (2019). Investigation of the mechanisms leading to the 2017 Montreal flood. Climate Dynamics, 52(7), 4193-4206. doi:10.1007/s00382-018-4375-0

**RC3-24.** Line 115: Clarify that water level monitoring at VP is groundwater level. Suggest shortening caption by using correct references for data sources and including them in the data list.
**Answer: Done.** The that data sources were added to Figure 1 caption upon request of the Editorial support (at the submission stage). However, we agree with the reviewer that a shorter caption would benefit the reading. To do so, we now provide a detailed listing of the freely accessed

geospatial data and the related sources in Appendix. Also, we opted to simply delete the information concerning VP in Figure 1 caption, because it is stated in the manuscript.

**RC3-25.** Line 124: To me this section reads more as a conceptual model of the Lake A water balance. Consider revising the title.
**Answer: Done.** The reviewer is correct. The title was corrected to: "2.2 Conceptual model of Lake A water balance"

**RC3-26.** Line 126: Suggest arranging so that the condition that comes first in the figure also comes first in the sentence – switch figure order or sentence order. I also suggest coming up with a different descriptor than "normal" – as previously mentioned, the time difference between the two conditions appears to be small.
**Answer: Done.** The sentence order was rearranged so that the "normal period" comes first in both the text and the figure. Also, we opted to modify the descriptors to "groundwater control period" and "flood-water control period". This modification was done in Figure 2 and throughout the manuscript.

**RC3-27.** Line 129: Fig 2b?
**Answer: Done.** It was corrected to: "(Fig. 2b)". Similarly, L131 was corrected to "(Fig. 2a)".

**RC3-28.** Line 130: replace "Contrastingly" with "In contrast".
**Answer: Done.** It was corrected to "In contrast".

**RC3-29.** Line 131: Replace "neglectable" with "negligible".
**Answer: Done.** It was corrected to "negligible".

**RC3-30.** Line 128-133: It would be helpful to rewrite this section to clarify your conceptual model of the lake water balance under the two conditions. My current understanding is as follows:

1. Flood-water input. The level of Lake DM rises quickly due to inputs from its larger catchment. Inputs to Lake A include surface water inputs (Is) by overland flow and streamflow (S2 and S3), and precipitation (P). The resulting water level in Lake A is assumed to be higher than the surrounding groundwater; this hydraulic gradient precludes groundwater input (Ig), but increases groundwater output (Qg) above that which occurs due to the bank filtration system. Water is also lost through evaporation (E). *Please clarify if there are streamflow losses during this period as indicated on Line 134, and if so, where.*

2. Otherwise: Without flood-water inputs, the water level of Lake A falls due to outputs to E and surface water outflows (Qs) through S2 and S3, and Qg due to bank filtration system. The water level in Lake A falls more quickly than the surrounding groundwater and so the hydraulic gradient between Lake A and groundwater switches, and groundwater flows into Lake A (Ig).

With the above clarifications, this seems like a fairly reasonable conceptual model. However, it does not quite place the lake in its full hydrological context. It may be reasonable to assume that lake water flows as groundwater NE from Lake A to the bank filtration system, but what is occurring at the other lake boundaries? Are there other areas where lake discharge occurs to the groundwater when flooding is not occurring? Also, given the situation described, it seems reasonable that overland flow infiltrates into the ground on the Lake DM side (SE?) of Lake A as well as being pushed out from Lake A as bank storage. With repeated flooding events, does this not have the ability to create groundwater with very different chemistry and isotopic signature to regional groundwater/groundwater on other sides of the lake that are not subjected to flooding?

Not that the bank storage and floodplain (albeit small) "groundwaters" will have essentially the same isotopic signature, and so are indistinguishable using the tools in this study.

**Answer: Clarification and done.** The reviewer's understanding of the conceptual model is correct. As suggested, we corrected L134, because there are no streamflow losses during the flood-water control period. Additionally, we reformulated this section to better "place the lake in its full hydrological context". Concerning the isotopic signature of the groundwater in the vicinity of the lake, we present an isotopic framework in section 4.1, where we show the cold-season bias to groundwater recharge. No information regarding the isotopic composition of groundwater was added in the conceptual model.

**S3 Methods**

**RC3-31a.** Line 141: Are the level loggers pressure transducers?

**Answer: Done.** The pressure sensors are piezo resistive ceramic ($Al_2O_3$) with thermal compensation. Note that a level logger was also installed on-site to measure the atmospheric pressure and perform barometric compensation on the water level measurements. This information was added to the manuscript.

**RC3-31b.** State the start and end time of the measurement periods rather than just the start.

**Answer: Done.** The end dates of the measurement periods were also added to the text.

**RC3-32.** Line 149: What are the further computations you are referring to? Atmospheric pressure corrections?

**Answer: Done.** We were referring to the isotopic mass balance model, but this information was removed from the manuscript. In fact, we rephrased the information concerning the meteorological data and added the distances between the stations and the study site, as suggested by the other reviewer (see response to comment RC4-4).

**RC3-33.** Line 154: "close to the surface near the lake edge". State the approximate depth and distance. Also include the timeframe for sampling.

**Answer: Done.** The samples were collected at approximately 0.3 m below the lake surface and 1 m from the lake shoreline. The timeframe for sampling was between February 9, 2017 and January 25, 2018. This information was added to the manuscript.

**RC3-34.** Line 163: Does this mean that the direction of regional groundwater flow is from NE to SW? Do you assume that this "regional groundwater" also contributes to Lake A, or is the bank filtration system a complete barrier?

**Answer: Clarification.** There is a groundwater flow in the NE-SW direction contributing to Lake B, but evidences suggest that it is not contributing to Lake A. In fact, while the bank filtration system is not a complete barrier, the water level of Lake A is higher than the one of Lake B. Hence, no water can flow from Lake B to Lake A.

**RC3-35.** Line 179-182: Is it necessary to include 17O? Results for this isotope are not reported in this manuscript.

**Answer: Done.** The reviewer is correct – the results are not reported in the manuscript. In fact, $\delta^{17}O$ does not provide additional information (in comparison to $\delta^{18}O$ and $\delta^2H$). Hence, we opted to delete the $^{17}O$ considerations from L179-182.

**RC3-36.** Line 186: for lakes?

**Answer: Done.** It is now specified that these considerations apply to lakes.

**RC3-37a.** Line 187: Did you perform computations with both types of models? If not, suggest rewording and adding a reference that shows that both models yield similar results.
**Answer: Clarification and done.** In our study, we only performed a well-mixed model. Arnoux et al. (2017b) reported similar results using both modeling methods. This was clarified in the manuscript by rewording L186-190.

**RC3-37b.** Do they provide an understanding of groundwater-surface water interactions or estimates of the sources of lake inputs and output flow paths?
**Answer: Clarification and done.** The reviewer is right - both types of models more correctly provide an estimation of the groundwater fluxes. We corrected as: "*Arnoux et al. (2017c) performed a comparison of both methods and reported that well-mixed and depth resolved multi-layered models yielded similar and showed that groundwater inputs and outputs play an important role on lake water budgets.*"

**RC3-38.** Line 193: Clarify the term water yield in your study context.
**Answer: Clarification.** The term "water yield" is not relevant in the context of our study – we mentioned this term as we reported the results of Gibson et al. (2017). We have reworded L191-197 in order to be more concise and strictly present the information that supports our modelling choice.

**RC3-39.** Line 197. Change "advocated" to "selected". This supports the general point that any quantities are only first-order estimates.
**Answer: Done.** L197-198 was rephrased (see response to comment RC3-37). The term "advocated" was corrected to "opted to develop". Additionally, we emphasized the fact our model yield first-order estimates by adding the following:
> "*Note that, despite the biases underlying well-mixed models, this approach remains adequate to characterize the relative importance of hydrological processes and is particularly useful to give first-order estimate of water fluxes in ungauged basins.*"

**RC3-40.** Line 205-206: "during the ice-free period". How do you justify applying it over the whole year then? The manuscript mentions that the lake freezes over. I have not further reviewed the isotopic model development.
**Answer: Clarification.** The justification to apply the model over the whole year is provided at L207-211, as recommended by another reviewer (see response to comment SC1-12b). This justification reads as:
> "*In this study, the potential impacts of the ice-cover formation and melting are neglected, as the ice volume is likely to represent only a small fraction (<2%) of the entire water body. Moreover, considering the ice-water isotopic separation factor, i.e., 3.1 ‰ for $\delta^{18}O$ and 19.3 ‰ for $\delta^{2}H$ (O'Neil, 1968) and assuming well-mixed conditions, the lake water isotopic variation would be comprised within the analytical uncertainty. Also, flood-water inputs from Lake DM were expected to be much more important and occurring simultaneously with ice-melt during the freshet period.*"

**RC3-41.** Line 236: Which observed values?
**Answer: Clarification.** We meant the isotopic signature of the lake ($\delta_L$). However, we opted to remove this sentence as it is redundant with the information stated at L241-242 which as added considering the suggestion of another reviewer (see RC1-5).

**RC3-42.** Line 237: More correctly, the outflow from the lake will be proportional to the difference between the lake water level and the adjacent groundwater level. It would be useful to plot the

difference between Lake A water level and the groundwater level to test whether this linear assumption is justified. It is rather unfortunate that the Lake A level data is not available for a longer period, as after high flows there is clearly a difference in the lake water levels, even if they follow a similar pattern. Is there any groundwater level data available on other sides of Lake A? This would also help determine if the Ig and Qg fluxes varied around the lake. Depending on the pumping volume of the bank filtration system, it is plausible that using the groundwater level at VP will overestimate the hydraulic gradient and hence the groundwater flux out of the lake. Again, what role is the bank filtration system playing here? How is the system operated? Continuously?

**Answer: Clarification.** The reviewer is correct – based on Darcy's Law, the outflow from the lake would be proportional to the difference between the lake water level and the adjacent groundwater level (i.e., the hydraulic gradient). However, there is no groundwater level data available in the vicinity of Lake A (except for the observation wells that are influenced by the pumping at the bank filtration system). It is thus impossible to estimate a realistic hydraulic gradient between Lake A and the surrounding aquifer. In this context, the daily water level was deemed to be the best available proxy to constrain the non-fractionating outflow fluxes (Q) from the lake. Besides, it is important to note that our study was not originally designed to perform an isotopic mass balance model, but we took the opportunity to do so when a major flood event occurred. Otherwise, we would have installed observation wells in the vicinity of the SW bank of Lake A to characterize the groundwater level variations.

Concerning the bank filtration system, it is operated continuously at pumping rates ranging from 4000 $m^3/d$ (in wintertime) to 7500 $m^3/d$ (in summertime). The estimated pumped volume is $2 \times 10^6\, m^3$ over the study period (i.e., from February 2017 to January 2018). This volume corresponds to 12% and 8% of the total estimated outflow according the scenario A and scenario B, respectively. Typically, only two to four pumping well are in operation, and there is no continuous hydraulic barrier between Lake A and Lake B. Moreover, Lake B water level is lower than Lake A. It is thus expected that a proportion of the groundwater outflows from Lake A discharge into Lake B.

**RC3-43.** Line 241: Qmin on Figure 3 corresponds to the lowest water level at VP, not the lowest water level measured in Lake A. The lowest level for Lake DM was in November. Clarify.

**Answer: Clarification.** $Q_{min}$ and $Q_{max}$ correspond to the lowest water level at Lake A. As measurements of water level at Lake A are only available for a short period, we did a reconstruction of Lake A water level from the available measurements. We described the use of Lake DM and observation well VP water levels as proxies for Lake A water level at L342-346 (Section 4). However, we conceive that it would benefit the reading to state this information earlier in the manuscript. Hence, we moved L342-346 to section 2.2 and added the following material to clarify:

> "*Lake A volume variations are estimated from daily water level changes and assuming a constant lake area. As water level measurement are only available for a short period at Lake A, water levels at Lake DM and observation well VP are used as proxies. Water levels at observation well VP were used as a proxy from August 24th, 2017 to October 30th, 2017, while water level at Lake DM was assumed representative of Lake A for the rest of the study period (i.e., from February 9th, 2017 to August 23th, 2017 and from October 31st, 2017 to January 25th, 2018). This approximation is deemed acceptable because the simulation of δL depends on the remaining fraction of lake water f (not the absolute water level), and daily variations of the water levels at Lake A, Lake DM and observation well VP were shown to be similar (see Sect. 2.2).*"

**RC3-44.** Line 245: It may be helpful here somewhere to simply state the water balance equations for the two conditions.

**Answer: Clarification.** We agree with the reviewer that it could help to state the balance equations for the two conditions. Hence, we added the following material to section 3.3:

> *"In the context of this study, the balance equations can be simplified based on the conceptual model. During the normal period, $I_S = 0$ and, thus, $I = I_G + P$ and $\delta_I = (\delta_G I_G + \delta_P I_P)/I$. In contrast, $I_G = 0$ during the flood-water control period, $I = I_S + P$ and $\delta_I = (\delta_{Is} I_S + \delta_P I_P)/I$. Note that $\delta_G$ and $\delta_{Is}$ are the isotopic signatures of groundwater and surface water inputs, respectively."*

**RC3-45.** Line 250: Unless the source water for S1 also comes from a large catchment at similar latitude, this is a bold statement. I suggest deleting and leaving the reason for excluding S1 to the fact that the flows are tiny by comparison. If you state this on Line 98, there is no reason to mention this stream here.

**Answer: Done.** As suggested, we opted to delete this statement as it is also explained earlier in the text. See response to comment R3-20.

**S4 Results**

**RC3-46.** Line 256: Suggest revising this statement to say that major flooding occurs as a result of springtime snowmelt and minor flooding due to fall precipitation.

**Answer: Clarification.** As explained in response to comment R3-23, the 2017 major flood event was caused by the combination of intense precipitations and snowpack melting over the Ottawa River watershed (Teufel et al., 2019). This was clarified in the manuscript.

**RC3-47.** Line 259: Figure 3?

**Answer: Done.** The reviewer is correct. However, we now present the "Hydrodynamics of the flood event" in section 2.2, and the figure is now numbered as Figure 2.

**RC3-48a.** Line 263-265: Are the water level variations synchronous (happening at exactly the same time) or following the same pattern, but with a small lag that could be expected due to travel time? Amend to be clear that data is not available for Lake A up to late July.

**Answer: Clarification and done.** When we take a close look at Fig. 2 (revised as Fig. 3), there is no phase shift between the peaks of Lake DM and observation well VP daily mean water levels from April to July (see figure below). Considering this, it is possible to think that the time lag < 1 day. However, it is not possible to verify if the water level variations are happening at exactly the same time, as only the daily mean water level is available for Lake DM. Besides, your comment made us realize that L263-265 may be confusing regarding the comparison of the water level variations of Lake A and observation well VP to the one of Lake DM. In this context, we opted to reword L256-274 and present it earlier in the manuscript (in 2.2) to help clarify the conceptual model. We now use the term "follow a similar pattern" to compare the evolution of Lake A and Lake DM water levels. More generally, we now use correlation metrics ($R^2$ and p-value) to compare the evolution of Lake DM, Lake A and observation well VP during the study period.

[Figure]

**RC3-48b.** Does the water level measured at VP ever exceed ground level?
**Answer: Clarification.** The maximum observed water level at VP (24.04 m.a.s.l.) did not exceed ground level (25.2 m.a.s.l.).

**RC3-49.** Line 267: What is this natural threshold? It needs to be clearly explained earlier in the manuscript.
**Answer: Done.** As explained in response to RC3-22, the topographic threshold was determined from a land survey and corresponds to the maximum elevation along S2 streambed. This information was added to the manuscript (Section 2.1).

**RC3-50.** Line 273: What I take from the manual water level measurements is that 1) the water level of Lake A is always higher than the groundwater level measured at VP and 2) Lake A and Lake DM do not always have similar water levels, regardless of whether the water level is above or below the topographic threshold. It would be useful to explain these aspects in the context of the Lake A water balance developed in S2.
**Answer: Done.** The reviewer is correct – the water level of Lake A is 1) always higher than the one of VP and 2) not always similar to the one of Lake DM. To help clarify the conceptual model, we moved the "Hydrodynamics of the major flood event" in Sect. 2 (before presenting the conceptual model). Also, the considerations from the manual measurements were added to this section.

**RC3-51.** Line 275: Isn't the actual volume increasing?
**Answer: Done.** The statement at L275 was modified to: "From February 23, 2017 to May 8, 2017, an overall volume increase is observed at Lake A"

**RC3-52.** Line 281: Was a manual measurement of Lake A water level taken at the start of the study period? It isn't shown on Fig 3.
**Answer: Done.** No manual measurement of Lake A water level was taken on April 17, 2017.

**RC3-53a.** Line 282-286: Unclear the importance of the lake being dredged. Unless there is evidence of a low hydraulic conductivity layer around the edges of the lake, wouldn't you expect hydraulic connection when the lake is situated in alluvial sands? Perhaps rephrase to indicate that gross water fluxes are likely to exceed net water fluxes due to the surrounding geology and measured hydraulic gradients.
**Answer: Done.** We agree with the reviewer. At L282-286, our intention was to point out that groundwater contribution can be expected for lakes sitting in permeable sediments. To clarify, we reformulate L282-284 to:

[revised manuscript text omitted]

**RC3-54.** Line 289: It is misleading to say that the shaded area represents flood water inputs. Won't surface water inputs from Lake DM to Lake A be received at all times that the water level of Lake DM is above the topographic threshold? Clarify throughout document and in the figure.
**Answer: Done.** Two conditions are needed for Lake A to receive surface water inputs from Lake DM. First, Lake DM water level needs to exceed the topographic threshold. Second, the water level of Lake A needs to be lower than the one of Lake DM. The direction of the surface water fluxes between Lake A and Lake DM was clarified in the conceptual model description (see response to comment RC3-53b). Additionally, we revised the period descriptors, as suggested in RC3-26. This was modified in the figure and throughout the document.

**RC3-55.** Line 298: Why 2016? The rest of the sampling was conducted in 2017.
**Answer: Clarification.** The regional amount-weighted mean $\delta_P$ is calculated from the precipitation volume and isotopic composition. While the isotopic composition of precipitations was available for the study period (February 2017 to January 2018), the volume measurements stopped in October 2017. Hence, the regional amount-weighted mean $\delta_P$ could only be estimated for 2016. More importantly, the yearly estimates for amount-weighted mean isotopic composition of precipitations ($\delta_P$) are likely to vary from one year to the other, and a long-term amount-weighted mean is more representative of the regional groundwaters (which mean age is expected to be > 1 year). This is why we compared the 1-year estimate from St-Bruno to the long-term estimate from Ottawa. However, we conceive that it may be confusing to present this information next the isotopic signature of precipitations. Hence, we opted to modify this statement and now compare the long-term amount-weighted means at Vaudreuil (27 km W from the study site) and at Ottawa (140 km W from the study site) with the $\delta_G$ estimated from the intersection of the LMWL and the Lake A LEL (see response to comment RC3-58).

**RC3-56.** Line 309: What is the justification for this? Do the three sample indicate temporal change in the flood-water inputs? How would this affect the calculated water budget?
**Answer: Clarification.** There is a temporal evolution of the isotopic signature of Lake DM, as demonstrated by Rosa et al. (2016). On an annual timescale, the evolution of the isotopic signature of Lake DM is mainly governed by evaporation. During springtime, the evolution is also expected to be controlled by the contribution of the snowmelt water – the snowmelt quantity and isotopic signature can evolve over time – and an enrichment of the isotopic signature of flood-water can be expected during springtime.

In our study, three flood-water samples were collected. The two most depleted samples were collected on April 19, 2017, while the most enriched flood-water sample was collected on May 10, 2017. The long-term (1997-2008) average, minimum and maximum isotopic signature of Ottawa River water at Carillon (~34 km upstream from Lake DM) for the month of April are -11.19 ‰, -

12.01 ‰ and -10.23 ‰ for $\delta^{18}O$ and -81 ‰, -85 ‰ and -77 ‰ for $\delta^2H$, respectively (Rosa et al., 2016). The mean and minimum values compare well with the observed isotopic signatures at Lake DM in our study.

As we collected three flood-water samples, it was difficult to correctly interpret the temporal evolution of the flood-water isotopic composition from late February to May. Hence, we opted to select a constant value of -12.00 ‰ for $\delta^{18}O$ and -83 ‰ $\delta^2H$ (i.e., the intersection between the LMWL and the flood-water regression line). The selected isotopic signature is assumed to be representative of the flood-water inputs in the earlier stages of the flood event, i.e., when the evaporation is null. This approach is conservative, as it estimates a minimal flood-water contribution to the lake water budget.

**RC3-57.** Line 310: Similar to what? Unclear why this is relevant.
**Answer: Done.** We meant similar to the selected $\delta_{Is}$ ($\delta^{18}O$ = -12.00 ‰ and $\delta^2H$ = -83 ‰). To clarify, we corrected L310 to:

> *"The long-term (1997-2008) average, minimum and maximum isotopic signature of Ottawa River water at Carillon (~34 km upstream from Lake DM) for the month of April are -11.19 ‰, -12.01 ‰ and -10.23 ‰ for $\delta^{18}O$ and -81 ‰, -85 ‰ and -77 ‰ for $\delta^2H$, respectively (Rosa et al., 2016). The mean and minimum values compare well with the observed isotopic signatures at Lake DM during springtime 2017."*

**RC3-58.** Line 313-Line 320: Suggest considering Jasechko et al. (2017) (doi: 10.1002/hyp.11175) and Welch et al. (2018) (doi: 10.1002/hyp.11396), which demonstrate widespread cold season bias to groundwater recharge in similar climates. Do you have any isotopic data from either of the observation wells?
**Answer: Done.** We are grateful to the reviewer for this suggestion as it helped strengthening our message. We reformulated Line 313-320 and now consider Jasechko et al. (2017) work's:

> *"The isotopic composition of groundwater ($\delta_G$) can be determined from direct groundwater samples or indirectly from the amount-weighted mean $\delta_P$. However, in highly seasonal climates, there is a widespread cold season bias to groundwater recharge (Jasechko et al., 2017), and estimating $\delta_G$ via groundwater samples or amount-weighted mean $\delta_P$ may be misleading. In fact, it has been argued that the LMWL-LEL intersection better represents the isotopic composition of the inflowing water to a lake and is thus commonly used to depict $\delta_G$ in isotopic mass balance applications (Gibson et al., 1993; Wolfe et al., 2007; Edwards et al., 2004). Concerning the study site, the estimated $\delta_G$ is -11.26 ‰ for $\delta^{18}O$ and -77 ‰ for $\delta^2H$ (i.e., the St-Bruno LMWL and Lake A LEL intersection). The latter compares well with the mean isotopic signature of groundwaters at Vaudreuil station (-11.1‰ for $\delta^{18}O$ and -78.5‰ for $\delta^2H$) (Larocque et al., 2015) and is more depleted than the long-term amount-weighted mean $\delta_P$ at Ottawa (-10.9‰ for $\delta^{18}O$ and -75‰ for $\delta^2H$) (IAEA/WMO, 2018)."*

Additionally, we used the suggested reference (Welch et al., 2018) in the revised introduction. Modifications were made to better contextualize our study (see response to comment RC3-8).

Finally, we do have isotopic data at the observation wells. However, we do not think it would help to discuss the cold season bias, as they are under the influence of the lake waters.

**RC3-59.** Line 330: Did you perform significance testing to determine this? If yes, present the results. If not, change the wording.
**Answer: Done.** No significance testing was done. The term "significantly" was deleted.

**RC3-60.** Line 357: What are these scenarios (A and B) and what do they represent? It is difficult to interpret the following results without understanding this.
**Answer: Done.** Scenario A and B are two different simulations of the evolution of the lake isotopic signature ($\delta_L$). In both scenario, the modelled $\delta_L$ was fitted on three depth-averaged $\delta_L$ (February 9, 2017, August 17, 2017 and January 25, 2018. The modelled $\delta_L$ was additionally constrained by a surface water sample during springtime, which is deemed to be representative of the well-mixed water column. In scenario A, we use the sample at the surface of Lake A on May 9-10, 2017 (i.e., $\delta^{18}O \approx$ -11.20 ‰ and $\delta^2H \approx$ -76 ‰) to constrain the model. Similarly, the April 27, 2017 sample is used to best-fit $\delta_L$ in scenario B. This was clarified in the manuscript, as well as in Table 1 and Figure 6.

**RC3-61.** Line 369: Why do you consider these have stopped? Fig 3 indicates that the level in Lake DM remains higher than Lake A until Lake A measurements cease.
**Answer: Clarification.** See response to comment RC3-53b.

**RC3-62.** Line 375: How do you reconcile this with the fact that inspection of Fig 6 indicates that the isotopic match is closer for Scenario B?
**Answer: Clarification and done.** As clarified in response to comment RC3-60, the modelled $\delta_L$ was fitted on four points. In Figure 6, the red squares correspond to the depth-averaged $\delta_L$ that were used to fit the modelled $\delta_L$. A revised version of Figure 6 now better illustrate the four values that were used to model $\delta_L$ for both scenarios. Additionally, the calculation of RMSE ($n = 4$) indicate similar matches for both scenarios.

**RC3-63.** Line 391: Consider moving the definition of mean flushing time to the methods. It is referred to extensively in the following pages and is hard to find here buried in a paragraph. The flushing time for Scenario B is approximately 30% lower than for Scenario A – is this not a fairly large difference?
**Answer: Clarification and done.** The definition of mean flushing time was moved to the methods. While we acknowledge the difference between the calculated flushing time for scenarios A and B, we think the results should be considered in the context and objectives of the study. Here, the estimates of the mean flushing time serve as a bulk parameter to discuss the resilience (and vulnerability) of the lake to changes, and a first-order estimate is appropriate to do so. Although there are differences between the two scenarios, both suggest that mean flushing time is within few months.

**RC3-64.** Line 401: Suggest including explanation provided to reviewer that this sensitivity analysis was conducted OAT.
**Answer: Done.** L402 was modified and can now read as:
> *"A one-at-a-time (OAT) sensitivity analysis was performed to grasp the relative impact of the input parameters' uncertainties on the model outputs."*

**RC3-65.** Line 404: This range of $\delta$Is does not cover the range of observed flood water inputs.
**Answer: Clarification.** The reviewer is correct – the tested range in the OAT sensitivity analysis does not cover the range of the three observed flood water inputs. However, it does cover the range of the two samples that were collected during the flood-water control period (see response to comment RC3-56), i.e. the period during which we conceptualized that flood-water inputs contribute to the lake water budget.

**RC3-66.** Line 408: Given the heat capacity of water, it seems unusual to use this as the only boundary for varying water temperature.

**Answer: Clarification.** The water surface temperature (T) was not measured continuously and was thus estimated based on the equilibrium method as described by de Bruin (1982), in order to take into account the heat capacity of water. In the OAT sensitivity analysis, we aimed at testing the robustness of the model against this estimated T. To do so, we selected a worst-case scenario, i.e., assuming that the $T=T_{air}$. As the results are similar to the reference scenarios, we concluded that T is not a sensitive parameter.

**RC3-67.** Line 416: As previously mentioned, this is not the only time when a hydraulic connection appears to form. How does amending this assumption affect the results?
**Answer: Done.** We are grateful to the reviewer for this comment. It made us realize that this statement was confusing. L416 was corrected and we can now read: *"As expected, the value of $\delta_{Is}$ is affecting the modelled $\delta_L$ exclusively during the flood-water control period."*

More isotopic data at Lake DM would have been necessary to correctly predict the impact of flood-water inputs at other times.

**RC3-68.** Line 420: It would be helpful to state specifics for the change in LMWL and δIs/g and water balance.
**Answer: Done.** The calculated LMWL with the PWLSR method and the estimated $\delta_{Is}$ and $\delta_G$ were added to L412, which can now read as:
> *"Using the PWLSR method, the LMWL is defined as $\delta^2H = 8.28 * \delta^{18}O + 17.73$, and $\delta_{Is}$ and $\delta_G$ are estimated at -12.39 ‰ and -11.74 ‰ for $\delta^{18}O$ and at -85 ‰ and -79 ‰ for $\delta^2H$, respectively. Recalculation of $\delta_{Is}$ and $\delta_G$ was needed, as they were both assumed to plot on the LMWL (see Sect. 4.1)."*

**RC3-69.** Line 421: What are the impacts on the water budget of holding δIs/g constant? This at least needs to be discussed as a study limitation. Similarly, the assumption of a well-mixed lake needs to be discussed, given the obvious isotopic stratification.
**Answer: Clarification and done.** See response RC3-56 for expected impact on the water budget of holding $\delta_{Is}$. The limitations concerning the evolution of $\delta_{Is}$ and the assumption of a well-mixed lake are discussed in section 5.2. See response to comment RC3-73 for added material concerning the isotopic signature of groundwater.

**RC3-70a.** Line 435: I suggest deleting this reference to the discussion and including analysis of the likely temporal change in groundwater inputs here in the results (Lines 448-470).
**Answer: Done.** L448-470 is now included in the results.

**RC3-70b.** Perhaps you could consider modifying your model to include two different groundwater inputs – one that reflects the regional groundwater, and one that reflects the mixture between flood-water inputs through the floodplain or bank storage and this regional groundwater.
**Answer: Clarifications.** While we agree that this approach would have been interesting, this is beyond the scope of our study. No data was available to characterize the potential spatiotemporal variability of the groundwater isotopic composition.

**RC3-70c.** Consider also doing a simple calculation to estimate the potential volume of flood water that could be stored in the subsurface using values for alluvial sands available in the literature (if there are no local measurements) and the maximum depth that the groundwater level lowers in the dry season.
**Answer: Clarifications.** The reviewer suggests to "estimate the potential volume of flood-water that could be stored in the subsurface". While we agree that such calculation can be useful as a "reality check", there are no information concerning the depth of the groundwater level in the

vicinity of Lake A. Also, in the context of this study, it is known that part of the groundwater outputs is pumped by the bank filtration system or discharges into Lake B. Hence, it is difficult to estimate what volume is stored in the subsurface.

**S5 Discussion**
**RC3-71.** Line 509: "expected increases in water levels…" Which water levels?
**Answer: Clarification.** We mean that river water levels are expected to increase. We reworded (L508-510). It now reads as:

> "*In Quebec (Canada), river stages are expected to increase across various watersheds in response to future climate scenarios (Roy et al., 2001; Minville et al., 2008; Dibike and Coulibaly, 2005).*

**RC3-72.** Line 521: It is unclear how this study tracks human impacts on the water cycle.
**Answer: Done.** We corrected "water cycle" to "water resources" at L520 and L521.

**RC3-73.** Line 535: Another big missing piece in the data currently presented is the lack of water level measurements around the lake and through time that support the hypothesis that groundwater discharges into the lake. See also comment on Line 421.
**Answer: Done.** We agree with the reviewer that such information should be added as another strategy to improve the effectiveness of our approach. The following material was added to the manuscript at L535:

> *"Groundwater level monitoring and groundwater sampling in the vicinity of the lake could also help to strengthen the conceptual model by providing data to interpret the direction of groundwater fluxes and the variability of isotopic composition through time."*

**S6 Conclusions**
**RC3-74.** This section can be made more succinct. Focus on the major findings. No need to repeat volumes.
**Answer: Done.** We made an effort at addressing this issue. The conclusion is now more succinct and focuses on the major findings. More precisely, we removed the information concerning the modeling methods (L541-543), 2) and the volume estimates (L554-557). We also reworded L548-549 and L557-560 in order to be more concise.

**Report #2**

Submitted on 19 Jan 2021
Referee #4: Matthew D. Jones, matthew.jones@nottingham.ac.uk

**Suggestions for revision or reasons for rejection (will be published if the paper is accepted for final publication):**

I enjoyed reading the manuscript by Masse-Dufresne and colleagues.

The paper outlines some interesting and potentially important reasons for understanding ground- and surface-water contributions to lake systems, and makes a case for the use of stable isotopes in modelling these systems.

I particularly liked the cross sections and three dimensional schematics of the lake isotope system, and the linking through of the importance of water quantity, and where it comes from, to issues of water quality.

I think the paper is suitable for publication after a few minor revisions, largely to help further clarify the modelling approach taken such that others can reproduce this approach for similar systems.

**RC4-1.** In general it would be interesting to know how this model was run i.e. in what software or using what code. Including a link to the code, or an example of the working through the equations which would help reviewers and readers understand more fully the steps taken.
**Answer: Done.** The model was implemented in Matlab®. As isotopic mass balance model are relatively easy to code or program in various softwares (Excel®, Matlab®, R) and as many isotopic models have been published already by other authors, we think that our code is not a major contribution to the scientific community. Hence, we opted to specify that "the code and data are available on request to the corresponding author".

My more specific points relate to the isotope hydrology work:

**RC4-2.** Equation 3 – please define delta-0.
**Answer: Done.** We added the definition of $\delta_0$ to the manuscript.

**RC4-3.** Line 259 – I think Fig. 3 is meant here?
**Answer: Done.** The reviewer is correct. However, we now present the "Hydrodynamics of the flood event" in section 2.2, and the figure is now numbered as Figure 2.

**RC4-4.** Section 4.2 – where are St-Bruno station and Ottawa in relation to the lake. Could you add these to Figure 1, or state distance and direction to/from lake for each site.
**Answer: Done.** The distances between the meteorological stations and the study site have been added to the manuscript (in section 3.1). Additionally, we modified Figure 1 and added the location of the different stations and cities.

**RC4-5.** Figure 4 and its discussion – the flood waters are from Lake DM (as noted in Fig. 5) so is the Flood-water LEL the Lake DM LEL? Is it the lake waters in Lake DM that are more directly affected by snow melt, or would you class the flood waters as separate? Is there any stable isotope data from Lake DM?
**Answer:** The flood-water samples do correspond to the Lake DM waters. Hence, the Flood-water LEL ($\delta^2H = 5.68 * \delta^{18}O - 12.80$) should correspond to the Lake DM LEL during springtime, but the

latter could differ for other periods as the governing parameters (e.g., RH, T) may evolve over time.

**RC4-6.** Figure 6 – may be worth changing the legend label for 'delta-L (observed)' – to make it clear that this is an average (mean?) of the blue dots – this is what I interpreted it as anyway.
**Answer:** The reviewer is correct. It was corrected to "depth-averaged".

**RC4-7.** As well as showing the range of water isotope data with depth in table 2 and Fig. 6 it would be interested to plot the profiles in an appendix.
**Answer: Done.** A figure showing the isotopic data with depth was added to Appendix E.

[Figure]

**RC4-8.** If useful another example of isotope mass balance modelling to understand water balance in artificial lakes (gravel-pits in this case) can be found in:
Jones, M.D. et al., 2016. Comparisons of observed and modelled lake $\delta^{18}O$ variability. Quaternary Science Reviews 131, Part B, 329-340.
**Answer:** We are grateful for this suggestion. We added the reference in section 4:
> *"Indeed, gross water fluxes are likely to exceed net water fluxes at natural and dredged lakes sitting in permeable sediments (Zimmermann, 1979; Arnoux et al., 2017a; Jones et al., 2016)"*

---

## Author Response (AR3)

Comments to the Author:
Dear authors,

Thank you for your patience during this review process. I have now heard back from the two reviewers who looked at the last two versions of your manuscript. Both of them commended you on the comprehensive nature of your last revisions. One reviewer has no further suggestions, while the other has a list of minor elements they suggest that you clarify or nuance in your manuscript. One of their remaining concerns is what you present as evidence of bank storage, which they do not find fully satisfactory. To address that concern, you may either choose to change your terminology surrounding bank storage throughout the whole manuscript, or provide additional justification and evidence using your isotopic data (in the absence of hydraulic data). Should the above-mentioned options (terminology change or isotope-based justification) not be possible, then you should probably nuance your bank storage-related findings by presenting them as first-order flux estimates or hypotheses, and discuss the uncertainty associated with them.

I am returning your manuscript for minor revisions and look forward to receiving the new iteration of your manuscript, together with your responses to reviewer comments (which I will evaluate).

With best regards,

Genevieve Ali

**Response to Editor**

Dear Prof. Ali,

We have considered all the comments and suggestions from the reviewer and revised the manuscript accordingly. To address the main remaining concerns (evidence of bank storage and the use of this term), we opted to i) change the terminology (with temporary subsurface storage), ii) provide additional simulations (to better link the potential flood-water-like subsurface inputs to the modelled isotopic signature of the lake) and iii) emphasize the first-order nature of the water flux estimates throughout the manuscript.

In addition to the reviewer's comments, we modified the definition of the flushing time ($t_f$). In the former version of the manuscript, we defined $t_f$ as the ratio of the lake volume to the total water inputs ($t_f = V/I$). In the revised version, we propose to specifically consider the flushing time by groundwater ($t_f = V/I_G$). This modification was deemed necessary to correctly compare our results with literature data (as in Arnoux et al., 2017b). Accordingly, modifications were made to Eq. (4) at Line 281 and throughout Sect. 5.1.

Also note that we checked for typos and have corrected two co-authors affiliations. These changes are indicated in the marked version of the manuscript.

Sincerely,

Janie Masse-Dufresne

**Report #1**

Submitted on 13 May 2021
Referee #3: Chani Welch, cwelch@okanagan.bc.ca

**Suggestions for revision or reasons for rejection (will be published if the paper is accepted for final publication):**

The manuscript requires further support of the process of bank storage, and further acknowledgement of the first-order nature of the flux estimates.

Hello,
This manuscript has vastly improved with the reorganisation and clarification provided. I have two general concerns remaining:

**RC3-1.** The link to bank storage is weak in the modelling and results, but prominent in the findings. The available hydraulic data do not provide supporting evidence (the groundwater level is not recorded above the water level of Lake A). I understand there were technical issues with the hydraulic measurements, so perhaps more use could be made of the isotopic model, or a clearer link provided to the hypothetical scenarios presented for the G-index. For example, can you get a good fit to the isotopic data with a groundwater input that has an isotopic composition intermediate to flood water and groundwater in the May – August time period?

**Answer: Done.** We are grateful to the reviewer for this comment. As suggested, we modelled $\delta_L$ when considering that groundwater inputs are characterized by the isotopic signature of flood-water ($\delta_{Is}$) after the flooding event. We modelled four hypothetical scenarios for which 25%, 50%, 75% and 100% of the outflows of the lake during the flood-water control period were stored in the subsurface and eventually discharged back to the lake (from May to August). It is noteworthy that a better fit is obtained between the modelled $\delta_L$ and depth-averaged $\delta_L$ when considering that 25% to 50% of the potentially stored flood-water returns to the lake. This result further supports our conceptual model and the fact that considering flood-water-like groundwater inputs is important when assessing water balances.

We thus modified Fig. 6 to illustrate these hypothetical scenarios. Note that the order of Sect. 4.3 (Temporal variability in the water balance partition) and Sect. 4.4 (Importance of bank storage on the water balance partition) has been reversed to accommodate for this modification. Also, Sect. 4.3 (formerly 4.4) was modified to discuss the results concerning the impact of flood-water-like groundwater inputs on the modelled $\delta_L$, while considerations regarding the evolution of the G-Index were merged to Sect. 4.4 (formerly 4.3).

**RC3-2.** The isotopic data presented in Appendix E (and Figure 6) clearly indicates that the lake is not well-mixed in August (the only depth profiles outside of winter). Arguments supporting the well-mixed model choice are provided; however, I do not think the first-order nature of the estimate of water fluxes is clearly emphasized throughout the manuscript (for example, it is not mentioned in the abstract).

**Answer: Done.** As suggested, we opted to specify the first-order nature of the water flux estimate at Line 17 and to state a range for the relative contribution of groundwater and surface water inputs to the lake (based on scenarios A and B) at Line19 (in the abstract). It was additionally specified at Line 402 and Line 469.

Please note that the listed line numbers in this response refer to the revised version of the manuscript.

***Other minor comments are provided below.***

**RC3-3.** Line 21: Lake A water budget?
**Answer: Done.** It was modified to "Lake A water budget".

**RC3-4.** Line 22: This is contradicted in the discussion.
**Answer: Clarification and done.** At L22, we refer to the resilience of the lake obtained from the reference scenarios A and B, which only considers the direct flood-water inputs. Then, we discuss the potential impact of considering subsurface flood-water-like inputs on the resilience of the lake to groundwater changes (see downward arrow in Fig. 8). To avoid any confusion, we opted to reword L20-24. It now reads as:
> *"However, when considering the potential temporary subsurface storage of flood-water, the partitioning between groundwater and surface water inputs tends to equalize, and Lake A water budget is found to be more resilient to groundwater quantity and quality changes."*

**RC3-5.** Line 48: Delete "nearly impossible". The reference doesn't say it's impossible. Just time-consuming, expensive, and difficult.
**Answer: Done.** It was modified to "time-consuming, expensive, and difficult". Note that it was also deleted in the abstract (at Line 14).

**RC3-6.** Line 54: This paragraph would benefit from a topic sentence before diving into the specifics of previous studies.
**Answer: Done.** We are grateful to the reviewer for this suggestion. The following sentence was added before the paragraph:
> *"To gain information on the timing of hydrological processes, one may use a transient and short time step isotopic mass balance."*

**RC3-7.** Line 66: I remain unconvinced that bank storage is the correct term – perhaps a more general delayed (in subsurface) and direct flood water inputs?
**Answer: Done.** We opted to modify for "direct flood-water inputs" vs "temporary subsurface storage of flood-water" at L66 and throughout the manuscript (including the 4.4 sub-section title). To better define these terms in the context of this study, we also added the following material at L53 (after the first mention of "temporary subsurface storage of flood-water", except for the abstract).
> *"In this study, we define the direct inputs refer to the flood-water that enter a lake via the surface (e.g., by inundating and/or flowing through a stream), while temporary subsurface storage of flood-water encompasses the flood-water-like inputs that reach the lake via subsurface (e.g., through floodplain recharge or bank storage)."*

**RC3-8.** Line 72: Flood water storage – surface (in the lake) or subsurface?
**Answer: Done.** To avoid any confusion, and in line with other comments (RC3-8), we opted to modify "implications of flood-water storage on the water balance" to:
> *"implications of flood-water-like subsurface inputs on the water balance partition".*

**RC3-9.** Line 81: "a" not "an"
**Answer: Done.** It was corrected to "a".

**RC3-10.** Line 100: Rephrase: The direction of the surface water flux in S2 reverses when the water level in Lake DM exceeds… Does flow reversal also occur in S3?

**Answer: Done.** Flow reversal also occur in S3, but no topographic land survey was conducted along this stream. Hence, the elevation of the topographic threshold is unknown. To clarify, we added the following material at L102:

> *"Flow reversal also occurs in S3, but the elevation of the topographic threshold is unknown."*

**RC3-11.** Line 124: "Tin Lake A" – rephrase or define.
**Answer: Done.** It was corrected to: "The water level in Lake A".

**RC3-12.** Line 125: Define the observed period here at first mention.
**Answer: Done.** We defined the observed period at L125 (April 27th, 2017 to May 17th, 2017).

**RC3-13.** Line 127: Suggest rephrasing - It remains an assumption that Lake DM controls the water level variation at obs well VP. Perhaps "the data indicates". To the reader it is unclear why the Lake A water level is not used – state here that the logger broke.
**Answer: Done.** It was rephrased to "the data indicates". Also, the following material was added to L128 to specify that logger in Lake A broke and prevented comparison:

> *"Lake A water level was also presumably controlled by Lake DM until late July 2017, but technical issues prevented confirmation (i.e., logger in Lake A broke on May 17th, 2017)."*

**RC3-14.** Line 131: Suggest rephrasing to say that Lake A water level is not controlled by Lake DM in this period.
**Answer: Done.** It was rephrased to:

> *"It is thus possible to infer that Lake A water level was not controlled by Lake DM from August 2017 to late October 2017."*

**RC3-15.** Line 137: Figure 2 indicates the water level in Lake A is below Lake DM in mid Dec (even considering error bars shown) – why wouldn't the water flow into it? Maybe it only happens for a short time (which could explain the lower correlation)
**Answer: Clarification and done.** The reviewer is correct – the water level in Lake A is below Lake DM in mid Dec. However, it is uncertain whether a hydraulic connection can establish during wintertime (due to freezing of the stream). To avoid misleading interpretation, we corrected L137 to:

> *"It is thus likely that Lake A received little to no surface water inputs from Lake DM from November 2017 to January 2018. In this context, surface water inflow from Lake DM during autumn and winter are considered negligible in this study and not included in the developed stable isotope mass balance model (Sect. 4.2)."*

**RC3-16.** Line 151: Could there not also be Qg out the southern end of Lake A? Ie in the direction of regional groundwater flow.
**Answer: Clarification and done.** The reviewer is correct – there is a regional groundwater flow in the NE-SW direction. It is noteworthy that this information does not appear in this version of the manuscript. It was only mentioned in response to the reviewer (last revision round, RC3-34). We opted to add the following material at L150:

> *"Given the regional groundwater flow in the NE to SW (Ageos, 2010), QG can also presumably occur along the SW bank of Lake A."*

**RC3-17.** Line 157: Rephrase to put Lake DM immediately after Qs otherwise this sentence reads as if Qg is possible going to Lake DM, which contradicts Line 150.
**Answer: Done.** It was rephrased accordingly to the reviewer suggestion.

**RC3-18.** Line 161: If the water causing the water level increases is not coming from Lake DM, then where is it coming from? Direct precipitation? S1?

**Answer: Clarification and done.** We are grateful to the reviewer for this comment. Some clarification was needed. L161 was corrected to:

> "In this context, we conceptualized that Lake A water level variations are mainly controlled by groundwater flows ($I_G$ and $Q_G$). Surface water inputs ($I_S$) are set to zero during this period (see Sect. 2.2)."

**RC3-19.** Line 274: The manuscript first argues that there is not a strong correlation, and then use Lake DM as representative? I get that the absolute level doesn't matter, but why not just use the obs well the whole time? Does this change the results?

**Answer: Clarification.** First, water level in the observation well VP was not available for the whole simulated period. The water level in Lake DM is the only available measurement for Feb 2017 to late March 2017. Second, during the period of high-water level (late April 2017 to mid-May 2017), there is a good correlation between Lake DM and Lake A water levels. During this period, the daily water level variations at observation well VP were less important, and using observation well VP as a proxy would have resulted in an overestimation of the surface and groundwater fluxes during springtime. Concerning the autumn-winter period (November 2017 to late January 2018), the reviewer is correct – the correlation between Lake DM and observation well VP water levels are lower. If the water level at the observation well VP is used as a proxy for Lake A for the autumn-winter period, the simulated water fluxes would be lower and the modelled $\delta_L$ would be more enriched. While using VP as a proxy during this period does not significantly impact the modelled $\delta_L$ in scenario A, it does slightly affect scenario B. To obtain a good fit between the modelled $\delta_L$ and depth-average $\delta_L$, the $Q_{min}$ can be adjusted to $1.5 \times 10^4$ m³/d (instead of $1 \times 10^3$ m³/d). Although the simulated fluxes (Q and I) increase, the share between $I_S$ and $I_G$ (40% - 60%) and the $t_f$ (103 days) remain similar to the reference scenario B (40% - 60%; 110 days).

**RC3-20.** Line 283: Are there 2 Penman-48 methods? Suggest rephrasing as it is unclear which is used in the model, the one that underestimates or the one that doesn't. If it's the one that doesn't underestimate in late summer-fall, the results are unlikely to be affected. Otherwise: Line 284: how did you resolve this? Specific heat capacity of water. Does this markedly affect the isotopic balance and interpretation?

**Answer: Done.** We opted to better specify the "standardized Penman-48" and the "simplified Penman-48" equations in the manuscript. In the model, we use the "standardized Penman-48" equation – the one that does not underestimate E in late summer-fall.

**RC3-21.** Line 291: "The outflows of the lake are thus…" I still find these comments misleading. I suggest rewording to explain that the change in outflows from the lake is roughly proportional to the change in water level.

**Answer: Done.** We clarified L290-293. This now reads as:

> "The direction and intensity of the water flux at the lake-aquifer interface can be conceptually described by Darcy's Law which states that Q = KAi, where K is the hydraulic conductivity, A is the cross-sectional area through which the water flows, and i is the hydraulic gradient. Given the significant depth of Lake A (i.e., 20 m) in comparison to the maximum water level change during the flooding event (i.e., 2.7 m), the variation of the A and K are expected to have minor impact on Q. Hence, the change in outflows from the lake is expected to be mainly controlled by i changes and, consequently, to be roughly proportional to the change in lake water level"

Note that L367-369 was also modified according to the reviewer's suggestion.

**RC3-22.** Line 302: Suggest inserting "it is assumed that" immediately before "the rising water level". Groundwater may still be entering lower in the lake.
**Answer: Done.** The reviewer is correct. We modified L302 accordingly to the reviewer's suggestion.

**RC3-23.** Line 344: Does this use of the LMWL-LEL account for flood waters? This assumption will likely only hold where groundwater is sourced from the local precipitation, and is not subjected to different rates of evaporation during infiltration. It is curious to me that no isotopic data collected from the groundwater wells at the site is presented to support this choice.
**Answer: Clarification.** Please see response to comment RC3-24.

**RC3-24.** Line 348: Where is Saint-Telesphore and why is it considered a useful comparison?
**Answer: Clarification.** St-Télesphore is located at ~43 km from the study site (Fig. 1b). It was used to compare the isotopic signature of groundwater in a similar context, i.e., where there is a widespread cold season bias to groundwater recharge. Such comparison was added to the manuscript in response to a previous comment (last revision round, RC3-58). Please note that this sampling station was first named "Vaudreuil" in the response to the reviewer. It was corrected to "Saint-Télesphore" in the last revised version of the manuscript.

Note that the following was added to L349:
*"Note that the location of Saint-Télesphore station and Ottawa are depicted in Fig.1b."*

**RC3-25.** Line 377: depth-average"d" – the "d" is missing
**Answer: Done.** It was corrected.

**RC3-26.** Line 381-384: Unclear where the dates and isotopic values come from until reading the caption to Table 1. It would help the reader if the scenarios were more clearly presented and their purpose more clearly articulated – ie is it to compare lengths of flood-water control?
**Answer: Done.** We made efforts at reorganizing the paragraph to better explain that one of the two samples collected at the surface of the lake may be representative of the whole water column. This now reads as:
*"While depth-averaged $\delta_L$ was not available during the flood-water control period (i.e., late February to early May), water samples from the surface of Lake A provide relevant evidence to better constrain the model. It is likely that Lake A was fully mixed during the flood-water control period, and that the water samples collected at the surface of Lake A on April 27th, 2017 or May 9-10th, 2017 are representative of the whole water body. Indeed, the observed surface water temperature was < 5°C until early May (see Fig. C1) and suggests a limited density gradient along the water column which does not allow for the development of thermal stratification. In this context, we opted to simulate two scenarios (A and B), for which the isotopic mass balance model is either constrained at $\delta^{18}O$ = -11.20 ‰ and $\delta^2H$ = -76 ‰ on May 9-10, 2017 or at $\delta^{18}O$ = -11.86 ‰ and $\delta^2H$ = -80.68 ‰ on April 27th, 2017.*

**RC3-27.** Section 2.2 Sensitivity analysis. Should this be Section 4.2.2? Given what the paper is aiming to do, this section would benefit from linking the effects of these changes to the different water fluxes.
**Answer: Done.** The reviewer is correct. It was modified to "4.2.2". Also, we added links between the uncertainties associated with the input variables and the flux estimate changes. This further helps to underline the first-order nature of the water flux estimates (see comment **RC3-2**).

**RC3-28.** Section 4.4 I'm struggling to reconcile this with the isotopic modelling – if the water discharging back to the lake as flood water, then wouldn't it have the isotopic signature of flood water? My understanding is that deltaG is held constant in the model. To test this, why not alter deltaG over this time period?

**Answer: Done.** The reviewer is correct – the $\delta_G$ is held constant in the model. Our intention to discuss the theoretical impact on the G-Index. However, we agree that our work could benefit from discussing the results when considering flood-water-like groundwater inputs after the flooding event. As mentioned in response to **RC3-1**, we did perform such simulation and the results were added to the manuscript (Fig.6, discussed in Sect. 4.3).

**RC3-29.** Line 577: This contradicts line 522 which categorises the lake as relatively resilient – clarify.

**Answer: Clarification and done.** At L577, we refer to the resilience of the lake obtained from the reference scenarios A and B and the associated scenarios considered in the sensitivity analysis, which only considers the direct flood-water inputs. At L522, we discuss the potential impact of considering the temporary subsurface flood-water inputs on the resilience of the lake to groundwater changes (see downward arrow in Fig. 8). To avoid any confusion, we opted to reword L577-580. It now reads as:

*"Despite sensitivity to some variables, all model scenarios considered in the sensitivity analysis converged on the results that Lake A is mainly dependent on groundwater inputs and has a rapid (<1 year) flushing time by groundwater, suggesting that Lake A would be highly sensitive to groundwater quantity and quality changes.*

*When taking into account for potential subsurface storage, a better fit could be obtained between the modelled and depth-averaged isotopic signature of the lake, suggesting that the contribution of flood-water-like subsurface inputs is important to consider when assessing for water balance at flood-affected lakes. In fact, the increased contribution of surface water (from subsurface storage) resulted in a lower the contribution from groundwater and, consequently, in an increased resilience to groundwater changes."*